# FUSE: Fast Semi-Supervised Node Embedding Learning via Structural and Label-Aware Optimization

## Abstract

Graph-based learning is a cornerstone for analyzing structured data, with node classification as a central task. However, in many real-world graphs, nodes lack informative feature vectors, leaving only neighborhood connectivity and class labels as available signals. In such cases, effective classification hinges on learning node embeddings that capture structural roles and topological context. We introduce a fast semi-supervised embedding framework that jointly optimizes three complementary objectives: (i) unsupervised structure preservation via scalable modularity approximation, (ii) supervised regularization to minimize intra-class variance among labeled nodes, and (iii) semi-supervised propagation that refines unlabeled nodes through random-walk-based label spreading with attention-weighted similarity. These components are unified into a single iterative optimization scheme, yielding high-quality node embeddings. On standard benchmarks, our method consistently achieves classification accuracy at par with or superior to state-of-the-art approaches, while requiring significantly less computational cost.

## 1 Introduction

Graph-based learning has emerged as a powerful paradigm for analyzing structured data, with applications in social networks (Li et al., 2023), citation graphs (Luo et al., 2023), knowledge graphs (Ye et al., 2022), and recommendation systems (Lu et al., 2025; Anand and Maurya, 2024). A central task is node classification, where a subset of nodes are labeled and the goal is to predict the labels of the remaining ones (Luo et al., 2024). This task is typically facilitated by node embeddings $X \in \mathbb{R}^{|V| \times k}$ that capture graph structure (Xiao et al., 2021).

In practice, node embeddings may not be explicitly available, especially in newly constructed or rapidly evolving graphs, even when partial labels are known. Existing approaches often rely on unsupervised (Duong et al., 2023) or self-supervised (Veličković et al., 2019) embedding generation, or directly employ Graph Neural Networks (GNNs) such as GCN (Kipf and Welling, 2017), GAT (Veličković et al., 2018), and GraphSAGE (Hamilton et al., 2017) in a semi-supervised fashion. In addition, there are a few semi-supervised approaches that combine GNNs as encoders and customized classifiers to solve node classification problems (Lee et al., 2022; Yan et al., 2023). The given features are enhanced using these semi-supervised node representation algorithms. However, when embeddings are missing, initializing GNNs with random embeddings is ineffective for downstream tasks. A more efficient strategy is to generate structured initial embeddings via unsupervised or self-supervised approaches, and then refine them with GNNs (Hamilton et al., 2017; Weihua Hu et al., 2020).

We propose a fast semi-supervised embedding generation framework designed specifically for cases where node embeddings are unavailable. Our method integrates three complementary optimization components:

1. **Unsupervised structure preservation**, capturing global connectivity through a novel scalable approximation of graph modularity (Newman, 2006; Yazdanparast et al., 2021).

2. **Supervised regularization**, aligning labeled nodes within the same class via compactness constraints.

3. **Semi-supervised propagation**, refining unlabeled nodes using random-walk-based label propagation (Raghavan et al., 2007) combined with attention-driven similarity weighting (Wang et al., 2020).

By unifying these three components into a single iterative gradient ascent framework, our approach produces high-quality node embeddings quickly and without requiring pre-existing features. The fast convergence of the optimization procedure can make it well-suited to settings where labels are introduced incrementally, making it especially relevant in real-world applications such as recommendation (Pei et al., 2020), cybersecurity (Fang et al., 2022), and financial transaction monitoring (Bukhori and Munir, 2023), where embeddings must be updated on the fly.

We evaluate our approach on standard benchmarks including Cora (McCallum et al., 2000), CiteSeer (Giles et al., 1998), WikiCS (Mernyei and Cangea, 2020), Amazon Photo (McAuley et al., 2015), PubMed (Namata et al., 2012) and ArXiV (Hu et al., 2020). We compare against widely used unsupervised methods such as Node2Vec (Grover and Leskovec, 2016), DeepWalk (Perozzi et al., 2014), VGAE (Kipf and Welling, 2016), M-NMF (Wang et al., 2017), the self-supervised DGI (Veličković et al., 2019), two semi supervised baselines GraFN (Lee et al., 2022), ReVAR (Yan et al., 2023) and precomputed embeddings. Downstream classification performance is assessed using GCN (Kipf and Welling, 2017), GAT (Veličković et al., 2018), and GraphSAGE (Hamilton et al., 2017).

**Contributions.** Our main contributions are as follows:

1. We introduce a fast semi-supervised embedding generation algorithm that requires no pre-defined node embeddings.

2. In particular, we propose a linear time approximation of the graph modularity gradient, which is fundamental to our fast embedding generation process.

3. Notably, the algorithm uses labels if available, but can be adapted to scenarios where labels are completely unavailable with some compromise in performance.

4. We design a unified optimization framework that equally integrates unsupervised, supervised, and semi-supervised components.

## 2    RELATED WORK

Our approach connects to several lines of research: unsupervised embedding methods, self-supervised, semi-supervised embedding methods, graph neural network baselines, and modularity-driven optimization.

**Unsupervised node embedding.** Random-walk-based approaches such as DeepWalk (Perozzi et al., 2014) and Node2Vec (Grover and Leskovec, 2016) learn node representations by applying Skip-Gram training to sequences generated from biased or unbiased random walks. Variational Graph Auto-Encoders (VGAE) (Kipf and Welling, 2016) extend autoencoding approaches to graphs by using a GCN encoder with a latent Gaussian distribution, achieving strong results in unsupervised link prediction. Another method, M-NMF Wang et al. (2017) learn node embeddings by factorizing the graph structure without using any label information. It integrates both the network's local structure (e.g., adjacency information) and global community structure (e.g., modularity) into a joint factorization framework. These methods demonstrate that structural information alone can be leveraged to build embeddings, since they are agnostic to label information.

**Self-supervised learning.** Contrastive frameworks such as Deep Graph Infomax (DGI) (Veličković et al., 2019) maximize mutual information between local node embeddings and global summaries, enabling representation learning without labels. Other approaches (e.g., SL-GAT (Wang et al., 2020)) refine attention-based architectures with self-supervised objectives. These methods reduce the reliance on labeled data but typically incur significant computational overhead.

**Semi-supervised learning.** Semi-supervised methods like GraFN (Lee et al., 2022) and ReVAR (Yan et al., 2023) address the limitations of purely supervised or self-supervised graph learning by combining a small amount of labeled data with structural information. GraFN aligns class predictions across augmented graph views to improve class-discriminative representations by combining

self-supervised and label-guided methods, while ReVAR, which is specifically designed for imbalanced scenarios, introduces variance-based regularization to mitigate class imbalance.

**Graph neural networks for classification.** Semi-supervised GNNs such as GCN (Kipf and Welling, 2017), GAT (Veličković et al., 2018), and GraphSAGE (Hamilton et al., 2017) refine embeddings through message passing and neighborhood aggregation, making them effective classifiers once initial embeddings are provided. Recent surveys highlight their utility across domains including social networks (Li et al., 2023), knowledge graphs (Ye et al., 2022), and recommender systems (Lu et al., 2025; Anand and Maurya, 2024). However, initializing GNNs with random embeddings is ineffective for downstream tasks (Wang et al., 2025), motivating the need for fast strategies that generate embeddings from scratch. It is to be noted that throughout the tables provided, we used "SAGE" to represent GraphSAGE primarily due to space constraints.

**Connections of proposed objective to prior works.** Our proposed objective unifies three complementary components, each drawing inspiration from existing lines of research:

1. **Unsupervised structural component.** Modularity (Newman, 2006) and its scalable variants (Yazdanparast et al., 2021; Lu et al., 2018) have long been used for identifying communities in graphs. Neural formulations such as DGCLUSTER (Bhowmick et al., 2023) further relaxed modularity maximization into differentiable objectives. Inspired by this line of work, we design an unsupervised objective that preserves structural regularities while avoiding the computational overhead of spectral methods.

2. **Supervised label-aware component.** Semi-supervised GNNs such as GCN, GAT, and GraphSAGE (Kipf and Welling, 2017; Veličković et al., 2018; Hamilton et al., 2017) incorporate label signals during message passing to improve classification performance. We adapt this idea directly at the embedding generation stage, encouraging nodes with the same label to have structurally similar embeddings. This distinguishes our approach from prior GNN methods, which rely on node features.

3. **Semi-supervised propagation component.** Label propagation (Raghavan et al., 2007) and attention-based refinements such as SL-GAT (Wang et al., 2020) have demonstrated the ability to diffuse label information across the graph in a scalable way. We build on these insights by incorporating a random-walk-based propagation mechanism that guides the embeddings of unlabeled nodes toward those of reachable labeled nodes.

This work bridges these strands by proposing a fast semi-supervised algorithm that avoids dependence on node features while combining the strengths of unsupervised structural preservation, supervised label regularization, and semi-supervised propagation.

## 3 THE FUSE ALGORITHM

Our approach, Fast Unified Semi-supervised Node Embedding Learning from Scratch (FUSE) combines linearized modularity optimization with supervised regularization and semi-supervised label propagation to generate embeddings that are both structurally coherent and class-discriminative. We introduce a differentiable formulation of modularity that enables gradient-based optimization and integrate random walk-based propagation with attention to refine unlabeled node embeddings.

### 3.1 PROBLEM SETTING

Let $\mathcal{G}$ be a simple, undirected graph with nodes $V$, edges $E$, and adjacency matrix $\boldsymbol{A}$. Let the degree of a node $v \in V$ be $d_v$, and let the vector of degrees be $\boldsymbol{d}$. Also let $m = |E|$ and $n = |V|$. Consider the classification task, where each node $v \in V$ is associated with a label $y_v \in \mathcal{C}$.

Let us choose an embedding dimensionality $k \in \mathbb{N}$. Consider an arbitrary downstream classification model $f : \mathbb{R}^k \to \mathcal{C}$. Our objective is to learn an embedding map $p : V \to \mathbb{R}^k$ such that the performance of the downstream task $f \circ p$ is maximized. We will learn $p$ as a continuous embedding matrix $\mathbf{S} \in \mathbb{R}^{n \times k}$, where each row $\mathbf{S}_{i,:}$ denotes the $k \ll n$-dimensional embedding of node $i$, i.e., $\mathbf{S}_{i,:} = p(i)$.

## 3.2 LINEAR MODULARITY OPTIMIZATION

We want to model modularity-aware embedding generation for graphs with unknown features with the matrix $\mathbf{S}$. The modularity function can be equivalently written as

$$Q(\mathbf{S}) = \frac{1}{2m} \sum_{i,j} \left( A_{ij} - \frac{d_i d_j}{2m} \right) \mathbf{s}_i^\mathsf{T} \mathbf{s}_j, \tag{1}$$

which, in matrix form, reduces to

$$Q(\mathbf{S}) = \frac{1}{2m} \mathrm{Tr}\left( \mathbf{S}^\mathsf{T} \boldsymbol{B} \mathbf{S} \right), \tag{2}$$

where $\boldsymbol{B} = \boldsymbol{A} - \frac{\boldsymbol{d}\boldsymbol{d}^\mathsf{T}}{2m}$ is the modularity matrix (Newman, 2006).

**Gradient Approximation.** Differentiating w.r.t. $\mathbf{S}$ yields

$$\nabla_\mathbf{S} Q_{\text{exact}} = \frac{1}{m} \left( \boldsymbol{A}\mathbf{S} - \frac{1}{2m} \boldsymbol{d}(\boldsymbol{d}^\mathsf{T}\mathbf{S}) \right). \tag{3}$$

However, for enhanced numerical stability and computational efficiency, we employ the following gradient approximation:

$$\nabla_\mathbf{S} Q_{\text{prop}} = \frac{1}{2m} \left( \boldsymbol{A}\mathbf{S} - \frac{1}{2m} \boldsymbol{d}(\mathbf{1}^\mathsf{T}\mathbf{S}) \right), \tag{4}$$

where $\mathbf{1}^\mathsf{T}\mathbf{S} = \sum_i \mathbf{S}_{i,:}$ is the unweighted sum of all node embeddings. We show in Appendix B that the proposed gradient updates are never too large (i.e., the proposed gradient function has no singularities).

**Interpretation.** The proposed gradient has an intuitive interpretation:

- The term $\boldsymbol{A}\mathbf{S}$ performs a local aggregation, where each node's embedding is updated by summing the embeddings of its neighbors. This pulls nodes towards the center of their immediate community.
- The term $\frac{1}{2m}\boldsymbol{d}(\mathbf{1}^\mathsf{T}\mathbf{S})$ acts as a global correction. It estimates the expected connection strength under the configuration model but uses the unweighted global average embedding $\frac{1}{2m}\mathbf{1}^\mathsf{T}\mathbf{S}$ instead of the degree-weighted average. This pushes nodes away from the global center of the graph, enhancing the separation between communities.
- The factor $\frac{1}{2m}$ scales the entire expression to be comparable across graphs of different sizes.

This approximation replaces the degree-weighted mean $\boldsymbol{d}^\mathsf{T}\mathbf{S}$ in the exact gradient with the unweighted mean $\mathbf{1}^\mathsf{T}\mathbf{S}$. This simplifies the computation and often leads to more stable optimization, as it reduces the influence of high-degree nodes (hubs) on the global correction term, preventing their features from overly dominating the global statistics.

**Computational Complexity.** The main steps of sparse matrix multiplication $\boldsymbol{A}\mathbf{S}$ and degree corrections scale as $O(|E|k + nk)$ ($|E|$ being the number of edges), while supervised gradient updates remain linear in the number of nodes. The additional semi-supervised components add costs of $O(w\ell)$ for $w$ random walks each of length $l$, and $O(nd_{\max}k)$ for attention updates, $d_{\max}$ being the maximum possible degree of a node. Orthonormalizing the $n \times k$ embedding matrix per iteration incurs a cost of $nk^2$ which is dominated by the sparse matrix multiplication $O(|E|k)$ for moderate $k$. Thus, the overall complexity is $O(|E|k + nk + nd_{\max}k + w\ell + nk^2)$, which is more scalable than spectral methods that require $O(n^3)$ for eigen-decomposition.

## 3.3 SUPERVISED AND SEMI-SUPERVISED COMPONENTS

While modularity optimization preserves structural properties, it does not enforce label consistency. We therefore introduce supervised and semi-supervised components.

**Supervised regularization.** Given a set of ground-truth labels $\boldsymbol{y} \in \mathbb{R}^n$, we minimize intra-class embedding variance by defining the loss

$$Q_{\text{sup}} = \sum_c \sum_{i \in C_c} \|\mathbf{S}_{i,:} - \boldsymbol{\mu}_c\|^2, \tag{5}$$

where $\boldsymbol{\mu}_c = \frac{1}{|C_c|} \sum_{i \in C_c} \mathbf{S}_{i,:}$ is the class mean. The gradient is

$$\nabla Q_{\text{sup}} = \mathbf{S} - \tilde{\mathbf{S}}, \quad \tilde{\mathbf{S}}_i = \boldsymbol{\mu}_c \text{ for } i \in C_c. \tag{6}$$

This ensures embeddings of labeled nodes in the same class remain clustered.

**Semi-supervised label propagation.** For unlabeled nodes, we employ biased random walks (Raghavan et al., 2007) that preferentially visit labeled nodes, allowing labels to diffuse across the network. At each step, if labeled neighbors exist, they are selected with higher probability; otherwise, the walk proceeds uniformly. Repeated walks per node accumulate labeled visits, defining a propagation distribution. We will denote each labeled random walk by $\mathcal{W}$.

To refine this signal, we adopt an attention mechanism (Veličković et al., 2018; Wang et al., 2020), which weights the contribution of labeled nodes by similarity. For an unlabeled node $i$ with embedding $\mathbf{S}_{i,:}$, the attention weight for node $j$ is

$$w_{ij} = \frac{\exp(\mathbf{S}_{i,:}^{\mathsf{T}} \mathbf{S}_{j,:})}{\sum_{k \in \rho(i)} \exp(\mathbf{S}_{i,:}^{\mathsf{T}} \mathbf{S}_{k,:})}, \tag{7}$$

where $\rho(i)$ denotes the set of nodes visited in random walks from $i$. The corresponding semi-supervised gradient is

$$\nabla_{\mathbf{S}} Q_{\text{semi}} = \mathbf{S}_{i,:} - \sum_j w_{ij} \mathbf{S}_{j,:}. \tag{8}$$

This encourages unlabeled embeddings to shift toward weighted averages of similar labeled neighbors.

### 3.4 OPTIMIZATION

We integrate modularity, supervised, and semi-supervised objectives into a unified gradient ascent update:

$$\nabla_{\mathbf{S}} Q_{\text{total}} = \nabla_{\mathbf{S}} Q_{\text{prop}} - \lambda_{\text{sup}} \nabla_{\mathbf{S}} Q_{\text{sup}} - \lambda_{\text{semi}} \nabla_{\mathbf{S}} Q_{\text{semi}}. \tag{9}$$

Embeddings are updated as

$$\mathbf{S} \leftarrow \mathbf{S} + \eta \nabla_{\mathbf{S}} Q_{\text{total}}, \tag{10}$$

where $\eta$ is the learning rate. To ensure stability, $\mathbf{S}$ is orthonormalized after each iteration via QR decomposition. The overall procedure is represented in Algorithm 1. Further implementation details can be found in Appendices A and C.

## 4 EXPERIMENTS

### 4.1 DATASETS

The evaluation of the proposed semi-supervised modularity-based node embedding method is conducted on six benchmark datasets: Cora (McCallum et al., 2000), CiteSeer (Giles et al., 1998), WikiCS (Mernyei and Cangea, 2020), Amazon Photo or Photo (McAuley et al., 2015), PubMed (Namata et al., 2012), and ArXiV (Hu et al., 2020). Each dataset consists of nodes representing entities and edges signifying relationships (Table 4). For experiments, whenever necessary, we mask labels of subsets of nodes (which are used for testing node classification). The experiments assume that node features are unavailable, except for the case of a trivial baseline described in Section 4.2.

---

**Algorithm 1** FUSE

---

**Input:** Graph $\mathcal{G}(V, E)$, Labels $\mathbf{y}$, Label Mask $\mathbf{M}$, Learning Rate $\eta$, Regularization $\lambda_{\text{sup}}, \lambda_{\text{semi}}$, Iterations $T$

**Output:** Optimized Embeddings $\mathbf{S}$

1: Convert $\mathcal{G}$ to adjacency $\boldsymbol{A}$, compute degrees $\boldsymbol{d}$, total edges $m$
2: Initialize $\mathbf{S}$ randomly, orthonormalize using QR
3: $\mathcal{W} \leftarrow \text{LABELEDRANDOMWALKS}(\mathcal{G}, \mathbf{M}, \mathbf{y})$        $\triangleright$ From Algorithm **??**
4: $\mathbf{W} \leftarrow \text{COMPUTEATTENTIONWEIGHTS}(\mathbf{S}, \mathcal{W})$        $\triangleright$ From Algorithm 3
5: **for** $t = 1$ to $T$ **do**
6:      $\nabla_{\mathbf{S}} Q_{\text{prop}} \leftarrow \frac{1}{2m} \left( \boldsymbol{A}\mathbf{S} - \frac{1}{2m} \boldsymbol{d}(\mathbf{1}^{\intercal}\mathbf{S}) \right)$        $\triangleright$ Modularity gradient
7:      $\nabla_{\mathbf{S}} Q_{\text{sup}} \leftarrow \mathbf{S} - \tilde{\mathbf{S}}$        $\triangleright$ Supervised gradient
8:      $\nabla_{\mathbf{S}} Q_{\text{semi}} \leftarrow \mathbf{S}_{i,:} - \sum_j w_{ij} \mathbf{S}_{j,:}$        $\triangleright$ Semi-supervised gradient
9:      $\mathbf{S} \leftarrow \mathbf{S} + \eta \left( \nabla_{\mathbf{S}} Q_{\text{prop}} - \lambda_{\text{sup}} \nabla_{\mathbf{S}} Q_{\text{sup}} - \lambda_{\text{semi}} \nabla_{\mathbf{S}} Q_{\text{semi}} \right)$
10:      Orthonormalize $\mathbf{S}$ using QR-decomposition
11: **end for**
12: **return** $\mathbf{S}$

---

## 4.2 BASELINES

We evaluate our approach against a range of baselines spanning unsupervised, self-supervised, semi-supervised and trivial embedding strategies:

- **Unsupervised baselines.** We use Node2Vec (Grover and Leskovec, 2016) and Deep-Walk (Perozzi et al., 2014), both random-walk-based methods that employ the Skip-Gram model for representation learning. In addition, we include Variational Graph Auto-Encoders (VGAE) (Kipf and Welling, 2016) as a neural network based unsupervised embedding method. For VGAE, we initialized the feature matrix as an identity matrix since we assumed that features were unavailable, as recommended by Kipf and Welling (2016). We also implemented M-NMF (Wang et al., 2017) for generating the $k$ dimensional embeddings, observing the downstream classification results later.

- **Self-supervised baseline.** We employ Deep Graph Infomax (DGI) (Veličković et al., 2019), which maximizes mutual information between node-level and graph-level representations. Here we initialized the feature matrix as a random $n \times k$ matrix ($n =$ number of nodes and $k = 150$) to compare with FUSE, since we assume that features were unavailable.

- **Semi-supervised baseline.** We employ GraFN (Lee et al., 2022) and ReVAR (Yan et al., 2023) under the *non-availability of features* setting, using random feature matrices. Both frameworks combine a GNN encoder with a classifier via customized losses, making embedding generation and classification degenerate or inseparable. Hence, we tested them with different encoders (GCN, GAT, GraphSAGE). As ReVAR targets imbalanced node classification, we adapted it to the non-imbalanced case to generate embeddings through the encoders and evaluate classifier performance. Reported runtime is the sum of embedding generation and classification, as both are degenerate.

- **Trivial baselines.** Random embeddings serve as a lower-bound baseline, while directly using the available node features act as an upper-bound benchmark.

Embeddings generated by each method are subsequently used as input to three GNN classifiers: GCN (Kipf and Welling, 2017), GAT (Veličković et al., 2018), and GraphSAGE (Hamilton et al., 2017). For all baselines, unless otherwise mentioned, we have assumed the default parameter values for all experiments. To ensure comparability, we fix the embedding dimension to 150 and maintain identical neural architectures across datasets: a two-layer vanilla GNN or MLP with no additional hyperparameter tuning. For our method, the initialization of the embedding matrix $\mathbf{S}$ is random, and dataset-specific parameter values of FUSE are summarized in Table 3 in Appendix A. All experiments, where runtime for embedding generation is reported, were conducted on a workstation equipped with an 13th Gen Intel(R) Core(TM) i9-13900 CPU, 64 GB of RAM; no GPU acceleration was used.

### 4.3 RESULTS

We now present the empirical evaluation of our proposed method across six benchmark datasets. Results are structured around five key aspects: (1) downstream classification performance and runtime efficiency, (2) ablation studies analyzing the contributions of unsupervised, semi-supervised components of the FUSE objective, (3) FUSE parameter sensitivity analysis, (4) scalability outcomes and (5) missingness experiments across different masking mechanisms.

#### 4.3.1 DOWNSTREAM CLASSIFICATION PERFORMANCE

Table 1 summarizes the classification accuracy and F1-scores obtained when embeddings from different methods are fed into GCN, GAT, and GraphSAGE under both 70-30 and 30-70 train-test splits. Several consistent trends emerge:

- **FUSE achieves competitive classification accuracy.** On both splits, FUSE performs on par with DeepWalk, Node2Vec and clearly outperforms self supervised algorithms like DGI along with unsupervised M-NMF and semi-supervised GraFN and ReVAR in nearly all cases. Similar to Node2Vec and DeepWalk it is robust across classifiers and also matches or even surpasses the classification performance of the given embedding.

- **FUSE facilitates superior learning for GCNs.** FUSE-generated embeddings especially enhance the learning capability of the GCN classifier. This is an important aspect in the context of speed and scalability since GCN is significantly faster than GAT or GraphSAGE.

Overall, these results confirm that generating embeddings via FUSE leads to strong downstream classification without requiring precomputed features.

#### 4.3.2 DOWNSTREAM NODE CLUSTERING PERFORMANCE

We conducted node clustering experiments to evaluate the performance of FUSE compared to existing baselines. We measured one intrinsic metric, the DB Index, as well as two extrinsic metrics, ARI and the V-Measure score. Dataset-wise results are presented in Tables 26- 37. We plotted these results for the embeddings learned through GAT using FUSE initialization in Figures 5 and 6. We observed that FUSE achieves the minimum DB index in most of the cases, indicating superior cluster separation in the learned embeddings for most datasets. We also observed that the embeddings for FUSE have the highest V-Measure score for all the datasets, which indicates that FUSE-initialized classifiers can learn embeddings where the clusters are consistent with class labels.isting baselines. We measured one intrinsic metric, the DB Index, as well as two extrinsic metrics,

#### 4.3.3 RUNTIME EFFICIENCY

Tables 2 and 5 report embedding generation times across datasets. Although DeepWalk and Node2Vec achieve downstream classification performance at par with FUSE, our algorithm exhibits a significant computational advantage, being approximately 5 times faster on average. This advantage is further supported by scalability studies on the ArXiV dataset (Appendix C.4, Tables 19 and 20), where FUSE is more than 7 times faster.

To address the potential concern that the default `walk_length` of 80 for Node2Vec and DeepWalk might inflate their runtimes, we conducted an additional experiment with a reduced `walk_length` of 5 for a single seed for these two algorithms only. Interestingly, across datasets, we observed that, performance remained comparable to that with the longer walk, and runtimes did improve significantly. Nonetheless, for larger datasets, especially with more edges, like Photos, WikiCS, and ArXiV (see Appendix C.4, Tables 16, 17 and 18), FUSE maintains its advantage, delivering superior classification performance while remaining around 3 times faster.

In fact, FUSE is faster than all compared unsupervised and self-supervised embedding algorithms except DGI, which performs poorly in downstream classification and node clustering under the assumption of feature unavailability. Semi-supervised algorithms like GraFN and ReVAR, while computationally feasible, display significantly lower performance than Node2Vec, DeepWalk, and FUSE (Table 1).

Execution times for our ablation variants are compared in Table 7. The semi-supervised modularity-based embeddings are only marginally slower than the purely unsupervised versions but are significantly more effective (see Table 6), confirming that label propagation is an efficient and beneficial addition.

| Classifier | Embedding | 70-30 Split | | 30-70 Split | |
|---|---|---|---|---|---|
| | | **Accuracy** | **F1** | **Accuracy** | **F1** |
| **GAT** | Random | $0.71 \pm 0.014$ | $0.68 \pm 0.016$ | $0.48 \pm 0.028$ | $0.40 \pm 0.033$ |
| | DeepWalk | $\underline{0.82 \pm 0.008}$ | $\underline{0.80 \pm 0.009}$ | $\mathbf{0.79 \pm 0.007}$ | $\mathbf{0.77 \pm 0.009}$ |
| | Node2Vec | $\underline{0.82 \pm 0.007}$ | $\underline{0.80 \pm 0.007}$ | $\mathbf{0.79 \pm 0.007}$ | $\mathbf{0.77 \pm 0.008}$ |
| | MNMF | $0.55 \pm 0.024$ | $0.52 \pm 0.026$ | $0.34 \pm 0.024$ | $0.29 \pm 0.022$ |
| | VGAE | $0.81 \pm 0.009$ | $0.79 \pm 0.010$ | $0.78 \pm 0.005$ | $0.76 \pm 0.005$ |
| | DGI | $0.59 \pm 0.073$ | $0.51 \pm 0.098$ | $0.54 \pm 0.070$ | $0.45 \pm 0.100$ |
| | GraFN | $0.76 \pm 0.012$ | $0.71 \pm 0.052$ | $0.70 \pm 0.011$ | $0.60 \pm 0.075$ |
| | ReVAR | $0.43 \pm 0.023$ | $0.29 \pm 0.029$ | $0.42 \pm 0.017$ | $0.29 \pm 0.029$ |
| | FUSE | $\mathbf{0.82 \pm 0.009}$ | $\mathbf{0.80 \pm 0.009}$ | $\underline{0.78 \pm 0.006}$ | $\underline{0.76 \pm 0.008}$ |
| | Given Emb. | $0.86 \pm 0.005$ | $0.84 \pm 0.006$ | $0.84 \pm 0.004$ | $0.82 \pm 0.006$ |
| **GCN** | Random | $0.49 \pm 0.031$ | $0.45 \pm 0.030$ | $0.37 \pm 0.032$ | $0.33 \pm 0.028$ |
| | DeepWalk | $\underline{0.64 \pm 0.039}$ | $\underline{0.58 \pm 0.050}$ | $\underline{0.67 \pm 0.027}$ | $\underline{0.61 \pm 0.039}$ |
| | Node2Vec | $0.64 \pm 0.042$ | $0.57 \pm 0.058$ | $0.66 \pm 0.026$ | $0.61 \pm 0.036$ |
| | MNMF | $0.46 \pm 0.044$ | $0.37 \pm 0.051$ | $0.36 \pm 0.032$ | $0.29 \pm 0.026$ |
| | VGAE | $\underline{0.71 \pm 0.017}$ | $\underline{0.68 \pm 0.022}$ | $\underline{0.69 \pm 0.017}$ | $\underline{0.66 \pm 0.017}$ |
| | DGI | $0.30 \pm 0.026$ | $0.12 \pm 0.049$ | $0.32 \pm 0.048$ | $0.15 \pm 0.081$ |
| | GraFN | $0.74 \pm 0.010$ | $0.72 \pm 0.009$ | $0.66 \pm 0.006$ | $0.64 \pm 0.007$ |
| | ReVAR | $0.35 \pm 0.019$ | $0.18 \pm 0.028$ | $0.35 \pm 0.017$ | $0.18 \pm 0.028$ |
| | FUSE | $\mathbf{0.78 \pm 0.014}$ | $\mathbf{0.76 \pm 0.013}$ | $\mathbf{0.73 \pm 0.020}$ | $\mathbf{0.71 \pm 0.017}$ |
| | Given Emb. | $0.58 \pm 0.022$ | $0.49 \pm 0.018$ | $0.56 \pm 0.023$ | $0.47 \pm 0.018$ |
| **SAGE** | Random | $0.56 \pm 0.018$ | $0.51 \pm 0.015$ | $0.35 \pm 0.018$ | $0.26 \pm 0.014$ |
| | DeepWalk | $\mathbf{0.81 \pm 0.011}$ | $\mathbf{0.79 \pm 0.012}$ | $\mathbf{0.78 \pm 0.008}$ | $\mathbf{0.76 \pm 0.009}$ |
| | Node2Vec | $\mathbf{0.81 \pm 0.010}$ | $\mathbf{0.79 \pm 0.009}$ | $\underline{0.77 \pm 0.007}$ | $\underline{0.75 \pm 0.008}$ |
| | MNMF | $0.52 \pm 0.016$ | $0.47 \pm 0.021$ | $0.33 \pm 0.019$ | $0.27 \pm 0.022$ |
| | VGAE | $\underline{0.80 \pm 0.009}$ | $\underline{0.78 \pm 0.011}$ | $0.76 \pm 0.010$ | $0.74 \pm 0.011$ |
| | DGI | $0.57 \pm 0.054$ | $0.48 \pm 0.088$ | $0.54 \pm 0.047$ | $0.46 \pm 0.070$ |
| | GraFN | $0.67 \pm 0.010$ | $0.63 \pm 0.010$ | $0.55 \pm 0.008$ | $0.51 \pm 0.010$ |
| | ReVAR | $0.25 \pm 0.009$ | $0.15 \pm 0.006$ | $0.24 \pm 0.005$ | $0.16 \pm 0.006$ |
| | FUSE | $\underline{0.80 \pm 0.012}$ | $\underline{0.77 \pm 0.013}$ | $0.75 \pm 0.008$ | $0.73 \pm 0.010$ |
| | Given Emb. | $0.85 \pm 0.008$ | $0.83 \pm 0.012$ | $0.83 \pm 0.006$ | $0.80 \pm 0.008$ |

Table 1: Classification accuracy and F1-score (mean $\pm$ standard deviation) across embedding methods and three classifiers for all the datasets (except ArXiV). Results are reported for both 70-30 and 30-70 train-test splits. Best and second-best (excluding given embeddings) are highlighted in **bold** and underlined, respectively.

### 4.3.4 ADDITIONAL ANALYSES

To substantiate the effectiveness and robustness of FUSE, we conducted ablation, sensitivity, scalability, and masking studies (details in Appendix C).

**Ablation Study.** We evaluated the individual contributions of the semi-supervised and unsupervised objectives (Appendix C.2), as well as their combination, under both the 30-70 and 70-30 train-test splits (assumed learning rate 0.05). The unsupervised component of the FUSE objective alone performs significantly well compared to the only semi-supervised counterpart, especially for the GraphSAGE classifier (Tables 6 and 8). This indicates that FUSE can also adapt well to scenarios where labels are completely unavailable, relying solely on the modularity-driven objective. The semi-supervised component alone is also at par with the unsupervised component in terms of classification performance. However, the unsupervised objective alone proves to be faster (Tables 7

| Embedding | Cora | CiteSeer | Amazon Photo | WikiCS | PubMed | Average |
|---|---|---|---|---|---|---|
| **70-30 Split** | | | | | | |
| Random | 0.01 | 0.01 | 0.01 | 0.03 | 0.04 | 0.02 |
| DeepWalk | 50.48 | 51.41 | 292.30 | 747.20 | 490.72 | 326.422 |
| Node2Vec | 47.26 | 50.32 | 288.33 | 745.33 | 453.74 | 316.996 |
| MNMF | 41.75 | 56.34 | 323.31 | 672.46 | 1742.94 | 567.36 |
| VGAE | 12.95 | 14.32 | 137.28 | 329.46 | 235.24 | 145.850 |
| DGI | **6.78** | **7.96** | 53.42 | 134.58 | **39.43** | **48.434** |
| FUSE | 12.52 | 13.36 | **49.47** | **86.45** | 95.79 | 51.518 |
| **30-70 Split** | | | | | | |
| Random | 0.01 | 0.01 | 0.01 | 0.03 | 0.04 | 0.02 |
| DeepWalk | 50.99 | 51.84 | 292.98 | 792.11 | 477.77 | 333.138 |
| Node2Vec | 47.49 | 50.65 | 290.95 | 785.70 | 448.58 | 324.674 |
| MNMF | 41.75 | 56.34 | 323.31 | 672.46 | 1742.94 | 567.36 |
| VGAE | 12.97 | 14.48 | 136.07 | 338.10 | 226.29 | 145.582 |
| DGI | **6.83** | **7.33** | **53.37** | **126.80** | **36.05** | **46.076** |
| FUSE | 14.42 | 14.31 | 64.55 | 128.92 | 109.15 | 66.27 |

Table 2: Runtime comparison (in seconds) of different embedding methods across datasets (except ArXiV) under 70-30 and 30-70 train-test splits. Reported values are averages over 5 runs. Best and second-best (excluding random embeddings) are highlighted in **bold** and underlined, respectively.

and 9). It is clear from the overall results, however, that incorporating all three components of the objective is indeed advantageous, especially for large-scale datasets.

**Sensitivity Analysis.** We analyzed robustness to hyperparameters (Tables 13 (a, b, c)). Learning rate $\eta$ and loss weights $\lambda_{\text{sup}}$, $\lambda_{\text{semi}}$ were most sensitive, while structural parameters ($r$, $L$, $L'$) tolerated wider ranges. Deeper settings sometimes improved accuracy but increased runtime disproportionately, suggesting moderate configurations as optimal (Appendix C.3).

**Scalability Experiments.** We additionally evaluated FUSE on a large-scale graph ArXiV to assess its applicability to real-world settings. The results and execution times are reported in Tables 19 and 20 (Appendix C.4). To further examine scalability under more challenging conditions, we conducted extended experiments on two substantially larger datasets: MAG ($\sim$736K nodes, $\sim$8M edges) and ogbn products ($\sim$2.45M nodes, $\sim$61.9M edges) using a 30-70 split. As detailed in Appendix C.4 (Tables 14 and 15), the unsupervised variant of FUSE remained highly efficient, completing in 25 minutes on MAG and approximately 2.5 hours on the ogbn products graph, while producing a substantially better F1-Score on ogbn-products, compared to the given embedding baseline. In contrast, DeepWalk, which is among one of the best performing benchmarks in terms of Accuracy and F1-Score, even with reduced walk parameters (walk length 5, 10 walks), failed to complete within 24 hours on ogbn products using a single CPU worker. While FUSE trades off accuracy and F1-score on these very large graphs, its substantial speed advantage and its compatibility with the faster GCN classifier, highlights its suitability for feature-agnostic settings where fast embedding generation is critical.

**Label masking Experiments.** In real-world datasets, class distributions among unlabeled nodes are often highly imbalanced. To assess the robustness of FUSE under such imbalance, we evaluated its performance under three label-masking strategies at 20%, 50%, and 80% missingness on the Cora and CiteSeer datasets (details in Appendix C.5). FUSE remained consistently competitive across all settings, showing a particular advantage with the GCN and GAT classifiers under high missingness rates (80%) and more challenging masking schemes (MAR, MNAR).

## 5 DISCUSSION AND CONCLUSION

In this paper, we introduce FUSE, a fast, scalable and high-performance node embedding generation algorithm that does not require predefined features. The objective function of FUSE integrates an unsupervised, a semi-supervised and a supervised component.

The unsupervised component of the FUSE objective is based on a novel linear-time maximization of graph modularity, which enables runtime and performance-efficient embedding generation even in the absence of labels. Modularity, being a global graph property, can be interpreted as learning global structural features. The semi-supervised component, on the other hand, leverages label-biased random walks and inter-node attention between labeled and unlabeled nodes. This component allows the model to capture local structures at the node or neighborhood level during feature learning. Supported by the global structure learning of the unsupervised module, we observe that FUSE can extract meaningful local features using short random walks of length as little as five. Jointly optimizing these two objectives also contributes to the overall runtime efficiency of FUSE. Finally, the supervised component reduces intra-class embedding variance, ensuring that nodes belonging to the same class are closely aligned in the embedding space. By combining these elements, FUSE achieves accuracy comparable to or better than established baselines, while being five to seven times faster, particularly on large-scale datasets such as ArXiv.

Nevertheless, FUSE has some limitations that we would like to highlight. FUSE is designed to operate in settings where node features are assumed to be unavailable. It is thus unable to incorporate information extraneous to the graph structure. A simple extension of the algorithm to incorporate node features would be to concatenate these features onto the embedding matrix $\mathbf{S}$. Another direction for future work is to investigate how this framework can be adapted to dynamically evolving graphs while maintaining its scalability benefits.

FUSE is designed for settings where node features are assumed to be unavailable, and therefore it cannot leverage information external to the graph structure. This places FUSE in a specific niche: feature-agnostic scenarios where fast, structure-driven embedding generation is required and moderate reductions in accuracy is acceptable. Our large-scale scalability experiments support this characterization. FUSE offers substantial computational advantages on large graphs such as MAG and ogbn products. Notably, the unsupervised variant completes in minutes to a few hours while relying solely on conventional CPU execution, with no GPU acceleration, multicore parallel processing, or specialized high-performance libraries. FUSE is most suitable for applications in which labels are available but features are absent or unreliable, and where scalability requirements outweigh the need for the highest predictive performance.

## REPRODUCIBILITY STATEMENT

All code used to perform the experiments and generate the results presented in this work is included in the supplementary material as a zip archive. The benchmarking experimental results presented in Section 4.3 can be obtained from the files `benchmarking_utils.py`, `benchmarking_runner.ipynb` and `aggregation.ipynb` files from the folder titled 'FUSE_Unsupervised_Self-supervised_Benchmarks'. MNMF results can be obtained across the datasets from `MNMF.ipynb` inside the 'MNMF_Benchmark' folder. For the other two semi-supervised benchmarks, namely GraFN and ReVAR, the results can be obtained from the files `GraFN.ipynb`, `load_datasets_revar.ipynb` and `ReVar.ipynb` inside the folders 'GraFN_Benchmark' and 'ReVAR_Benchmark' respectively. Ablation results in Appendix C.2 can be found from the notebooks inside the folder 'Ablation_study'. The sensitivity analysis in Appendix C.3 (for each of the five datasets except ArXiV) and scalability results in Appendix C.4 are verifiable from the codes inside 'Sensitivity_Analysis' and 'Scalability_Experiments'. The experiments in Appendix C.5 can be run with the file `benchmark.py` in the folder 'Experiments_with_masking'.

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

# A DATASETS, ALGORITHM DETAILS AND VISUALIZATIONS

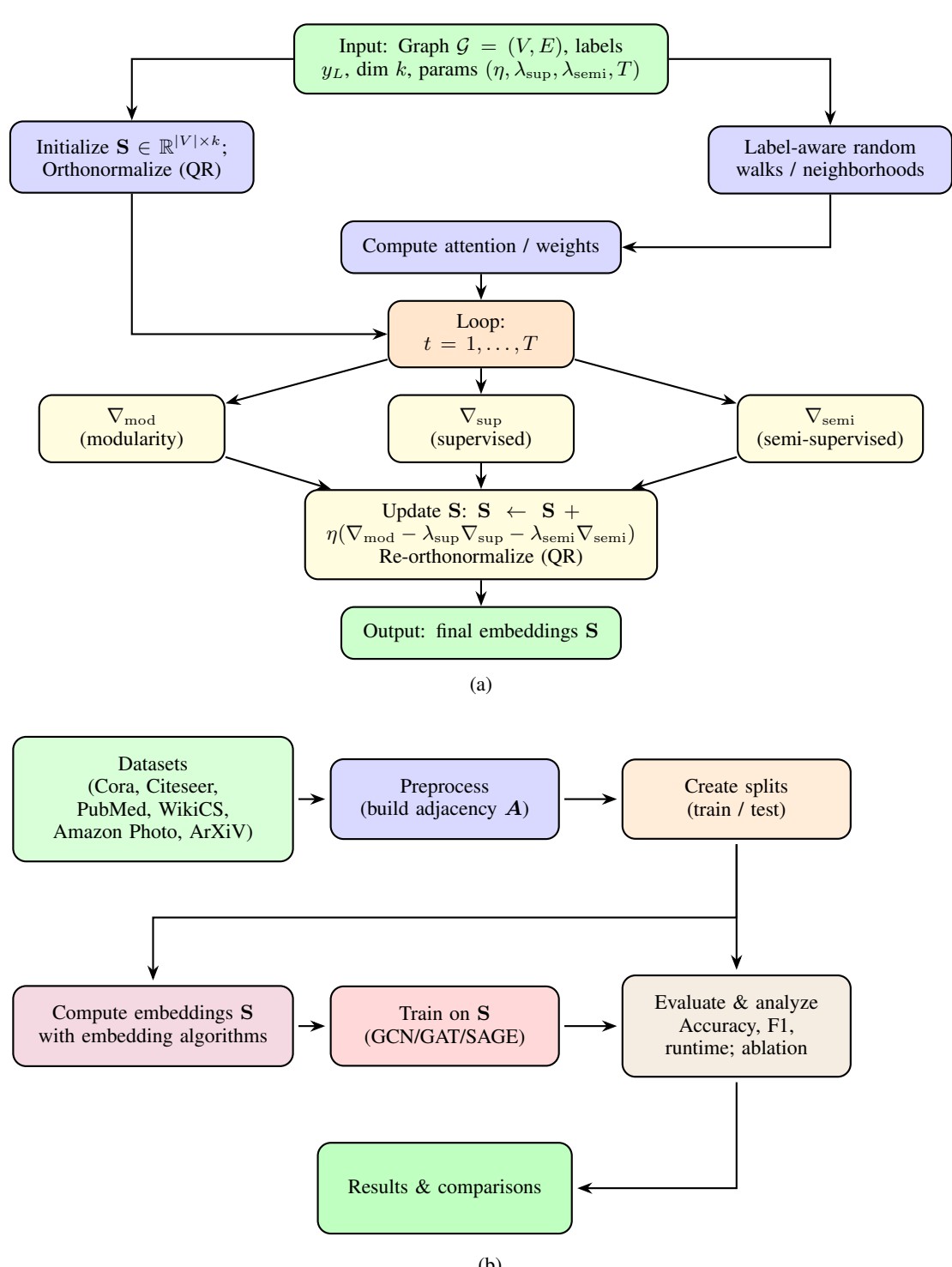

Figure 1: Overview of the method and experiments. (a) Algorithm pipeline (FUSE) and (b) Experimental workflow.

**Algorithm 2** Labeled Random Walks

**Input:** Graph $\mathcal{G}(V, E)$, Label Mask $\mathbf{M}$, Labels $\mathbf{y}$, Walks per node $r$, Walk length $L$, Max labeled steps $L'$
**Output:** Set of labeled walks $\mathcal{W}$
1: Initialize $\mathcal{W} \leftarrow \emptyset$
2: **for** each node $i \in V$ **do**
3:     **for** $w = 1$ to $r$ **do**
4:         Initialize walk $P \leftarrow [i]$, labeled_count $\leftarrow 0$
5:         **for** $t = 1$ to $L - 1$ **do**
6:             Let $\mathcal{N}(v_t)$ be the neighbors of current node $v_t$
7:             **if** $\mathcal{N}(v_t)$ is empty **then**
8:                 **break**
9:             **end if**
10:            $\mathcal{N}_L(v_t) \leftarrow \{u \in \mathcal{N}(v_t) \mid \mathbf{M}[u] = 1\}$
11:            **if** $|\mathcal{N}_L(v_t)| > 0$ **and** labeled_count $< L'$ **then**
12:                Choose next node $v_{t+1}$ uniformly from $\mathcal{N}_L(v_t)$         ▷ label-preferential step
13:                labeled_count $\leftarrow$ labeled_count $+1$
14:            **else**
15:                Choose next node $v_{t+1}$ uniformly from $\mathcal{N}(v_t)$         ▷ unbiased step
16:            **end if**
17:            Append $v_{t+1}$ to $P$
18:            **if** $\mathbf{M}[v_{t+1}] = 1$ **then**
19:                Add $v_{t+1}$ to $\mathcal{W}[i]$
20:            **end if**
21:         **end for**
22:     **end for**
23: **end for**
24: **return** $\mathcal{W}$

**Algorithm 3** Compute Attention Weights

**Input:** Embeddings $\mathbf{S}$, Labeled Walks $\mathcal{W}$
**Output:** Attention Weights $W$
1: **for** each unlabeled node $i \in V$ **do**
2:     **for** each labeled node $j \in \mathcal{W}[i]$ **do**
3:         Compute similarity: $s_{ij} = \mathbf{S}_{i,:}^{\mathsf{T}} \mathbf{S}_{j,:}$
4:         Compute attention: $w_{ij} = \dfrac{\exp(s_{ij})}{\sum_{k \in \mathcal{W}[i]} \exp(s_{ik})}$
5:     **end for**
6: **end for**
7: **return** $W$

| Parameter | Value | Description |
|---|---|---|
| $k$ | 150 | Learnt node embedding dimension (in case node embeddings are not given) |
| $\eta$ | 0.05 | Learning rate |
| $\lambda_{\text{supervised}}$ | 1.0 | Supervised loss weight |
| $\lambda_{\text{semi-supervised}}$ | 2.0 | Semi-supervised loss weight |
| $T$ | 200 | Number of gradient ascent iterations |
| $r$ | 10 | Number of random walks per node |
| $L$ | 5 | Length of each random walk |
| $L'$ | 3 | Maximum labeled steps in a walk |

Table 3: Hyperparameters used in semi-supervised modularity optimization for all datasets.

| Dataset | # Nodes | # Edges | # Classes | Given Embedding Dim. |
|---------|---------|---------|-----------|----------------------|
| Cora | 2,708 | 5,429 | 7 | 1,433 |
| CiteSeer | 3,327 | 9,104 | 6 | 3,703 |
| PubMed | 19,717 | 44,338 | 3 | 500 |
| Amazon Photo | 7,487 | 119,043 | 8 | 745 |
| WikiCS | 11,701 | 216,123 | 10 | 300 |
| ArXiV | 1,69,343 | 1,166,243 | 40 | 128 |

Table 4: Statistics of the benchmark datasets used in the experiments.

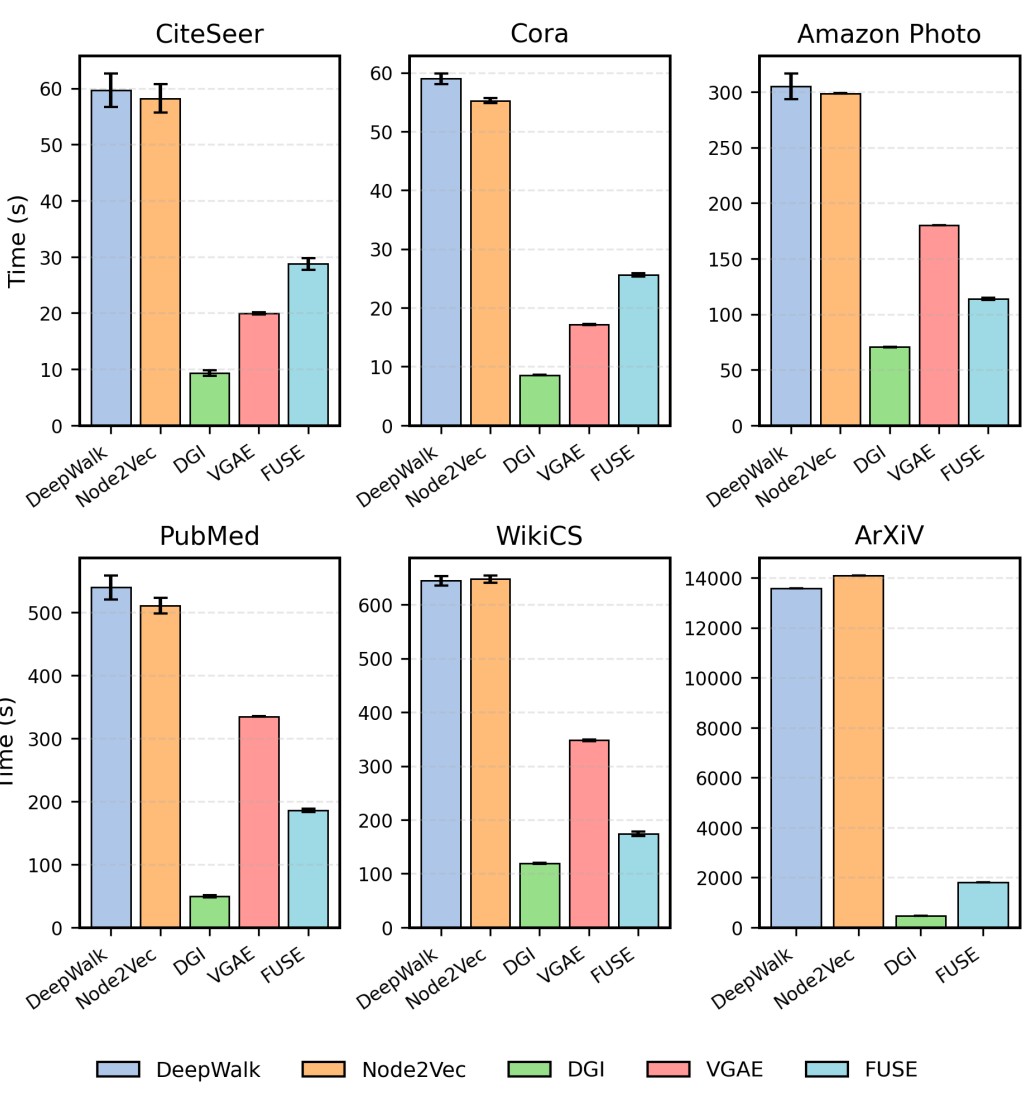

Figure 2: Runtimes averaged across seeds for several datasets. FUSE shows clear advantage in compared to Node2Vec and DeepWalk with default parameters. Even though DGI and VGAE are faster than FUSE for some datasets, FUSE outperforms them significantly in terms of Accuracy and F1-Score ass seen in Tables 1, 19 and 20.

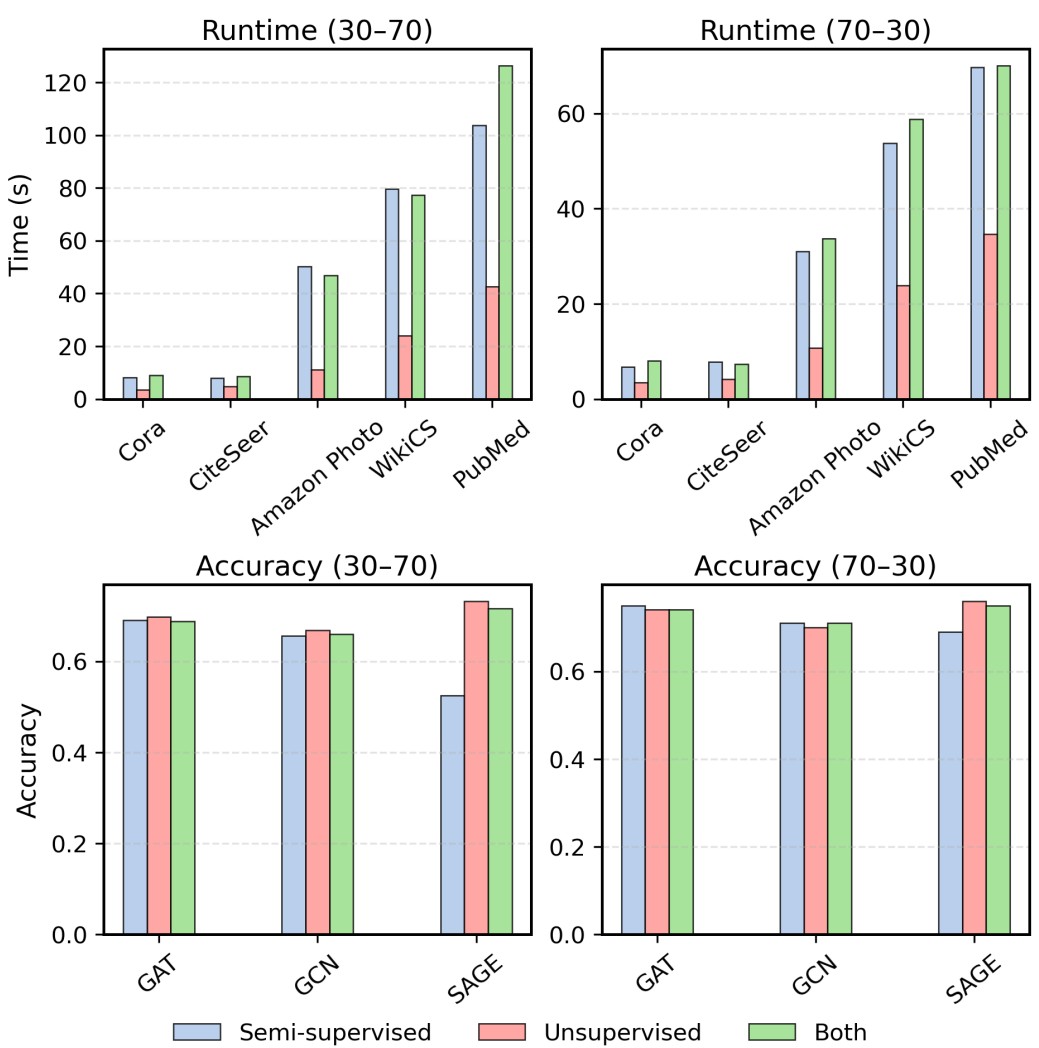

Figure 3: This Figure present accuracies and runtimes averaged across datasets for the three Ablation cases of FUSE algorithm as presented in Section C.2.

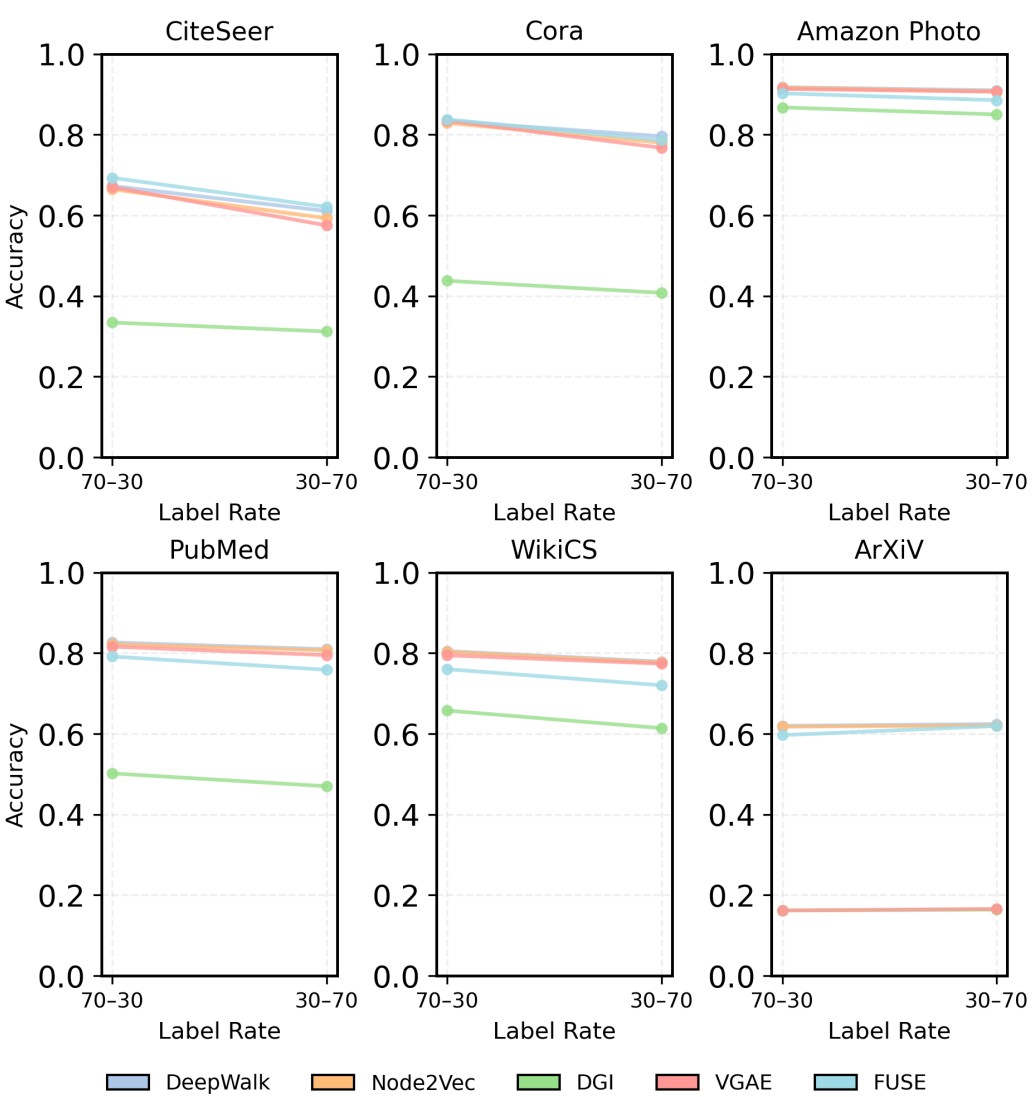

Figure 4: In this Figure we show Accuracy vs label rates for SAGE across several datasets. We do not observe significant changes in accuracy with change in label rate for any of the algorithms. There is a slight downward trend in most cases, with reduced proportion of labeled nodes (training data), as expected.

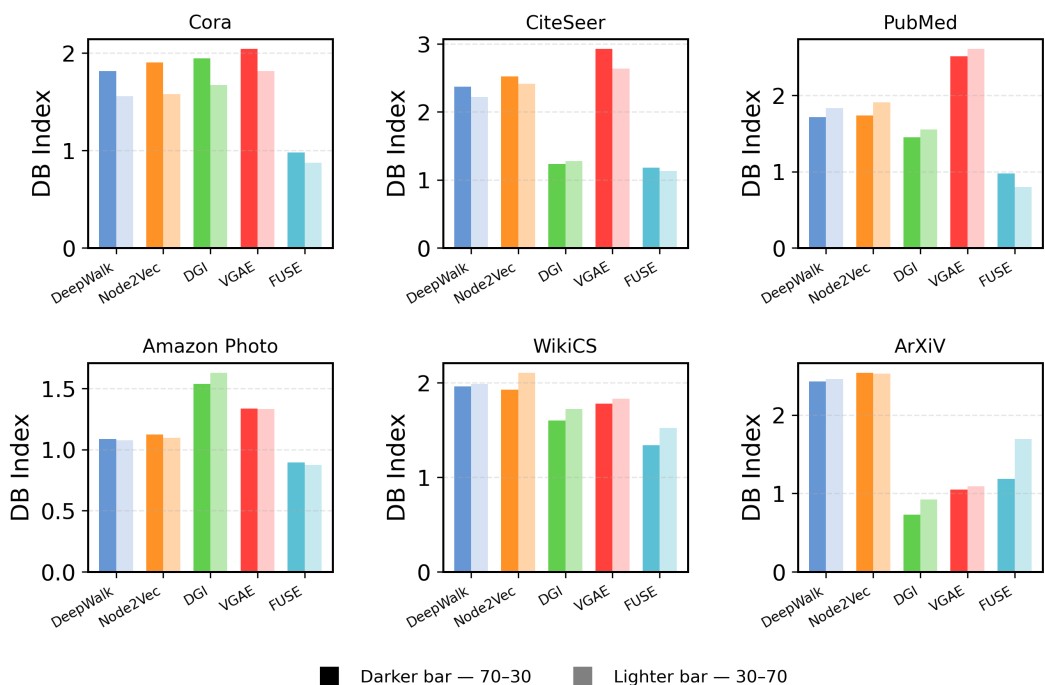

Figure 5: The Figure shows DB Index comparison for GAT-learnt embeddings across several initializations of the embedding generation benchmarks. We observe that the embeddings for FUSE has the least DB index, indicating superior cluster separation in the learned embeddings for most datasets

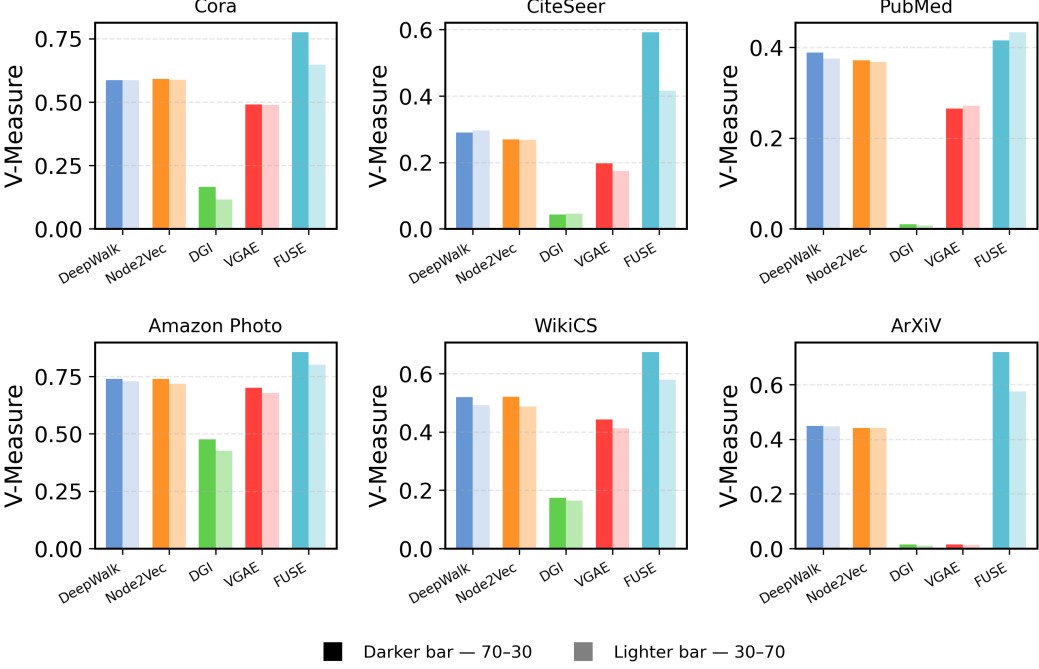

Figure 6: The Figure shows V-Measure comparison for GAT-learnt embeddings across several initializations of the embedding generation benchmarks. We observe that the embeddings for FUSE have the highest V-Measure for all of the datasets. This indicates that FUSE-initialized classifiers can learn embeddings where clusters are consistent with known class labels.

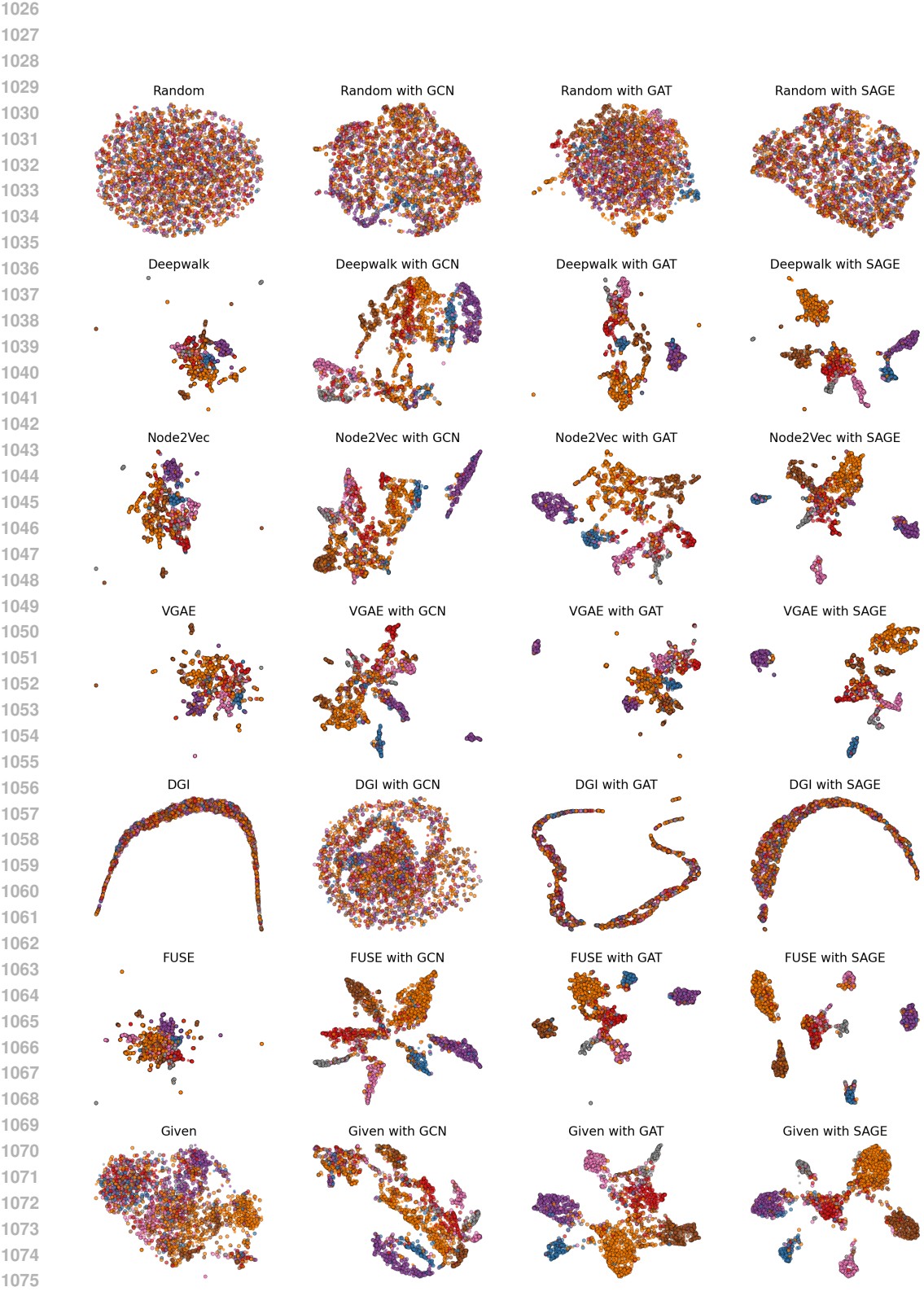

Figure 7: UMAP visualizations of Cora 70-30 embeddings

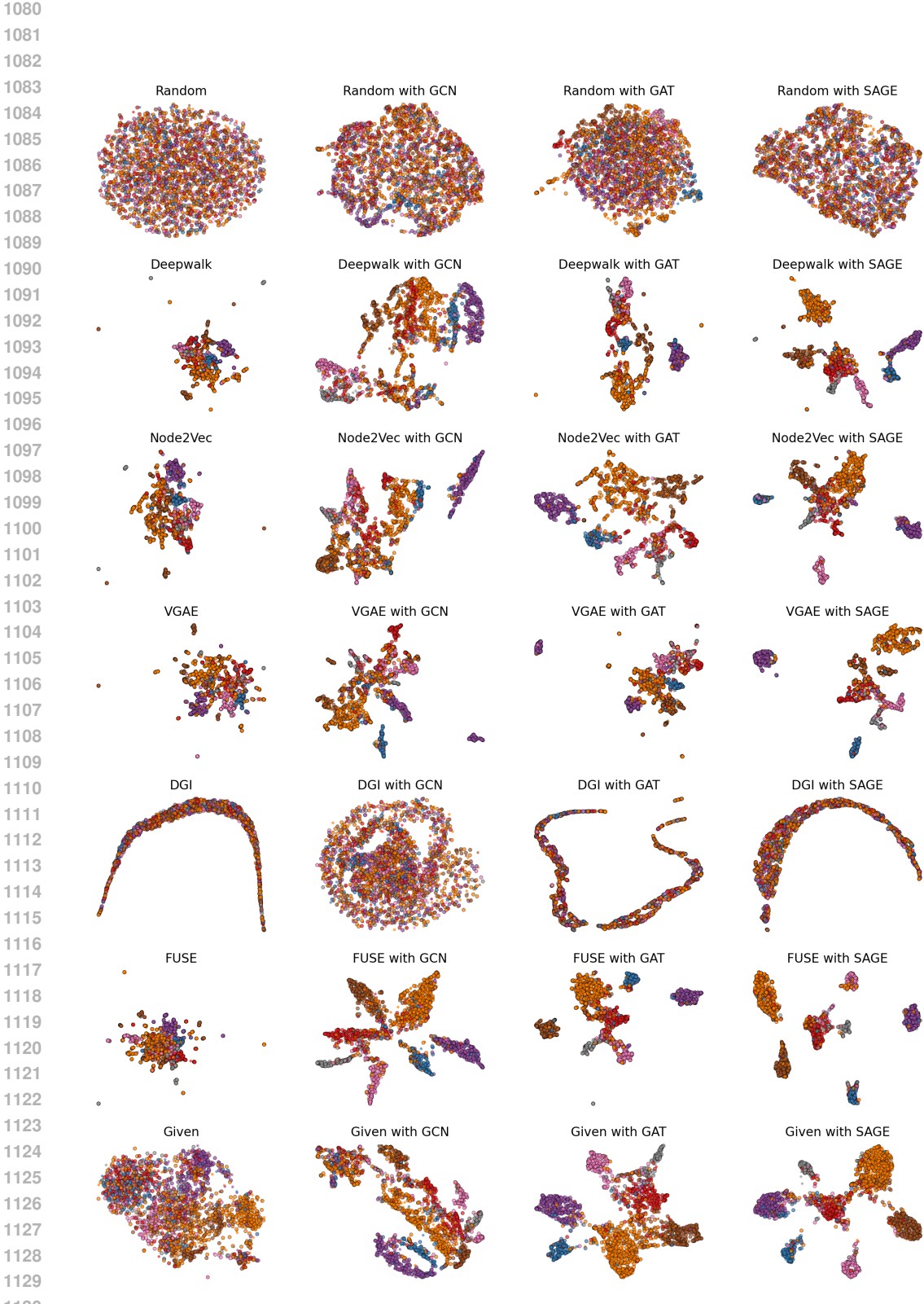

Figure 8: UMAP visualizations of Cora 30-70 embeddings

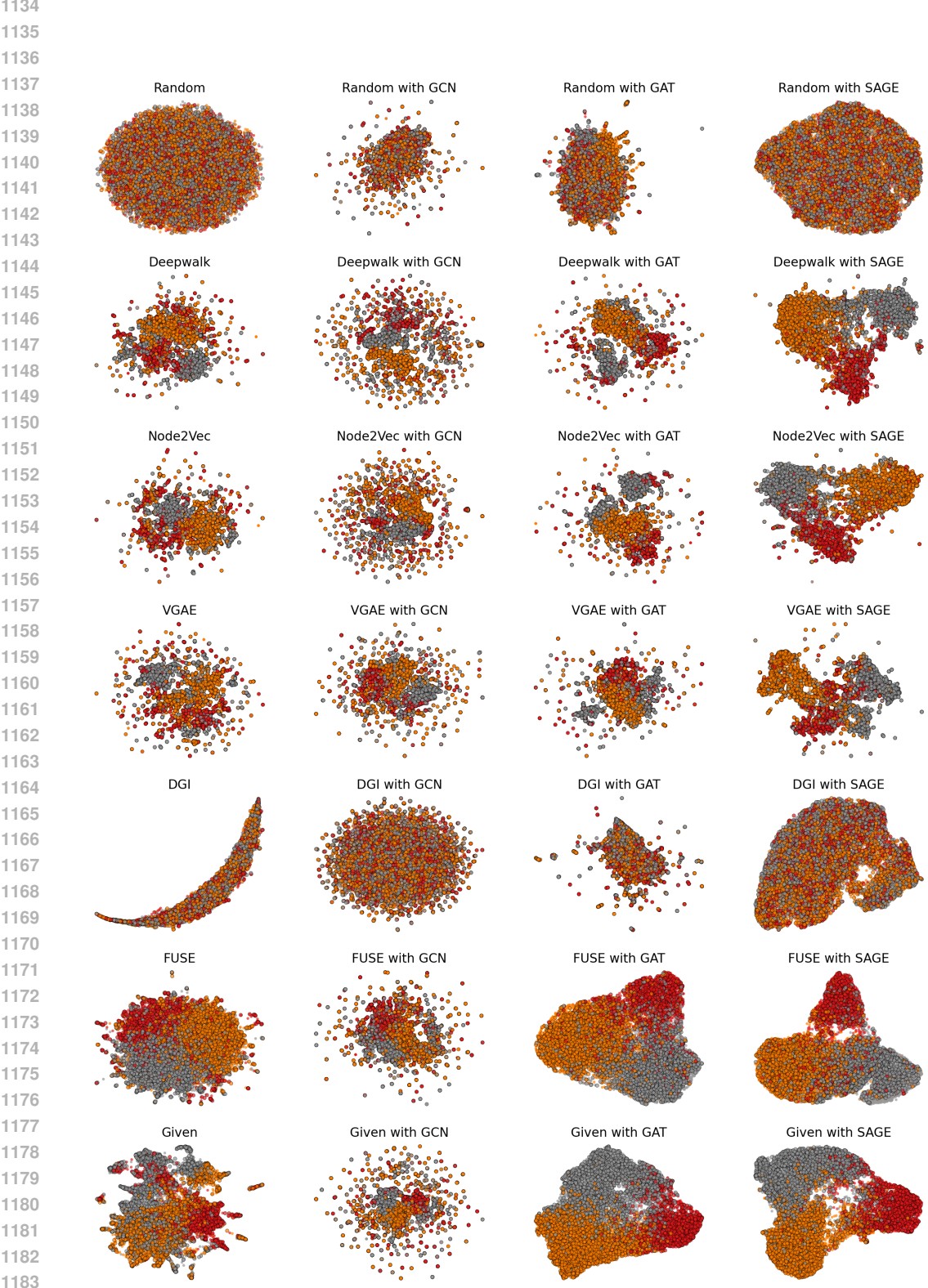

Figure 9: UMAP visualizations of PubMed 70-30 embeddings

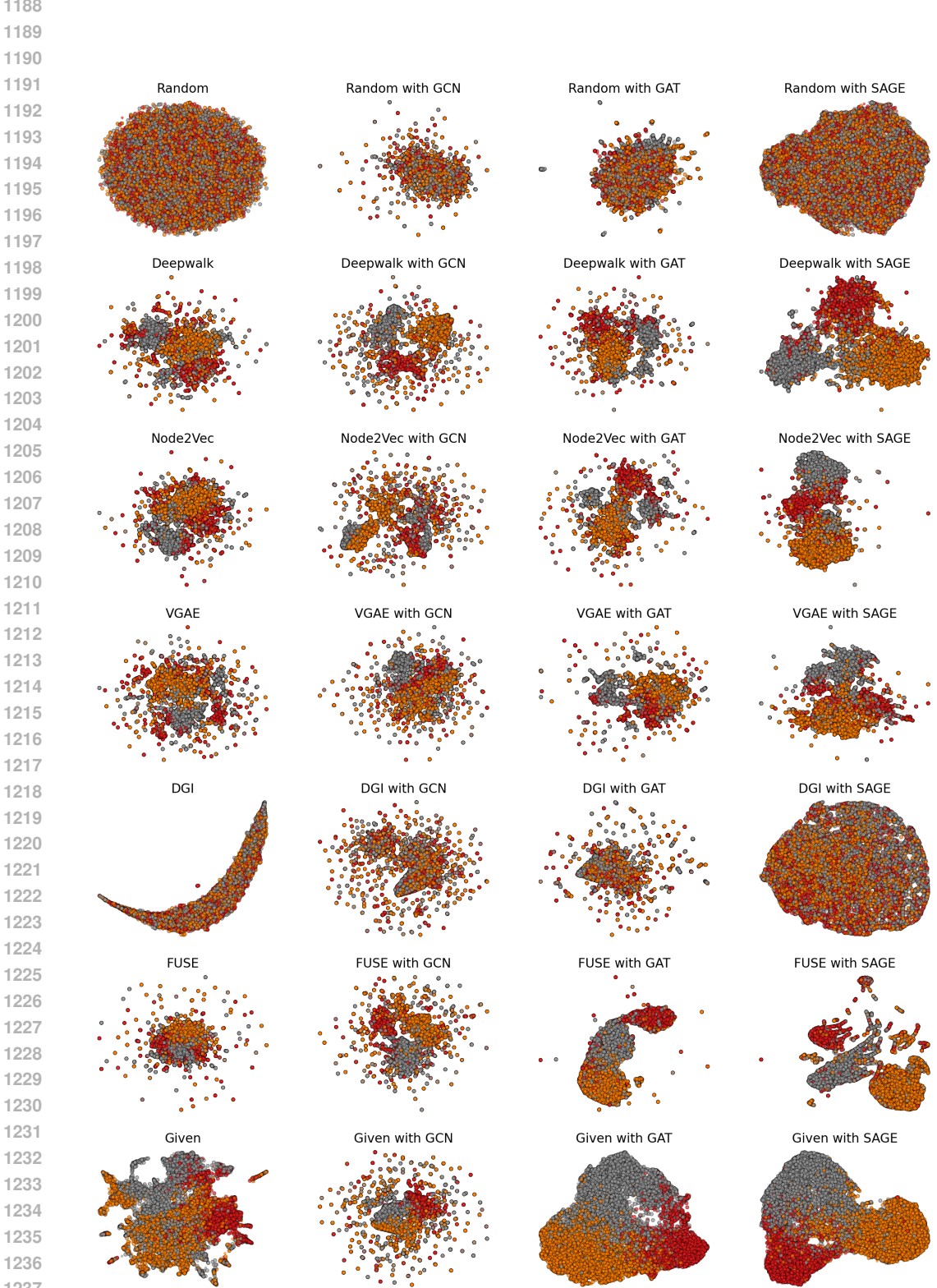

Figure 10: UMAP visualizations of PubMed 30-70 embeddings

## B   THEORETICAL RESULTS

We will show that the operator norm, and hence the Fröbenius norm of the surrogate gradient $\nabla_{\mathbf{S}} Q_{\text{prop}}$ in equation 4 is bounded above.

**Proposition 1.**

$$\sup_{||\boldsymbol{x}|| \leq 1} \frac{\left|\left|(\boldsymbol{A} - \frac{1}{2m}\boldsymbol{d}\mathbf{1}^{\intercal})\mathbf{S}\boldsymbol{x}\right|\right|}{||\boldsymbol{x}||} \leq M \tag{11}$$

*for some $M \in \mathbb{R}^+$*

*Proof.* Since $\boldsymbol{S}$ is orthonormal, without loss of generality we can replace $\boldsymbol{S}\boldsymbol{x}$ with $\boldsymbol{x}$ in the numerator, since for all $\boldsymbol{x} \in \mathbb{R}^k$ there exists $\tilde{\boldsymbol{x}} \in \mathbb{R}^k$ such that $\boldsymbol{x} = \boldsymbol{S}\tilde{\boldsymbol{x}}$ and $||\boldsymbol{x}|| = ||\tilde{\boldsymbol{x}}||$. So it is enough to show that

$$\sup_{||\boldsymbol{x}|| \leq 1} \frac{\left|\left|(\boldsymbol{A} - \frac{1}{2m}\boldsymbol{d}\mathbf{1}^{\intercal})\boldsymbol{x}\right|\right|^2}{||\boldsymbol{x}||^2} \leq M$$

for some $M \in \mathbb{R}^+$. We have,

$$\left|\left|(\boldsymbol{A} - \frac{1}{2m}\boldsymbol{d}\mathbf{1}^{\intercal})\boldsymbol{x}\right|\right|^2 = \boldsymbol{x}^{\intercal}(\boldsymbol{A} - \frac{1}{2m}\boldsymbol{d}\mathbf{1}^{\intercal})^{\intercal}(\boldsymbol{A} - \frac{1}{2m}\boldsymbol{d}\mathbf{1}^{\intercal})\boldsymbol{x}$$

$$= \boldsymbol{x}^{\intercal}(\boldsymbol{A}^{\intercal}\boldsymbol{A} + \frac{\mathbf{1}\boldsymbol{d}^{\intercal}\boldsymbol{d}\mathbf{1}^{\intercal}}{4m^2} - \frac{\mathbf{1}\boldsymbol{d}^{\intercal}\boldsymbol{A} + \boldsymbol{A}^{\intercal}\boldsymbol{d}\mathbf{1}^{\intercal}}{2m})\boldsymbol{x}$$

$$= \boldsymbol{x}^{\intercal}\boldsymbol{A}^{\intercal}\boldsymbol{A}\boldsymbol{x} + \boldsymbol{x}^{\intercal}\left(\sum_{i=1}^{n}\frac{d_i^2}{4m^2}\mathbf{1}_{n \times n} - \frac{\mathbf{1}\boldsymbol{d}^{\intercal}\boldsymbol{A} + \boldsymbol{A}^{\intercal}\boldsymbol{d}\mathbf{1}^{\intercal}}{2m}\right)\boldsymbol{x}$$

Let us denote the neighborhood of a node $v_i$ by $\mathcal{N}(v_i)$. Then,

$$\boldsymbol{x}^{\intercal}\mathbf{1}\boldsymbol{d}^{\intercal}\boldsymbol{A}\boldsymbol{x} = \left(\sum_{i=1}^{n}x_i\right)\left(\sum_{i=1}^{n}\left(\sum_{j:v_j \in \mathcal{N}(v_i)}d_j\right)x_i\right)$$

$$\implies \boldsymbol{x}^{\intercal}\boldsymbol{A}^{\intercal}\boldsymbol{d}\mathbf{1}^{\intercal}\boldsymbol{x} = \left(\sum_{i=1}^{n}x_i\right)\left(\sum_{i=1}^{n}\left(\sum_{j:v_j \in \mathcal{N}(v_i)}d_j\right)x_i\right)$$

Additionally, it is easy to show that $\sum_{i=1}^{n}d_i^2 = \sum_{i=1}^{n}\left(\sum_{j:v_j \in \mathcal{N}(v_i)}d_j\right)$. Let us denote $r_i = \sum_{j:v_j \in \mathcal{N}(v_i)}d_j$. Then we have

$$\boldsymbol{x}^{\intercal}\left(\sum_{i=1}^{n}\frac{d_i^2}{4m^2}\mathbf{1}_{n \times n} - \frac{\mathbf{1}\boldsymbol{d}^{\intercal}\boldsymbol{A} + \boldsymbol{A}^{\intercal}\boldsymbol{d}\mathbf{1}^{\intercal}}{2m}\right)\boldsymbol{x} = \left(\sum_{i=1}^{n}\frac{x_i}{2m}\right)^2\sum_{i=1}^{n}r_i - \sum_{i=1}^{n}\left(\frac{x_i}{m}\right)\left(\sum_{i=1}^{n}r_ix_i\right)$$

By Cauchy-Schwarz inequality, we have,

$$-\sum_{i=1}^{n}r_ix_i \leq \sqrt{\sum_{i=1}^{n}r_i^2}\sqrt{\sum_{i=1}^{n}x_i^2}$$

$$\left(\sum_{i=1}^{n}x_i\right) \leq \sqrt{n\sum_{i=1}^{n}x_i^2}$$

$$\left(\sum_{i=1}^{n}x_i\right)\left(\sum_{i=1}^{n}r_i\right) \leq n\sqrt{\sum_{i=1}^{n}r_i^2}\sqrt{\sum_{i=1}^{n}x_i^2}$$

Hence,

$$\boldsymbol{x}^{\intercal} \left( \sum_{i=1}^{n} \frac{d_i^2}{4m^2} \mathbf{1}_{n \times n} - \frac{\mathbf{1}\boldsymbol{d}^{\intercal}\boldsymbol{A} + \boldsymbol{A}^{\intercal}\boldsymbol{d}\mathbf{1}^{\intercal}}{2m} \right) \boldsymbol{x} \leq \frac{n^{\frac{3}{2}}\sqrt{\sum_{i=1}^{n} r_i^2} \left( \sum_{i=1}^{n} x_i^2 \right)}{4m^2}(n+4m)$$

$$= \frac{n^{\frac{3}{2}}(n+4m)}{4m^2}\sqrt{\sum_{i=1}^{n} r_i^2} \, ||x||^2$$

From Proposition 3.1.2 in Brouwer and Haemers (2011) we know that $||A||_{\text{op}} \leq d_{\max}$, the maximum degree of the graph. So finally, we choose $M = \frac{n^{\frac{3}{2}}(n+4m)}{4m^2}\sqrt{\sum_{i=1}^{n} r_i^2} + d_{\max}^2$, and we are done. $\square$

Since the Fröbenius norm of a matrix is upper bounded by the square root of the rank times the operator norm (Equation (2.3.7) in Golub and Van Loan (1996)), we finally have $||\nabla_{\mathbf{S}} Q_{\text{prop}}||_F^2 \leq \frac{n^{\frac{5}{2}}(n+4m)}{4m^2}\sqrt{\sum_{i=1}^{n} r_i^2} + nd_{\max}^2$, or as a coarser upper bound, $||\nabla_{\mathbf{S}} Q_{\text{prop}}||_F \leq O(n^{1.75}m^{-0.25} + n^{1.25}m^{0.75} + n^{0.5}m)$. This indicates that the entries in the surrogate gradient matrix cannot be too large, and the surrogate gradient function has no singularities.

## C  EXTENDED RESULTS

### C.1  SEMI-SUPERVISED BASELINES

| Model | Encoder (Split) | Accuracy | F1 | Time (s) |
|---|---|---|---|---|
| GraFN | GCN (70-30) | $0.74 \pm 0.010$ | $0.72 \pm 0.009$ | 18.65 |
| | GCN (30-70) | $0.66 \pm 0.006$ | $0.64 \pm 0.007$ | 18.64 |
| | GAT (70-30) | $0.76 \pm 0.012$ | $0.71 \pm 0.052$ | 103.80 |
| | GAT (30-70) | $0.70 \pm 0.011$ | $0.60 \pm 0.075$ | 103.83 |
| | SAGE (70-30) | $0.67 \pm 0.010$ | $0.63 \pm 0.010$ | 10.89 |
| | SAGE (30-70) | $0.55 \pm 0.008$ | $0.51 \pm 0.010$ | 10.89 |
| ReVAR | GCN (70-30) | $0.35 \pm 0.019$ | $0.18 \pm 0.028$ | 43.74 |
| | GCN (30-70) | $0.35 \pm 0.017$ | $0.18 \pm 0.028$ | 43.53 |
| | GAT (70-30) | $0.43 \pm 0.023$ | $0.29 \pm 0.029$ | 385.26 |
| | GAT (30-70) | $0.42 \pm 0.017$ | $0.29 \pm 0.029$ | 378.15 |
| | SAGE (70-30) | $0.25 \pm 0.009$ | $0.15 \pm 0.006$ | 27.04 |
| | SAGE (30-70) | $0.24 \pm 0.005$ | $0.16 \pm 0.006$ | 28.67 |

Table 5: Performance metrics (Accuracy, F1-score, and Execution Time in seconds) of the semi supervised baselines for all datasets (except ArXiV) across 70-30 and 30-70 splits. Values are averages over five runs.

Table 5 represents the results along with the time required for each of ReVAR and GraFN. In these models, the embedding generation and classification process is degenerate for which the times reported are a combination of the two instead of just the embedders as reported in Table 2.

### C.2  ABLATION STUDY

To complement the analysis in the main text, we provide a more detailed view of the ablation experiments that disentangle the contributions of the semi-supervised and unsupervised components within FUSE. The learning rate of the FUSE algorithm was adjusted to $10^3$ for the *Only Unsupervised Component* case. All other relevant parameter values remain the same. Tables 6 and 7 summarize performance and runtime, respectively, across different classifiers and datasets for the 30-70 split, while Tables 8 and 9 show the same for the 70-30 split.

We conducted an experiment where, instead of orthonormalizing the embedding matrix after every iteration, we orthonormalized it at the very end. The runtimes have been reported in Table 10. The

| Classifier | Loss | Accuracy | F1 |
|------------|------|----------|-----|
| | Only Semi-supervised Component | 0.690 | 0.657 |
| **GAT** | Both components | 0.697 | 0.666 |
| | Only Unsupervised Component | 0.688 | 0.657 |
| | Only Semi-supervised Component | 0.656 | 0.633 |
| **GCN** | Both components | 0.668 | 0.649 |
| | Only Unsupervised Component | 0.660 | 0.637 |
| | Only Semi-supervised Component | 0.525 | 0.530 |
| **SAGE** | Both components | 0.732 | 0.707 |
| | Only Unsupervised Component | 0.716 | 0.696 |

Table 6: Classification accuracy and F1-score across different FUSE variants and classifiers on the 30-70 split averaged across datasets.

| Embedding | Dataset | | | | | Average |
|-----------|---------|----------|--------------|--------|--------|---------|
| | Cora | CiteSeer | Amazon Photo | WikiCS | PubMed | |
| Only Semisupervised | 8.18 | 8.04 | 50.18 | 79.60 | 103.62 | 49.124 |
| Only Unsupervised | 3.44 | 4.84 | 11.04 | 24.02 | 42.63 | 17.194 |
| Both | 8.99 | 8.63 | 46.86 | 77.39 | 126.36 | 53.246 |

Table 7: Execution times (in seconds) of different FUSE components across datasets for 30-70 split.

| Classifier | Loss | Accuracy | F1 |
|------------|------|----------|-----|
| | Only Semi-supervised Component | 0.75 | 0.72 |
| **GAT** | Both components | 0.74 | 0.72 |
| | Only Unsupervised Component | 0.74 | 0.72 |
| | Only Semi-supervised Component | 0.71 | 0.68 |
| **GCN** | Both components | 0.70 | 0.68 |
| | Only Unsupervised Component | 0.71 | 0.68 |
| | Only Semi-supervised Component | 0.69 | 0.66 |
| **SAGE** | Both components | 0.76 | 0.73 |
| | Only Unsupervised Component | 0.75 | 0.73 |

Table 8: Classification accuracy and F1-score across different FUSE variants and classifiers on the 70-30 split averaged across datasets.

| Embedding | Dataset | | | | | Average |
|-----------|---------|----------|--------------|--------|--------|---------|
| | Cora | CiteSeer | Amazon Photo | WikiCS | PubMed | |
| Only Semisupervised | 6.74 | 7.79 | 31.04 | 53.81 | 69.69 | 33.814 |
| Only Unsupervised | 3.49 | 4.15 | 10.73 | 23.84 | 34.66 | 15.574 |
| Both | 8.08 | 7.29 | 33.76 | 58.75 | 70.07 | 35.59 |

Table 9: Execution times (in seconds) of different FUSE components across datasets for 70-30 split.

Table 10: Runtime (seconds) of FUSE across datasets for the 70–30 and 30–70 splits.

| Split | Cora | CiteSeer | Amazon Photo | WikiCS | PubMed | ArXiV |
|-------|------|----------|--------------|--------|--------|-------|
| **70–30** | $10.51 \pm 0.479$ | $10.19 \pm 0.139$ | $56.22 \pm 0.961$ | $92.04 \pm 1.549$ | $104.62 \pm 1.358$ | 1221.85 |
| **30–70** | $13.18 \pm 0.289$ | $11.89 \pm 0.211$ | $83.51 \pm 0.783$ | $137.48 \pm 0.820$ | $136.89 \pm 1.368$ | 1705.38 |

Table 11: Accuracy and F1 (70–30 split) of FUSE across datasets using GAT, GCN and SAGE.

| Dataset | GAT | | GCN | | SAGE | |
|---------|-----|-----|-----|-----|------|-----|
| | **Accuracy** | **F1** | **Accuracy** | **F1** | **Accuracy** | **F1** |
| **CiteSeer** | $0.72 \pm 0.016$ | $0.68 \pm 0.012$ | $0.67 \pm 0.008$ | $0.64 \pm 0.004$ | $0.69 \pm 0.014$ | $0.67 \pm 0.010$ |
| **Cora** | $0.86 \pm 0.009$ | $0.85 \pm 0.012$ | $0.82 \pm 0.013$ | $0.80 \pm 0.018$ | $0.84 \pm 0.007$ | $0.83 \pm 0.008$ |
| **Amazon Photo** | $0.92 \pm 0.004$ | $0.91 \pm 0.004$ | $0.91 \pm 0.006$ | $0.90 \pm 0.007$ | $0.89 \pm 0.007$ | $0.88 \pm 0.010$ |
| **PubMed** | $0.79 \pm 0.011$ | $0.79 \pm 0.011$ | $0.80 \pm 0.007$ | $0.79 \pm 0.008$ | $0.77 \pm 0.011$ | $0.75 \pm 0.013$ |
| **WikiCS** | $0.81 \pm 0.002$ | $0.79 \pm 0.008$ | $0.76 \pm 0.006$ | $0.74 \pm 0.009$ | $0.73 \pm 0.011$ | $0.69 \pm 0.017$ |
| **Averaged** | $0.82 \pm 0.008$ | $0.80 \pm 0.009$ | $0.79 \pm 0.008$ | $0.77 \pm 0.009$ | $0.79 \pm 0.010$ | $0.76 \pm 0.012$ |
| **ArXiV$_1$** | 0.40 | 0.09 | 0.53 | 0.25 | 0.47 | 0.11 |
| **ArXiV$_2$** | 0.63 | 0.44 | 0.49 | 0.24 | 0.59 | 0.19 |
| **ArXiV$_3$** | 0.68 | 0.46 | 0.50 | 0.24 | 0.62 | 0.24 |

dataset-wise (except ArXiV) and the averaged results are shown in Table 11 and Table 12. ArXiV$_1$ denotes the results for the ArXiV dataset in which we orthonormalize the embedding matrix at the very end instead of doing it every iteration.

**Case-1** : We orthonormalized S at the very end instead of doing it every iteration. It took 367.31 seconds for the 70-30 split and 492.46 seconds for the 30-70 split.

**Case-2** : We orthonormalized S in every iteration. It took 650.91 seconds for the 70-30 split and 782.43 seconds for the 30-70 split.

The results have been reported in Table 11 and Table 12 respectively as ArXiV$_2$ (for Case-1) and ArXiV$_3$ (for Case-2). From the results, we observe that orthonormalizing at the very end instead of every iteration indeed takes slightly less time (the margin is greater when performed on a stronger CPU), but degrades performance in some cases, such as ArXiV. Hence, we recommend using orthonormalization per iteration, which incurs a cost of $O(nk^2)$. This is included in Section 3.2 (paragraph: Computational Complexity).

**Performance Across Classifiers.** As shown in Tables 6 and 8, the relative contribution of each component is consistent across GAT, GCN, and GraphSAGE. Notably, embeddings trained with only the unsupervised modularity term are better than those using only the semi-supervised term on an average. This confirms that community structure provides a strong inductive bias even when label information is sparse. However, combining both objectives consistently yields the highest accuracy and F1-scores overall, demonstrating that structural and label-based signals are complementary rather than interchangeable. Interestingly, the performance gap between "Both" and "Unsupervised only" is smaller than that between "Both" and "Semi-supervised only", especially for GraphSAGE, suggesting that topology carries more transferable information than a small label set in these benchmarks.

**Runtime Considerations.** Tables 7 and 9 highlight that the efficiency of FUSE is not compromised by integrating multiple objectives. The combined loss incurs only a marginal overhead relative to either component in isolation, while producing markedly better embeddings. This efficiency gain stems from the linearized modularity update, which dominates the runtime irrespective of whether label propagation is included. We also observe that datasets with a larger number of nodes and denser connectivity like PubMed and WikiCS yield proportionally higher execution times, but the scaling behavior remains consistent across variants.

These results provide additional evidence that FUSE's strength lies not in any single component, but in their unification. The unsupervised modularity term ensures that embeddings respect community structure, while the semi-supervised propagation aligns them with available labels. Their joint optimization balances exploration of global topology with exploitation of label information, leading to robust performance without significant runtime penalties.

Table 12: Accuracy and F1 (30–70 split) of FUSE across datasets using GAT, GCN and SAGE.

| Dataset | GAT | | GCN | | SAGE | |
|---|---|---|---|---|---|---|
| | Accuracy | F1 | Accuracy | F1 | Accuracy | F1 |
| CiteSeer | $0.63 \pm 0.003$ | $0.59 \pm 0.003$ | $0.61 \pm 0.009$ | $0.57 \pm 0.009$ | $0.62 \pm 0.009$ | $0.59 \pm 0.008$ |
| Cora | $0.81 \pm 0.007$ | $0.79 \pm 0.008$ | $0.78 \pm 0.002$ | $0.77 \pm 0.004$ | $0.79 \pm 0.004$ | $0.78 \pm 0.003$ |
| Amazon Photo | $0.92 \pm 0.002$ | $0.91 \pm 0.003$ | $0.90 \pm 0.004$ | $0.89 \pm 0.004$ | $0.89 \pm 0.002$ | $0.87 \pm 0.003$ |
| PubMed | $0.79 \pm 0.003$ | $0.79 \pm 0.003$ | $0.79 \pm 0.002$ | $0.78 \pm 0.002$ | $0.76 \pm 0.003$ | $0.74 \pm 0.004$ |
| WikiCS | $0.79 \pm 0.002$ | $0.77 \pm 0.002$ | $0.76 \pm 0.006$ | $0.73 \pm 0.006$ | $0.72 \pm 0.007$ | $0.68 \pm 0.009$ |
| Averaged | $0.79 \pm 0.003$ | $0.77 \pm 0.004$ | $0.77 \pm 0.004$ | $0.75 \pm 0.005$ | $0.76 \pm 0.005$ | $0.73 \pm 0.005$ |
| $ArXiV_1$ | 0.49 | 0.16 | 0.54 | 0.27 | 0.50 | 0.14 |
| $ArXiV_2$ | 0.64 | 0.47 | 0.41 | 0.10 | 0.59 | 0.26 |
| $ArXiV_3$ | 0.64 | 0.44 | 0.45 | 0.14 | 0.59 | 0.24 |

## C.3 SENSITIVITY ANALYSIS

To further assess the robustness of FUSE, we carried out a sensitivity analysis of its main hyperparameters across datasets. Tables 13 (a, b, c) summarize the optimal settings discovered under two search protocols. These results provide insights into which hyperparameters consistently influence performance and which are less critical.

**Influential hyperparameters.** Among the parameters, the learning rate $\eta$ and the loss weights $\lambda_{sup}$ and $\lambda_{semi}$ emerge as the most sensitive across datasets. Small variations in $\eta$ often lead to pronounced differences in both accuracy and convergence speed, indicating the need for dataset-specific tuning. Similarly, the balance between the supervised and semi-supervised terms must be carefully adjusted, as an overemphasis on one can suppress the benefits of the other. By contrast, the neighborhood radius $r$ and structural depths $L, L'$ showed more stable behavior, with broad ranges yielding near-optimal accuracy.

**Consistency Across Datasets.** Interestingly, although the exact optimal values vary, the relative importance of hyperparameters remains consistent. For example, on both Cora and PubMed, adjusting $\lambda_{semi}$ within $[2, 2.5]$ was essential to achieve competitive performance, while on WikiCS and CiteSeer, a more balanced weighting was required. The Amazon Photo dataset was less sensitive overall, achieving high accuracy under multiple configurations, suggesting that denser graphs with richer labels are inherently more robust to hyperparameter shifts.

**Runtime Trade-offs.** The sensitivity analysis also reveals a runtime–performance trade-off. While larger values of $T$ or deeper $L, L'$ occasionally yield marginal accuracy gains, they incur disproportionately higher costs in training time (e.g., PubMed in Table 13 (a and b)). This indicates diminishing returns from overparameterization, and reinforces the practical value of moderate configurations that balance accuracy and efficiency.

## C.4 SCALABILITY EXPERIMENTS

1. **Purpose and Setup for ArXiV.** In addition to the experiments above, we performed another benchmarking experiment on dataset, namely ArXiV($\sim$ 169K nodes, $\sim$ 1.1M edges) for investigating the scalability of FUSE. This dataset is highly imbalanced as well. Given that the dataset is significantly larger than others for FUSE we have considered a learning rate of 0.05 to ensure convergence within 200 iterations. For VGAE, we took the initial matrix to be a $n \times k$ random matrix instead of the $n \times n$ identity matrix as assumed in other experiments. This is to avoid scalability issues due to very large value of $n$ for this dataset. All other parameters remain the same.

   **Observations on ArXiV.** The results across the two splits (30-70 and 70-30) for a fixed seed is given in Tables 19 and 20. The results reveal that FUSE is not only scalable and robust to labels, but performs at par with unsupervised algorithms like Node2Vec and DeepWalk in terms of performance metrics. Furthermore, it offers a significant advantage in terms of computational time. In addition, it outperforms the semi supervised algorithms like GraFN and ReVAR; in terms of Accuracy, F1 Score by a large margin. Notably, the MNMF algorithm was not scalable to this particular dataset.

2. We performed additional scalability analyses on two yet larger datasets: MAG ($\sim$736K nodes, $\sim$8M citation edges) and ogbn products ($\sim$2.45M nodes, $\sim$61.9M edges) using a 30-70 split (70% label masking). Since DeepWalk and Node2Vec consistently achieve the strongest accuracy and F1 scores among the baseline embedding methods, and because their performance remains stable even with shorter walk lengths (5) and fewer walks (10), we report comparisons against DeepWalk using these reduced parameters. We also include the given embedding as a high-end benchmark. These reductions substantially lower the computational cost of the random-walk baselines while preserving their representative performance, providing a meaningful reference point for FUSE in terms of scalability. We exclude GAT from these comparisons due to its high computational overhead and instead evaluate against GCN and GraphSAGE.

Our observations (Tables 14 and 15) are as follows:

   (a) FUSE remains faster on MAG compared to DeepWalk, with the unsupervised variant being at least three times faster. On the ogbn products dataset, the unsupervised version of FUSE completes in approximately 2.5 hours. In contrast, DeepWalk could not complete within 24 hours while the full version takes a little more than 10 hours using the standard Python implementation with a single CPU worker and no GPU.

   (b) While FUSE is fast, in a few cases it sacrifices Accuracy and F1-Score, and this performance gap becomes more pronounced on larger datasets. Therefore, the applicability of FUSE is most relevant in feature-agnostic settings where fast embedding generation is the primary requirement.

   (c) FUSE is compatible with GCN but performs less effectively with GraphSAGE.

   isting baselines. We measured one intrinsic metric, the DB Index, as well as two extrinsic metrics,

3. **Analysis for Node2Vec and DeepWalk for a lower `walk_length`.** To address the potential concern that the default `walk_length` of 80 for Node2Vec and DeepWalk might inflate their runtimes, we conducted an additional experiment with a reduced `walk_length` of 5 for a single seed for these two algorithms. Tables 16, 17, and 18 summarize the results of this experiment across all datasets, reporting classification accuracy, F1-score, and runtime for both 70-30 and 30-70 train-test splits.

   **Performance Analysis:** For most datasets, the classification performance of Node2Vec and DeepWalk with the shorter walk length remained largely comparable to that obtained with the default longer walk, suggesting that reducing the walk length does not severely compromise the quality of learned embeddings.

   **Runtime Comparison:** Reducing the `walk_length` substantially improved the runtime of both Node2Vec and DeepWalk across datasets. As reported in Table 18, runtime reductions of FUSE regarding these two algorithms are particularly significant for large datasets like Photos, WikiCS, and ArXiV with more edges. For example, on ArXiV, DeepWalk and Node2Vec required approximately 3,100–3,200 seconds for the 70-30 split, whereas FUSE completed within 1,360 seconds, which is roughly a 3 times improvement in speed.

   **FUSE Advantage:** Despite the reduction in random walk length for Node2Vec and DeepWalk, FUSE was consistently equally or more effective in both performance and runtime metrics. FUSE embeddings yielded higher classification accuracy and F1-scores compared to DeepWalk and Node2Vec especially for a larger dataset like ArXiV with a higher number of edges, even when the latter used a very short walk length. This indicates that FUSE's embedding methodology is not only scalable but also robust to variations in graph size and connectivity, offering a more efficient alternative for large-scale graph representation learning (Tables 16, 17).

## C.5 EXPERIMENTS ON DIFFERENT MASKING MECHANISMS

We also performed experiments on various masking rates and mechanisms to investigate the robustness of our method. We analyzed our method on 3 types of simulated masking mechanisms, based on the 3 types of missingness as described in Rubin (1976). The notations of MCAR, MAR and MNAR have been redefined for our specific use case. We describe these mechanisms here:

| Dataset | $k$ | $\eta$ | $\lambda_{\text{sup}}$ | $\lambda_{\text{semi}}$ | $T$ | $r$ | $L$ | $L'$ | Accuracy (%) | Time (s) |
|---|---|---|---|---|---|---|---|---|---|---|
| Cora | 145 | 0.31 | 0.6 | 1.9 | 200 | 20 | 4 | 1 | 80.47 | 18.92 |
| CiteSeer | 135 | 0.51 | 0.8 | 1.5 | 450 | 13 | 5 | 3 | 63.32 | 25.56 |
| PubMed | 155 | 0.11 | 0.9 | 2.0 | 450 | 12 | 9 | 1 | 81.17 | 226.58 |
| WikiCS | 130 | 0.28 | 1.1 | 1.1 | 200 | 20 | 3 | 2 | 74.05 | 76.66 |
| Amazon Photo | 100 | 0.21 | 1.7 | 2.5 | 300 | 13 | 3 | 2 | 89.15 | 59.47 |

(a) Optimal hyperparameters in the 30-70 setup.

| Dataset | $k$ | $\eta$ | $\lambda_{\text{sup}}$ | $\lambda_{\text{semi}}$ | $T$ | $r$ | $L$ | $L'$ | Accuracy (%) | Time (s) |
|---|---|---|---|---|---|---|---|---|---|---|
| Cora | 170 | 0.35 | 0.5 | 2.5 | 250 | 20 | 3 | 2 | 85.59 | 21.14 |
| CiteSeer | 200 | 0.79 | 2.2 | 1.1 | 300 | 15 | 4 | 3 | 73.74 | 24.75 |
| PubMed | 180 | 0.59 | 2.2 | 1.2 | 350 | 18 | 3 | 3 | 84.09 | 303.46 |
| WikiCS | 140 | 0.37 | 1.8 | 2.3 | 250 | 20 | 3 | 1 | 76.75 | 78.22 |
| Amazon Photo | 120 | 0.79 | 2.1 | 1.1 | 100 | 12 | 4 | 3 | 90.71 | 34.42 |

(b) Optimal hyperparameters in the 70-30 setup.

| Dataset | $\eta$ | $\lambda_{\text{sup}}$ | $\lambda_{\text{semi}}$ | $r$ | $L$ | $L'$ | Accuracy (%) | Time (s) |
|---|---|---|---|---|---|---|---|---|
| Cora | 0.25 | 0.9 | 2.3 | 17 | 4 | 9 | 80.16 | 21.12 |
| CiteSeer | 0.35 | 1.3 | 1.7 | 15 | 7 | 2 | 63.27 | 35.57 |
| PubMed | 0.46 | 0.8 | 2.5 | 18 | 4 | 1 | 80.93 | 79.95 |
| WikiCS | 0.03 | 0.8 | 1.3 | 15 | 3 | 2 | 73.79 | 71.44 |
| Amazon Photo | 0.49 | 1.1 | 2.3 | 9 | 3 | 10 | 89.09 | 44.61 |

(c) Optimal hyperparameters under $k=150$, $T=200$ for the 30-70 setup.

Table 13: Optimal hyperparameters of FUSE.

Table 14: Results on the MAG dataset with mask fraction **0.7** (30-70 split).

| Embedding | Classifier | Embed Time (s) | Accuracy | F1 Score |
|---|---|---|---|---|
| FUSE | GCN | 4075.66 | 0.241 | 0.094 |
| FUSE | SAGE | 4075.66 | 0.154 | 0.009 |
| FUSE (unsup) | GCN | 1520.15 | 0.13 | 0.018 |
| FUSE (unsup) | SAGE | 1520.15 | 0.12 | 0.008 |
| DeepWalk (walk_length=5) | GCN | 5549.27 | 0.041 | 0.000 |
| DeepWalk (walk_length=5) | SAGE | 5549.27 | 0.223 | 0.203 |
| Given | GCN | - | 0.082 | 0.002 |
| Given | SAGE | - | 0.224 | 0.023 |

Table 15: Results on the obgn products dataset with mask fraction **0.3** (30-70 split).

| Embedding | Classifier | Embed Time (s) | Accuracy | F1 Score |
|---|---|---|---|---|
| FUSE | GCN | 36571.08 | 0.801 | 0.443 |
| FUSE | SAGE | 36571.08 | 0.273 | 0.009 |
| FUSE (unsup) | GCN | 10334.95 | 0.706 | 0.326 |
| FUSE (unsup) | SAGE | 10334.95 | 0.510 | 0.175 |
| DeepWalk (walk_length=5) | GCN | NA | NA | NA |
| DeepWalk (walk_length=5) | SAGE | NA | NA | NA |
| Given | GCN | - | 0.61 | 0.255 |
| Given | SAGE | - | 0.759 | 0.251 |

| Classifier | Embedding | 70-30 Split | | 30-70 Split | |
|---|---|---|---|---|---|
| | | Accuracy | F1 | Accuracy | F1 |
| GAT | DeepWalk (walk_length=5) | 0.81 | 0.793 | **0.78** | **0.764** |
| | Node2Vec (walk_length=5) | 0.81 | 0.792 | **0.78** | 0.756 |
| | FUSE | **0.82** | **0.795** | **0.78** | 0.751 |
| GCN | DeepWalk (walk_length=5) | 0.62 | 0.552 | 0.65 | 0.578 |
| | Node2Vec (walk_length=5) | 0.62 | 0.554 | 0.65 | 0.598 |
| | FUSE | **0.77** | **0.753** | **0.73** | **0.699** |
| SAGE | DeepWalk (walk_length=5) | **0.81** | **0.786** | **0.78** | **0.754** |
| | Node2Vec (walk_length=5) | **0.81** | 0.781 | 0.77 | 0.751 |
| | FUSE | 0.79 | 0.769 | 0.75 | 0.731 |

Table 16: Classification accuracy and F1-score (averaged) for DeepWalk (walk_length=5), Node2Vec (walk_length=5) and FUSE across three classifiers for all the datasets (except ArXiV) for a fixed seed. Results are reported for both 70-30 and 30-70 train-test splits.

| Classifier | Embedding | 70-30 Split | | 30-70 Split | |
|---|---|---|---|---|---|
| | | Accuracy | F1 | Accuracy | F1 |
| GAT | DeepWalk (walk_length=5) | 0.66 | 0.42 | **0.65** | 0.40 |
| | Node2Vec (walk_length=5) | 0.64 | 0.39 | 0.64 | 0.38 |
| | FUSE | **0.67** | **0.47** | 0.64 | **0.43** |
| GCN | DeepWalk (walk_length=5) | 0.47 | 0.18 | 0.48 | 0.21 |
| | Node2Vec (walk_length=5) | 0.46 | 0.15 | **0.49** | **0.22** |
| | FUSE | **0.50** | **0.24** | 0.45 | 0.14 |
| SAGE | DeepWalk (walk_length=5) | 0.61 | **0.23** | **0.60** | 0.23 |
| | Node2Vec (walk_length=5) | 0.59 | 0.22 | 0.58 | 0.21 |
| | FUSE | **0.62** | **0.23** | **0.60** | **0.25** |

Table 17: Classification accuracy and F1-score for DeepWalk (walk_length=5), Node2Vec (walk_length=5) and FUSE across three classifiers for ArXiV for a fixed seed. Results are reported for both 70-30 and 30-70 train-test splits. The best metric values across each classifier have been highlighted in **bold**.

| Embedding | Cora | CiteSeer | Amazon Photo | WikiCS | PubMed | ArXiV |
|---|---|---|---|---|---|---|
| **70-30 Split** | | | | | | |
| DeepWalk (walk_length=5) | 3.92 | 4.23 | 152.48 | 412.52 | 36.04 | 3290.21 |
| Node2Vec (walk_length=5) | **3.64** | **3.84** | 154.83 | 417.46 | **35.44** | 3217.85 |
| FUSE | 12.67 | 13.30 | **49.15** | **84.88** | 96.76 | **1360.30** |
| **30-70 Split** | | | | | | |
| DeepWalk (walk_length=5) | 3.72 | 3.81 | 155.53 | 421.54 | 36.01 | 3143.32 |
| Node2Vec (walk_length=5) | **3.65** | **3.80** | 154.27 | 423.19 | **35.90** | 3141.81 |
| FUSE | 14.25 | 14.55 | **64.63** | **111.87** | 104.89 | 1**698.52** |

Table 18: Runtime comparison (in seconds) of DeepWalk (walk_length=5), Node2Vec (walk_length=5) and FUSE across datasets under 70-30 and 30-70 train-test splits for a fixed seed. The least runtimes have been highlighted in **bold**.

- Masking-Completely-At-Random (MCAR): The probability of a node label being masked is independent of the data.

- Masking-At-Random (MAR): The probability of a node label being masked is dependent on the feature vector of the node.

- Masking-Not-At-Random (MNAR): The probability of a node label being masked depends both on the feature vector of the node, and the label itself.

We simulated these masking scenarios using a procedure similar to Jarrett et al. (2022), where the masks were generated using a logistic model with random coefficients. Further details can be found in the attached code. For each masking scenario, we tested 3 masking rates: $0.2, 0.5$ and $0.8$, and reported the mean and standard deviations of the classification accuracy and F1 score over 10 iterations with different random seeds. The Multi-Layer Perceptron (MLP) was chosen to have depth and width equivalent to the graph neural network models, in this case 2 and 16 respectively. The associated results are given in Tables 38– 43.

| Classifier | Embedding | Accuracy (%) | F1 Score | Time (s) |
|---|---|---|---|---|
| **GCN** | Random | 22.58 | 0.05 | 0.4303 |
| | DeepWalk | 51.43 | **0.27** | 12996.76 |
| | Node2Vec | 50.32 | 0.25 | 12038.33 |
| | VGAE | 16.17 | 0.01 | 1098.25 |
| | DGI | 16.16 | 0.01 | 758.04 |
| | FUSE | **59.65** | 0.25 | 1698.52 |
| | GraFN | 26.28 | 0.08 | **360.64** |
| | ReVAR | 16.14 | 0.01 | 468.55 |
| | Given | 41.22 | 0.10 | 0.0521 |
| **GAT** | Random | 19.08 | 0.02 | 0.4303 |
| | DeepWalk | **67.65** | **0.44** | 12996.76 |
| | Node2Vec | 66.86 | **0.44** | 12038.33 |
| | VGAE | 16.16 | 0.01 | 1098.25 |
| | DGI | 16.16 | 0.01 | **758.04** |
| | FUSE | 63.83 | 0.43 | 1698.52 |
| | GraFN | 57.04 | 0.39 | 13712.21 |
| | ReVAR | 16.16 | 0.01 | 9265.35 |
| | Given | 56.74 | 0.28 | 0.0521 |
| **SAGE** | Random | 15.13 | 0.02 | 0.4303 |
| | DeepWalk | **61.95** | 0.23 | 12996.76 |
| | Node2Vec | 61.75 | 0.24 | 12038.33 |
| | VGAE | 16.16 | 0.01 | 1098.25 |
| | DGI | 16.16 | 0.01 | 758.04 |
| | FUSE | 59.65 | **0.25** | 1698.52 |
| | GraFN | 36.49 | 0.13 | **199.11** |
| | ReVAR | 15.81 | 0.01 | 285.55 |
| | Given | 53.65 | 0.18 | 0.0521 |

Table 19: Performance of different embedding–classifier pairs (except GraFN and ReVAR as they do have degenerate embedders and classifiers) on the ArXiv dataset (30–70 split) for a fixed seed. Embedding generation times were added across each of the embeddings except GraFN and ReVAR for which the time required by each encoder is given separately. The best and second-best in each metric for each classifier are highlighted in **bold** and underlined, respectively.

| Classifier | Embedding | Accuracy | F1 Score | Time (s) |
|------------|-----------|----------|----------|----------|
| **GCN** | Random | 33.19 | 0.4628 | |
| | DeepWalk | **50.04** | 0.2180 | 13029.78 |
| | Node2Vec | 49.20 | 0.1992 | 12899.23 |
| | VGAE | 13.12 | 0.0058 | 1072.42 |
| | DGI | 16.37 | 0.0070 | 633.06 |
| | FUSE | 49.97 | **0.2353** | 1360.30 |
| | GraFN | 26.21 | 0.07 | **360.17** |
| | ReVAR | 16.37 | 0.01 | 432.88 |
| | Given | 38.19 | 0.0794 | 0.0473 |
| **GAT** | Random | 22.37 | 0.0300 | 0.4628 |
| | DeepWalk | **68.58** | 0.4601 | 13029.78 |
| | Node2Vec | 67.87 | 0.4506 | 12899.23 |
| | VGAE | 13.44 | 0.0082 | 1072.42 |
| | DGI | 13.13 | 0.0073 | **633.06** |
| | FUSE | 67.45 | **0.4682** | 1360.30 |
| | GraFN | 61.35 | 0.43 | 13564.56 |
| | ReVAR | 16.37 | 0.01 | 9462.59 |
| | Given | 58.74 | 0.3294 | 0.0473 |
| **SAGE** | Random | 16.15 | 0.0163 | 0.4628 |
| | DeepWalk | **62.39** | **0.2421** | 13029.78 |
| | Node2Vec | 62.03 | **0.2421** | 12899.73 |
| | VGAE | 16.53 | 0.0092 | 1072.42 |
| | DGI | 16.37 | 0.0070 | 633.06 |
| | FUSE | 61.91 | 0.2344 | 1360.30 |
| | GraFN | 44.73 | 0.17 | **248.13** |
| | ReVAR | 16.13 | 0.01 | 321.53 |
| | Given | 54.16 | 0.1792 | 0.0473 |

Table 20: Performance of different embedding–classifier pairs (except GraFN and ReVAR as they have degenerate embedders and classifiers) on the ArXiv dataset (70–30 split) for a fixed seed. Embedding generation times were added across each of the embeddings except GraFN and ReVAR for which the time required by each encoder is given separately. The best and second-best in each metric, for each classifier are highlighted in **bold** and underlined, respectively.

Table 21: Cora – Accuracy and F1 for 70–30 and 30–70 Splits

| Classifier | Embedding | 70–30 Split | | 30–70 Split | |
|---|---|---|---|---|---|
| | | Accuracy | F1 | Accuracy | F1 |
| GAT | DeepWalk | 0.85 ± 0.010 | 0.84 ± 0.012 | 0.80 ± 0.009 | 0.79 ± 0.010 |
| | DGI | 0.62 ± 0.225 | 0.58 ± 0.262 | 0.56 ± 0.225 | 0.47 ± 0.339 |
| | FUSE | 0.86 ± 0.009 | 0.85 ± 0.011 | 0.81 ± 0.007 | 0.80 ± 0.006 |
| | Given | 0.87 ± 0.011 | 0.86 ± 0.015 | 0.83 ± 0.010 | 0.82 ± 0.010 |
| | Node2Vec | 0.85 ± 0.014 | 0.84 ± 0.016 | 0.80 ± 0.005 | 0.78 ± 0.007 |
| | Random | 0.84 ± 0.008 | 0.83 ± 0.012 | 0.74 ± 0.004 | 0.73 ± 0.005 |
| | VGAE | 0.86 ± 0.007 | 0.86 ± 0.009 | 0.79 ± 0.010 | 0.78 ± 0.010 |
| GCN | DeepWalk | 0.82 ± 0.012 | 0.80 ± 0.013 | 0.78 ± 0.016 | 0.76 ± 0.018 |
| | DGI | 0.30 ± 0.101 | 0.10 ± 0.084 | 0.28 ± 0.123 | 0.08 ± 0.060 |
| | FUSE | 0.82 ± 0.011 | 0.81 ± 0.012 | 0.79 ± 0.006 | 0.78 ± 0.004 |
| | Given | 0.82 ± 0.007 | 0.81 ± 0.011 | 0.81 ± 0.011 | 0.79 ± 0.015 |
| | Node2Vec | 0.81 ± 0.015 | 0.80 ± 0.017 | 0.78 ± 0.012 | 0.76 ± 0.013 |
| | Random | 0.68 ± 0.019 | 0.67 ± 0.019 | 0.60 ± 0.011 | 0.58 ± 0.013 |
| | VGAE | 0.82 ± 0.009 | 0.81 ± 0.009 | 0.78 ± 0.008 | 0.76 ± 0.010 |
| SAGE | DeepWalk | 0.84 ± 0.021 | 0.83 ± 0.024 | 0.79 ± 0.009 | 0.78 ± 0.011 |
| | DGI | 0.57 ± 0.168 | 0.51 ± 0.221 | 0.52 ± 0.143 | 0.45 ± 0.201 |
| | FUSE | 0.84 ± 0.012 | 0.83 ± 0.012 | 0.79 ± 0.006 | 0.78 ± 0.006 |
| | Given | 0.86 ± 0.012 | 0.85 ± 0.014 | 0.83 ± 0.012 | 0.81 ± 0.020 |
| | Node2Vec | 0.84 ± 0.016 | 0.83 ± 0.016 | 0.78 ± 0.014 | 0.77 ± 0.017 |
| | Random | 0.62 ± 0.038 | 0.59 ± 0.036 | 0.41 ± 0.031 | 0.34 ± 0.029 |
| | VGAE | 0.84 ± 0.006 | 0.83 ± 0.012 | 0.78 ± 0.009 | 0.77 ± 0.008 |

Table 22: CiteSeer – Accuracy and F1 for 70–30 and 30–70 Splits

| Classifier | Embedding | 70–30 Split | | 30–70 Split | |
|---|---|---|---|---|---|
| | | Accuracy | F1 | Accuracy | F1 |
| **GAT** | DeepWalk | 0.70 ± 0.021 | 0.66 ± 0.019 | 0.61 ± 0.010 | 0.58 ± 0.009 |
| | DGI | 0.31 ± 0.190 | 0.20 ± 0.234 | 0.31 ± 0.144 | 0.20 ± 0.179 |
| | FUSE | 0.71 ± 0.014 | 0.68 ± 0.010 | 0.63 ± 0.008 | 0.59 ± 0.006 |
| | Given | 0.74 ± 0.016 | 0.71 ± 0.013 | 0.69 ± 0.006 | 0.66 ± 0.006 |
| | Node2Vec | 0.69 ± 0.019 | 0.65 ± 0.018 | 0.60 ± 0.004 | 0.57 ± 0.005 |
| | Random | 0.70 ± 0.014 | 0.66 ± 0.011 | 0.58 ± 0.006 | 0.55 ± 0.007 |
| | VGAE | 0.71 ± 0.012 | 0.67 ± 0.010 | 0.61 ± 0.005 | 0.58 ± 0.005 |
| **GCN** | DeepWalk | 0.60 ± 0.026 | 0.56 ± 0.027 | 0.57 ± 0.011 | 0.54 ± 0.013 |
| | DGI | 0.20 ± 0.024 | 0.06 ± 0.003 | 0.22 ± 0.029 | 0.08 ± 0.046 |
| | FUSE | 0.67 ± 0.012 | 0.64 ± 0.010 | 0.62 ± 0.004 | 0.58 ± 0.004 |
| | Given | 0.69 ± 0.005 | 0.66 ± 0.007 | 0.68 ± 0.009 | 0.64 ± 0.010 |
| | Node2Vec | 0.59 ± 0.020 | 0.55 ± 0.028 | 0.56 ± 0.013 | 0.53 ± 0.014 |
| | Random | 0.50 ± 0.011 | 0.47 ± 0.011 | 0.42 ± 0.009 | 0.40 ± 0.009 |
| | VGAE | 0.61 ± 0.012 | 0.58 ± 0.010 | 0.57 ± 0.011 | 0.54 ± 0.012 |
| **SAGE** | DeepWalk | 0.68 ± 0.011 | 0.64 ± 0.013 | 0.61 ± 0.005 | 0.57 ± 0.008 |
| | DGI | 0.31 ± 0.102 | 0.24 ± 0.117 | 0.32 ± 0.085 | 0.26 ± 0.101 |
| | FUSE | 0.70 ± 0.015 | 0.67 ± 0.011 | 0.62 ± 0.010 | 0.59 ± 0.009 |
| | Given | 0.75 ± 0.016 | 0.72 ± 0.014 | 0.70 ± 0.008 | 0.66 ± 0.004 |
| | Node2Vec | 0.67 ± 0.015 | 0.63 ± 0.013 | 0.59 ± 0.011 | 0.55 ± 0.007 |
| | Random | 0.48 ± 0.021 | 0.43 ± 0.018 | 0.30 ± 0.019 | 0.25 ± 0.015 |
| | VGAE | 0.68 ± 0.010 | 0.63 ± 0.006 | 0.58 ± 0.003 | 0.54 ± 0.007 |

Table 23: Amazon Photo – Accuracy and F1 for 70–30 and 30–70 Splits

| Classifier | Embedding | 70–30 Split | | 30–70 Split | |
| --- | --- | --- | --- | --- | --- |
| | | Accuracy | F1 | Accuracy | F1 |
| GAT | DeepWalk | 0.93 ± 0.005 | 0.93 ± 0.005 | 0.92 ± 0.003 | 0.92 ± 0.004 |
| | DGI | 0.89 ± 0.011 | 0.87 ± 0.014 | 0.89 ± 0.010 | 0.88 ± 0.014 |
| | FUSE | 0.92 ± 0.006 | 0.91 ± 0.008 | 0.92 ± 0.003 | 0.91 ± 0.003 |
| | Given | 0.94 ± 0.002 | 0.93 ± 0.004 | 0.94 ± 0.003 | 0.93 ± 0.003 |
| | Node2Vec | 0.93 ± 0.005 | 0.93 ± 0.007 | 0.92 ± 0.003 | 0.92 ± 0.003 |
| | Random | 0.92 ± 0.007 | 0.91 ± 0.009 | 0.91 ± 0.004 | 0.90 ± 0.003 |
| | VGAE | 0.92 ± 0.005 | 0.92 ± 0.008 | 0.92 ± 0.004 | 0.91 ± 0.003 |
| GCN | DeepWalk | 0.83 ± 0.043 | 0.72 ± 0.069 | 0.83 ± 0.042 | 0.74 ± 0.086 |
| | DGI | 0.18 ± 0.073 | 0.05 ± 0.029 | 0.21 ± 0.051 | 0.05 ± 0.009 |
| | FUSE | 0.91 ± 0.007 | 0.90 ± 0.009 | 0.90 ± 0.004 | 0.90 ± 0.005 |
| | Given | 0.18 ± 0.116 | 0.06 ± 0.067 | 0.15 ± 0.082 | 0.04 ± 0.015 |
| | Node2Vec | 0.79 ± 0.081 | 0.65 ± 0.124 | 0.78 ± 0.085 | 0.70 ± 0.139 |
| | Random | 0.86 ± 0.013 | 0.79 ± 0.044 | 0.84 ± 0.013 | 0.77 ± 0.043 |
| | VGAE | 0.86 ± 0.011 | 0.80 ± 0.036 | 0.86 ± 0.004 | 0.79 ± 0.025 |
| SAGE | DeepWalk | 0.92 ± 0.005 | 0.91 ± 0.005 | 0.91 ± 0.004 | 0.90 ± 0.006 |
| | DGI | 0.87 ± 0.013 | 0.85 ± 0.016 | 0.87 ± 0.020 | 0.84 ± 0.040 |
| | FUSE | 0.90 ± 0.003 | 0.89 ± 0.008 | 0.89 ± 0.005 | 0.87 ± 0.005 |
| | Given | 0.95 ± 0.006 | 0.93 ± 0.011 | 0.94 ± 0.003 | 0.93 ± 0.005 |
| | Node2Vec | 0.92 ± 0.005 | 0.91 ± 0.008 | 0.91 ± 0.005 | 0.90 ± 0.006 |
| | Random | 0.89 ± 0.004 | 0.88 ± 0.008 | 0.83 ± 0.010 | 0.80 ± 0.011 |
| | VGAE | 0.91 ± 0.007 | 0.90 ± 0.009 | 0.91 ± 0.006 | 0.90 ± 0.006 |

Table 24: PubMed – Accuracy and F1 for 70–30 and 30–70 Splits

| Classifier | Embedding | 70–30 Split | | 30–70 Split | |
| --- | --- | --- | --- | --- | --- |
| | | Accuracy | F1 | Accuracy | F1 |
| GAT | DeepWalk | 0.84 ± 0.003 | 0.82 ± 0.004 | 0.82 ± 0.004 | 0.81 ± 0.004 |
| | DGI | 0.55 ± 0.092 | 0.46 ± 0.124 | 0.58 ± 0.087 | 0.52 ± 0.132 |
| | FUSE | 0.81 ± 0.008 | 0.80 ± 0.008 | 0.80 ± 0.004 | 0.79 ± 0.004 |
| | Given | 0.88 ± 0.003 | 0.87 ± 0.003 | 0.87 ± 0.002 | 0.86 ± 0.002 |
| | Node2Vec | 0.83 ± 0.004 | 0.82 ± 0.004 | 0.82 ± 0.003 | 0.81 ± 0.004 |
| | random | 0.81 ± 0.004 | 0.80 ± 0.005 | 0.78 ± 0.005 | 0.77 ± 0.005 |
| | VGAE | 0.83 ± 0.005 | 0.82 ± 0.005 | 0.82 ± 0.002 | 0.80 ± 0.002 |
| GCN | DeepWalk | 0.80 ± 0.011 | 0.78 ± 0.012 | 0.79 ± 0.001 | 0.77 ± 0.002 |
| | DGI | 0.44 ± 0.080 | 0.26 ± 0.119 | 0.46 ± 0.092 | 0.31 ± 0.146 |
| | FUSE | 0.81 ± 0.006 | 0.80 ± 0.007 | 0.80 ± 0.002 | 0.79 ± 0.003 |
| | Given | 0.84 ± 0.003 | 0.83 ± 0.003 | 0.83 ± 0.004 | 0.82 ± 0.004 |
| | Node2Vec | 0.79 ± 0.007 | 0.78 ± 0.010 | 0.79 ± 0.005 | 0.78 ± 0.005 |
| | random | 0.72 ± 0.003 | 0.70 ± 0.004 | 0.69 ± 0.003 | 0.67 ± 0.003 |
| | VGAE | 0.80 ± 0.006 | 0.79 ± 0.007 | 0.79 ± 0.002 | 0.78 ± 0.002 |
| SAGE | DeepWalk | 0.83 ± 0.004 | 0.82 ± 0.004 | 0.81 ± 0.003 | 0.80 ± 0.004 |
| | DGI | 0.54 ± 0.070 | 0.46 ± 0.110 | 0.53 ± 0.039 | 0.44 ± 0.077 |
| | FUSE | 0.80 ± 0.005 | 0.79 ± 0.007 | 0.78 ± 0.004 | 0.76 ± 0.005 |
| | Given | 0.88 ± 0.004 | 0.87 ± 0.004 | 0.86 ± 0.008 | 0.85 ± 0.008 |
| | Node2Vec | 0.83 ± 0.004 | 0.81 ± 0.004 | 0.81 ± 0.005 | 0.80 ± 0.005 |
| | Random | 0.65 ± 0.009 | 0.62 ± 0.007 | 0.55 ± 0.013 | 0.51 ± 0.015 |
| | VGAE | 0.82 ± 0.006 | 0.81 ± 0.006 | 0.80 ± 0.004 | 0.78 ± 0.004 |

Table 25: WikiCS – Accuracy and F1 for 70–30 and 30–70 Splits

| Classifier | Embedding | 70–30 Split | | 30–70 Split | |
|---|---|---|---|---|---|
| | | **Accuracy** | **F1** | **Accuracy** | **F1** |
| **GAT** | DeepWalk | $0.82 \pm 0.002$ | $0.80 \pm 0.004$ | $0.81 \pm 0.002$ | $0.78 \pm 0.003$ |
| | DGI | $0.76 \pm 0.006$ | $0.72 \pm 0.005$ | $0.75 \pm 0.011$ | $0.70 \pm 0.016$ |
| | FUSE | $0.81 \pm 0.003$ | $0.79 \pm 0.006$ | $0.80 \pm 0.002$ | $0.76 \pm 0.005$ |
| | Given | $0.84 \pm 0.003$ | $0.82 \pm 0.002$ | $0.83 \pm 0.004$ | $0.81 \pm 0.005$ |
| | Node2Vec | $0.82 \pm 0.002$ | $0.80 \pm 0.005$ | $0.81 \pm 0.002$ | $0.78 \pm 0.003$ |
| | Random | $0.80 \pm 0.003$ | $0.77 \pm 0.003$ | $0.78 \pm 0.005$ | $0.75 \pm 0.007$ |
| | VGAE | $0.80 \pm 0.005$ | $0.77 \pm 0.007$ | $0.80 \pm 0.001$ | $0.77 \pm 0.003$ |
| **GCN** | DeepWalk | $0.67 \pm 0.092$ | $0.55 \pm 0.123$ | $0.66 \pm 0.073$ | $0.54 \pm 0.089$ |
| | DGI | $0.19 \pm 0.067$ | $0.04 \pm 0.009$ | $0.18 \pm 0.075$ | $0.03 \pm 0.012$ |
| | FUSE | $0.77 \pm 0.007$ | $0.74 \pm 0.008$ | $0.76 \pm 0.004$ | $0.73 \pm 0.003$ |
| | Given | $0.44 \pm 0.132$ | $0.25 \pm 0.141$ | $0.39 \pm 0.147$ | $0.22 \pm 0.150$ |
| | Node2Vec | $0.69 \pm 0.058$ | $0.60 \pm 0.069$ | $0.64 \pm 0.069$ | $0.50 \pm 0.056$ |
| | Random | $0.74 \pm 0.009$ | $0.68 \pm 0.031$ | $0.72 \pm 0.011$ | $0.65 \pm 0.037$ |
| | VGAE | $0.73 \pm 0.039$ | $0.69 \pm 0.034$ | $0.71 \pm 0.014$ | $0.62 \pm 0.027$ |
| **SAGE** | DeepWalk | $0.81 \pm 0.005$ | $0.78 \pm 0.007$ | $0.79 \pm 0.006$ | $0.75 \pm 0.007$ |
| | DGI | $0.69 \pm 0.020$ | $0.56 \pm 0.062$ | $0.69 \pm 0.021$ | $0.58 \pm 0.054$ |
| | FUSE | $0.78 \pm 0.006$ | $0.74 \pm 0.010$ | $0.74 \pm 0.008$ | $0.70 \pm 0.010$ |
| | Given | $0.84 \pm 0.005$ | $0.82 \pm 0.008$ | $0.83 \pm 0.003$ | $0.80 \pm 0.004$ |
| | Node2Vec | $0.81 \pm 0.005$ | $0.77 \pm 0.007$ | $0.79 \pm 0.008$ | $0.75 \pm 0.010$ |
| | Random | $0.76 \pm 0.005$ | $0.73 \pm 0.008$ | $0.68 \pm 0.009$ | $0.63 \pm 0.008$ |
| | VGAE | $0.80 \pm 0.002$ | $0.76 \pm 0.002$ | $0.79 \pm 0.003$ | $0.76 \pm 0.004$ |

Table 26: Clustering results for Cora (70–30)

| Classifier | Embedding | DB | ARI | V-Measure |
|---|---|---|---|---|
| **GAT** | DeepWalk | $1.814 \pm 0.069$ | $0.535 \pm 0.035$ | $0.587 \pm 0.017$ |
| | DGI | $1.946 \pm 0.585$ | $0.075 \pm 0.037$ | $0.166 \pm 0.055$ |
| | FUSE | $0.979 \pm 0.140$ | $0.810 \pm 0.015$ | $0.775 \pm 0.019$ |
| | Given | $1.153 \pm 0.037$ | $0.781 \pm 0.016$ | $0.748 \pm 0.013$ |
| | Node2Vec | $1.905 \pm 0.083$ | $0.565 \pm 0.049$ | $0.591 \pm 0.021$ |
| | Random | $4.278 \pm 0.405$ | $0.175 \pm 0.062$ | $0.240 \pm 0.061$ |
| | VGAE | $2.043 \pm 0.134$ | $0.377 \pm 0.055$ | $0.491 \pm 0.037$ |
| **GCN** | DeepWalk | $1.126 \pm 0.124$ | $0.062 \pm 0.020$ | $0.239 \pm 0.011$ |
| | DGI | $0.819 \pm 0.321$ | $0.007 \pm 0.009$ | $0.036 \pm 0.049$ |
| | FUSE | $1.266 \pm 0.033$ | $0.392 \pm 0.044$ | $0.476 \pm 0.022$ |
| | Given | $1.142 \pm 0.110$ | $0.106 \pm 0.025$ | $0.254 \pm 0.050$ |
| | Node2Vec | $1.193 \pm 0.044$ | $0.065 \pm 0.019$ | $0.224 \pm 0.015$ |
| | Random | $2.114 \pm 0.110$ | $0.013 \pm 0.007$ | $0.043 \pm 0.017$ |
| | VGAE | $1.257 \pm 0.172$ | $0.073 \pm 0.020$ | $0.214 \pm 0.014$ |
| **SAGE** | DeepWalk | $1.168 \pm 0.149$ | $0.412 \pm 0.028$ | $0.537 \pm 0.027$ |
| | DGI | $1.514 \pm 0.315$ | $0.068 \pm 0.036$ | $0.126 \pm 0.065$ |
| | FUSE | $0.540 \pm 0.044$ | $0.886 \pm 0.007$ | $0.854 \pm 0.007$ |
| | Given | $0.860 \pm 0.031$ | $0.864 \pm 0.008$ | $0.823 \pm 0.006$ |
| | Node2Vec | $1.275 \pm 0.109$ | $0.453 \pm 0.054$ | $0.560 \pm 0.011$ |
| | Random | $1.980 \pm 0.033$ | $0.006 \pm 0.007$ | $0.026 \pm 0.007$ |
| | VGAE | $1.269 \pm 0.139$ | $0.287 \pm 0.045$ | $0.471 \pm 0.022$ |
| **Raw** | DeepWalk | $3.035 \pm 0.026$ | $0.350 \pm 0.005$ | $0.440 \pm 0.005$ |
| | DGI | $2.074 \pm 1.133$ | $-0.001 \pm 0.002$ | $0.013 \pm 0.001$ |
| | FUSE | $5.068 \pm 3.156$ | $0.050 \pm 0.080$ | $0.082 \pm 0.118$ |
| | Given | $6.380 \pm 0.231$ | $0.081 \pm 0.013$ | $0.142 \pm 0.026$ |
| | Node2Vec | $3.337 \pm 0.043$ | $0.298 \pm 0.040$ | $0.410 \pm 0.026$ |
| | Random | $8.593 \pm 0.082$ | $-0.000 \pm 0.001$ | $0.004 \pm 0.001$ |
| | VGAE | $3.725 \pm 0.208$ | $0.202 \pm 0.029$ | $0.353 \pm 0.019$ |

Table 27: Clustering results for Cora (30-70)

| Classifier | Embedding | DB | ARI | V-Measure |
|---|---|---|---|---|
| **GAT** | DeepWalk | $1.557 \pm 0.079$ | $0.567 \pm 0.043$ | $0.586 \pm 0.016$ |
| | DGI | $1.673 \pm 1.013$ | $0.055 \pm 0.054$ | $0.116 \pm 0.082$ |
| | FUSE | $0.872 \pm 0.047$ | $0.670 \pm 0.015$ | $0.647 \pm 0.014$ |
| | Given | $1.148 \pm 0.042$ | $0.689 \pm 0.017$ | $0.673 \pm 0.009$ |
| | Node2Vec | $1.580 \pm 0.054$ | $0.592 \pm 0.023$ | $0.588 \pm 0.015$ |
| | Random | $4.349 \pm 0.181$ | $0.054 \pm 0.007$ | $0.093 \pm 0.007$ |
| | VGAE | $1.816 \pm 0.112$ | $0.417 \pm 0.072$ | $0.488 \pm 0.032$ |
| **GCN** | DeepWalk | $1.159 \pm 0.066$ | $0.041 \pm 0.024$ | $0.147 \pm 0.049$ |
| | DGI | $0.455 \pm 0.256$ | $0.011 \pm 0.022$ | $0.027 \pm 0.034$ |
| | FUSE | $1.104 \pm 0.065$ | $0.278 \pm 0.083$ | $0.414 \pm 0.027$ |
| | Given | $1.176 \pm 0.083$ | $0.041 \pm 0.011$ | $0.107 \pm 0.025$ |
| | Node2Vec | $1.212 \pm 0.066$ | $0.055 \pm 0.026$ | $0.162 \pm 0.035$ |
| | Random | $2.108 \pm 0.124$ | $0.022 \pm 0.004$ | $0.034 \pm 0.007$ |
| | VGAE | $1.205 \pm 0.096$ | $0.071 \pm 0.024$ | $0.220 \pm 0.015$ |
| **SAGE** | DeepWalk | $1.177 \pm 0.232$ | $0.399 \pm 0.044$ | $0.527 \pm 0.019$ |
| | DGI | $1.315 \pm 0.500$ | $0.040 \pm 0.031$ | $0.087 \pm 0.060$ |
| | FUSE | $0.689 \pm 0.058$ | $0.673 \pm 0.014$ | $0.647 \pm 0.009$ |
| | Given | $1.105 \pm 0.049$ | $0.676 \pm 0.010$ | $0.638 \pm 0.009$ |
| | Node2Vec | $1.181 \pm 0.119$ | $0.407 \pm 0.035$ | $0.526 \pm 0.015$ |
| | Random | $2.073 \pm 0.077$ | $0.001 \pm 0.000$ | $0.011 \pm 0.002$ |
| | VGAE | $1.262 \pm 0.067$ | $0.270 \pm 0.025$ | $0.446 \pm 0.027$ |
| **Raw** | DeepWalk | $3.035 \pm 0.026$ | $0.350 \pm 0.005$ | $0.440 \pm 0.005$ |
| | DGI | $2.060 \pm 1.165$ | $0.002 \pm 0.003$ | $0.015 \pm 0.002$ |
| | FUSE | $5.045 \pm 0.749$ | $0.223 \pm 0.159$ | $0.289 \pm 0.166$ |
| | Given | $6.380 \pm 0.231$ | $0.081 \pm 0.013$ | $0.142 \pm 0.026$ |
| | Node2Vec | $3.337 \pm 0.043$ | $0.298 \pm 0.040$ | $0.410 \pm 0.026$ |
| | Random | $8.593 \pm 0.082$ | $-0.000 \pm 0.001$ | $0.004 \pm 0.001$ |
| | VGAE | $3.862 \pm 0.249$ | $0.222 \pm 0.043$ | $0.350 \pm 0.026$ |

Table 28: Clustering results for CiteSeer (70-30)

| Classifier | Embedding | DB | ARI | V-Measure |
|---|---|---|---|---|
| **GAT** | DeepWalk | $2.368 \pm 0.115$ | $0.201 \pm 0.018$ | $0.290 \pm 0.011$ |
| | DGI | $1.232 \pm 1.053$ | $0.028 \pm 0.005$ | $0.043 \pm 0.007$ |
| | FUSE | $1.179 \pm 0.032$ | $0.627 \pm 0.016$ | $0.592 \pm 0.015$ |
| | Given | $1.407 \pm 0.020$ | $0.646 \pm 0.007$ | $0.611 \pm 0.008$ |
| | Node2Vec | $2.521 \pm 0.208$ | $0.159 \pm 0.016$ | $0.269 \pm 0.006$ |
| | Random | $5.178 \pm 0.196$ | $0.060 \pm 0.011$ | $0.088 \pm 0.006$ |
| | VGAE | $2.927 \pm 0.233$ | $0.108 \pm 0.008$ | $0.197 \pm 0.020$ |
| **GCN** | DeepWalk | $1.105 \pm 0.126$ | $0.010 \pm 0.006$ | $0.104 \pm 0.004$ |
| | DGI | $0.273 \pm 0.473$ | $0.002 \pm 0.004$ | $0.011 \pm 0.021$ |
| | FUSE | $1.492 \pm 0.038$ | $0.235 \pm 0.042$ | $0.324 \pm 0.013$ |
| | Given | $1.336 \pm 0.056$ | $0.035 \pm 0.004$ | $0.179 \pm 0.016$ |
| | Node2Vec | $1.240 \pm 0.174$ | $0.010 \pm 0.002$ | $0.110 \pm 0.017$ |
| | Random | $2.098 \pm 0.049$ | $0.002 \pm 0.003$ | $0.017 \pm 0.006$ |
| | VGAE | $1.685 \pm 0.125$ | $0.004 \pm 0.003$ | $0.071 \pm 0.011$ |
| **SAGE** | DeepWalk | $1.533 \pm 0.153$ | $0.158 \pm 0.023$ | $0.254 \pm 0.038$ |
| | DGI | $1.041 \pm 0.503$ | $0.011 \pm 0.012$ | $0.031 \pm 0.019$ |
| | FUSE | $0.623 \pm 0.023$ | $0.786 \pm 0.018$ | $0.742 \pm 0.017$ |
| | Given | $1.182 \pm 0.070$ | $0.759 \pm 0.021$ | $0.724 \pm 0.013$ |
| | Node2Vec | $1.499 \pm 0.141$ | $0.129 \pm 0.024$ | $0.240 \pm 0.026$ |
| | Random | $2.165 \pm 0.042$ | $0.002 \pm 0.002$ | $0.008 \pm 0.001$ |
| | VGAE | $1.714 \pm 0.063$ | $0.065 \pm 0.022$ | $0.164 \pm 0.017$ |
| **Raw** | DeepWalk | $3.707 \pm 0.459$ | $0.111 \pm 0.014$ | $0.217 \pm 0.020$ |
| | DGI | $1.396 \pm 0.852$ | $0.014 \pm 0.003$ | $0.024 \pm 0.008$ |
| | FUSE | $6.872 \pm 1.213$ | $0.596 \pm 0.331$ | $0.564 \pm 0.310$ |
| | Given | $8.539 \pm 0.133$ | $0.177 \pm 0.041$ | $0.220 \pm 0.050$ |
| | Node2Vec | $4.221 \pm 0.125$ | $0.094 \pm 0.012$ | $0.193 \pm 0.008$ |
| | Random | $9.140 \pm 0.043$ | $0.000 \pm 0.001$ | $0.003 \pm 0.001$ |
| | VGAE | $4.606 \pm 0.056$ | $0.054 \pm 0.009$ | $0.132 \pm 0.017$ |

Table 29: Clustering results for CiteSeer (30-70)

| Classifier | Embedding | DB | ARI | V-Measure |
|---|---|---|---|---|
| **GAT** | DeepWalk | $2.219 \pm 0.116$ | $0.213 \pm 0.023$ | $0.296 \pm 0.015$ |
| | DGI | $1.279 \pm 0.964$ | $0.028 \pm 0.011$ | $0.045 \pm 0.008$ |
| | FUSE | $1.131 \pm 0.059$ | $0.445 \pm 0.013$ | $0.416 \pm 0.011$ |
| | Given | $1.382 \pm 0.062$ | $0.536 \pm 0.017$ | $0.522 \pm 0.012$ |
| | Node2Vec | $2.413 \pm 0.044$ | $0.185 \pm 0.018$ | $0.268 \pm 0.012$ |
| | Random | $4.963 \pm 0.286$ | $0.025 \pm 0.007$ | $0.041 \pm 0.010$ |
| | VGAE | $2.635 \pm 0.193$ | $0.091 \pm 0.016$ | $0.173 \pm 0.011$ |
| **GCN** | DeepWalk | $1.038 \pm 0.082$ | $0.004 \pm 0.001$ | $0.068 \pm 0.013$ |
| | DGI | $0.344 \pm 0.595$ | $0.000 \pm 0.000$ | $0.006 \pm 0.010$ |
| | FUSE | $1.164 \pm 0.097$ | $0.128 \pm 0.016$ | $0.242 \pm 0.012$ |
| | Given | $1.141 \pm 0.079$ | $0.001 \pm 0.002$ | $0.043 \pm 0.017$ |
| | Node2Vec | $1.093 \pm 0.146$ | $0.007 \pm 0.006$ | $0.074 \pm 0.014$ |
| | Random | $2.247 \pm 0.109$ | $-0.000 \pm 0.001$ | $0.011 \pm 0.003$ |
| | VGAE | $1.461 \pm 0.059$ | $0.004 \pm 0.004$ | $0.100 \pm 0.017$ |
| **SAGE** | DeepWalk | $1.556 \pm 0.065$ | $0.138 \pm 0.027$ | $0.232 \pm 0.039$ |
| | DGI | $1.164 \pm 0.580$ | $0.011 \pm 0.009$ | $0.026 \pm 0.008$ |
| | FUSE | $0.667 \pm 0.028$ | $0.475 \pm 0.010$ | $0.442 \pm 0.011$ |
| | Given | $1.366 \pm 0.068$ | $0.532 \pm 0.013$ | $0.512 \pm 0.012$ |
| | Node2Vec | $1.662 \pm 0.115$ | $0.109 \pm 0.021$ | $0.198 \pm 0.037$ |
| | Random | $2.252 \pm 0.065$ | $0.001 \pm 0.001$ | $0.005 \pm 0.001$ |
| | VGAE | $1.761 \pm 0.166$ | $0.063 \pm 0.013$ | $0.132 \pm 0.018$ |
| **Raw** | DeepWalk | $3.707 \pm 0.459$ | $0.111 \pm 0.014$ | $0.217 \pm 0.020$ |
| | DGI | $1.291 \pm 0.793$ | $0.016 \pm 0.005$ | $0.025 \pm 0.005$ |
| | FUSE | $6.725 \pm 0.984$ | $0.106 \pm 0.118$ | $0.150 \pm 0.126$ |
| | Given | $8.539 \pm 0.133$ | $0.177 \pm 0.041$ | $0.220 \pm 0.050$ |
| | Node2Vec | $4.221 \pm 0.125$ | $0.094 \pm 0.012$ | $0.193 \pm 0.008$ |
| | Random | $9.140 \pm 0.043$ | $0.000 \pm 0.001$ | $0.003 \pm 0.001$ |
| | VGAE | $4.610 \pm 0.091$ | $0.049 \pm 0.008$ | $0.123 \pm 0.013$ |

Table 30: Clustering results for PubMed (70-30)

| Classifier | Embedding | DB | ARI | V-Measure |
|---|---|---|---|---|
| **GAT** | DeepWalk | $1.713 \pm 0.048$ | $0.414 \pm 0.019$ | $0.389 \pm 0.010$ |
| | DGI | $1.452 \pm 0.297$ | $0.008 \pm 0.003$ | $0.010 \pm 0.004$ |
| | FUSE | $0.978 \pm 0.056$ | $0.492 \pm 0.042$ | $0.416 \pm 0.035$ |
| | Given | $1.098 \pm 0.033$ | $0.521 \pm 0.019$ | $0.511 \pm 0.011$ |
| | Node2Vec | $1.736 \pm 0.030$ | $0.390 \pm 0.041$ | $0.372 \pm 0.026$ |
| | Random | $6.159 \pm 0.278$ | $0.001 \pm 0.001$ | $0.002 \pm 0.001$ |
| | VGAE | $2.512 \pm 0.142$ | $0.198 \pm 0.061$ | $0.265 \pm 0.049$ |
| **GCN** | DeepWalk | $0.871 \pm 0.069$ | $0.003 \pm 0.014$ | $0.047 \pm 0.017$ |
| | DGI | $0.087 \pm 0.150$ | $-0.000 \pm 0.000$ | $0.000 \pm 0.000$ |
| | FUSE | $1.581 \pm 0.109$ | $0.051 \pm 0.016$ | $0.109 \pm 0.026$ |
| | Given | $1.351 \pm 0.203$ | $0.028 \pm 0.077$ | $0.040 \pm 0.065$ |
| | Node2Vec | $0.964 \pm 0.154$ | $0.002 \pm 0.011$ | $0.046 \pm 0.013$ |
| | Random | $2.274 \pm 0.031$ | $-0.006 \pm 0.002$ | $0.002 \pm 0.001$ |
| | VGAE | $1.697 \pm 0.269$ | $-0.002 \pm 0.004$ | $0.017 \pm 0.006$ |
| **SAGE** | DeepWalk | $1.397 \pm 0.080$ | $0.453 \pm 0.043$ | $0.400 \pm 0.026$ |
| | DGI | $1.052 \pm 0.415$ | $0.009 \pm 0.010$ | $0.009 \pm 0.011$ |
| | FUSE | $1.348 \pm 0.124$ | $0.415 \pm 0.041$ | $0.360 \pm 0.031$ |
| | Given | $1.221 \pm 0.067$ | $0.623 \pm 0.024$ | $0.558 \pm 0.025$ |
| | Node2Vec | $1.483 \pm 0.106$ | $0.457 \pm 0.034$ | $0.393 \pm 0.024$ |
| | Random | $2.753 \pm 0.109$ | $-0.001 \pm 0.002$ | $0.001 \pm 0.001$ |
| | VGAE | $2.009 \pm 0.183$ | $0.260 \pm 0.095$ | $0.303 \pm 0.043$ |
| **Raw** | DeepWalk | $4.580 \pm 0.011$ | $0.304 \pm 0.001$ | $0.296 \pm 0.001$ |
| | DGI | $1.167 \pm 0.441$ | $0.007 \pm 0.002$ | $0.003 \pm 0.001$ |
| | FUSE | $12.849 \pm 0.128$ | $0.062 \pm 0.046$ | $0.050 \pm 0.037$ |
| | Given | $5.161 \pm 0.009$ | $0.280 \pm 0.001$ | $0.312 \pm 0.001$ |
| | Node2Vec | $4.885 \pm 0.027$ | $0.279 \pm 0.002$ | $0.288 \pm 0.003$ |
| | Random | $12.406 \pm 0.015$ | $0.000 \pm 0.000$ | $0.000 \pm 0.000$ |
| | VGAE | $3.900 \pm 0.256$ | $0.029 \pm 0.022$ | $0.152 \pm 0.024$ |

Table 31: Clustering results for PubMed (30-70)

| Classifier | Embedding | DB | ARI | VMeasure |
|---|---|---|---|---|
| **GAT** | DeepWalk | $1.835 \pm 0.073$ | $0.402 \pm 0.048$ | $0.375 \pm 0.027$ |
| | DGI | $1.556 \pm 0.382$ | $0.006 \pm 0.004$ | $0.007 \pm 0.004$ |
| | FUSE | $0.796 \pm 0.013$ | $0.518 \pm 0.005$ | $0.433 \pm 0.005$ |
| | Given | $1.060 \pm 0.022$ | $0.470 \pm 0.012$ | $0.466 \pm 0.005$ |
| | Node2Vec | $1.909 \pm 0.049$ | $0.390 \pm 0.050$ | $0.368 \pm 0.031$ |
| | Random | $6.677 \pm 0.152$ | $0.001 \pm 0.000$ | $0.001 \pm 0.001$ |
| | VGAE | $2.610 \pm 0.104$ | $0.216 \pm 0.023$ | $0.271 \pm 0.014$ |
| **GCN** | DeepWalk | $1.183 \pm 0.121$ | $0.007 \pm 0.003$ | $0.038 \pm 0.011$ |
| | DGI | $0.821 \pm 0.203$ | $-0.004 \pm 0.002$ | $0.001 \pm 0.001$ |
| | FUSE | $1.205 \pm 0.093$ | $0.026 \pm 0.032$ | $0.100 \pm 0.009$ |
| | Given | $1.241 \pm 0.124$ | $-0.008 \pm 0.008$ | $0.037 \pm 0.010$ |
| | Node2Vec | $1.245 \pm 0.212$ | $0.004 \pm 0.008$ | $0.034 \pm 0.005$ |
| | Random | $2.389 \pm 0.099$ | $-0.005 \pm 0.001$ | $0.001 \pm 0.001$ |
| | VGAE | $1.361 \pm 0.399$ | $-0.005 \pm 0.007$ | $0.025 \pm 0.008$ |
| **SAGE** | DeepWalk | $1.565 \pm 0.185$ | $0.443 \pm 0.038$ | $0.386 \pm 0.013$ |
| | DGI | $1.310 \pm 0.428$ | $0.011 \pm 0.012$ | $0.008 \pm 0.007$ |
| | FUSE | $1.377 \pm 0.204$ | $0.311 \pm 0.112$ | $0.300 \pm 0.037$ |
| | Given | $1.358 \pm 0.062$ | $0.561 \pm 0.036$ | $0.500 \pm 0.022$ |
| | Node2Vec | $1.659 \pm 0.125$ | $0.409 \pm 0.055$ | $0.365 \pm 0.022$ |
| | Random | $2.933 \pm 0.143$ | $0.000 \pm 0.001$ | $0.000 \pm 0.000$ |
| | VGAE | $2.264 \pm 0.160$ | $0.169 \pm 0.048$ | $0.241 \pm 0.048$ |
| **Raw** | DeepWalk | $4.580 \pm 0.011$ | $0.304 \pm 0.001$ | $0.296 \pm 0.001$ |
| | DGI | $1.291 \pm 0.380$ | $0.007 \pm 0.001$ | $0.003 \pm 0.001$ |
| | FUSE | $11.476 \pm 0.126$ | $0.400 \pm 0.017$ | $0.329 \pm 0.014$ |
| | Given | $5.161 \pm 0.009$ | $0.280 \pm 0.001$ | $0.312 \pm 0.001$ |
| | Node2Vec | $4.885 \pm 0.027$ | $0.279 \pm 0.002$ | $0.288 \pm 0.003$ |
| | Random | $12.406 \pm 0.015$ | $0.000 \pm 0.000$ | $0.000 \pm 0.000$ |
| | VGAE | $4.014 \pm 0.319$ | $0.034 \pm 0.016$ | $0.154 \pm 0.017$ |

Table 32: Clustering results for Photo (70-30)

| Classifier | Embedding | DB | ARI | V-Measure |
|---|---|---|---|---|
| **GAT** | DeepWalk | $1.085 \pm 0.029$ | $0.628 \pm 0.007$ | $0.740 \pm 0.005$ |
| | DGI | $1.538 \pm 0.106$ | $0.383 \pm 0.032$ | $0.476 \pm 0.027$ |
| | FUSE | $0.894 \pm 0.016$ | $0.873 \pm 0.017$ | $0.857 \pm 0.015$ |
| | Given | $1.065 \pm 0.081$ | $0.660 \pm 0.026$ | $0.720 \pm 0.017$ |
| | Node2Vec | $1.122 \pm 0.028$ | $0.635 \pm 0.014$ | $0.739 \pm 0.009$ |
| | Random | $3.233 \pm 0.180$ | $0.262 \pm 0.061$ | $0.402 \pm 0.044$ |
| | VGAE | $1.336 \pm 0.084$ | $0.571 \pm 0.031$ | $0.700 \pm 0.022$ |
| **GCN** | DeepWalk | $0.808 \pm 0.101$ | $0.030 \pm 0.015$ | $0.232 \pm 0.021$ |
| | DGI | $0.222 \pm 0.257$ | $-0.007 \pm 0.008$ | $0.022 \pm 0.029$ |
| | FUSE | $0.916 \pm 0.051$ | $0.162 \pm 0.018$ | $0.455 \pm 0.022$ |
| | Given | $0.017 \pm 0.024$ | $-0.001 \pm 0.001$ | $0.001 \pm 0.002$ |
| | Node2Vec | $0.671 \pm 0.059$ | $0.013 \pm 0.018$ | $0.231 \pm 0.025$ |
| | Random | $1.974 \pm 0.099$ | $0.021 \pm 0.011$ | $0.096 \pm 0.023$ |
| | VGAE | $0.967 \pm 0.148$ | $0.038 \pm 0.020$ | $0.253 \pm 0.014$ |
| **SAGE** | DeepWalk | $0.808 \pm 0.088$ | $0.699 \pm 0.060$ | $0.762 \pm 0.026$ |
| | DGI | $1.067 \pm 0.103$ | $0.528 \pm 0.034$ | $0.601 \pm 0.028$ |
| | FUSE | $0.597 \pm 0.098$ | $0.892 \pm 0.022$ | $0.881 \pm 0.019$ |
| | Given | $0.874 \pm 0.042$ | $0.773 \pm 0.012$ | $0.824 \pm 0.012$ |
| | Node2Vec | $0.906 \pm 0.054$ | $0.681 \pm 0.069$ | $0.746 \pm 0.026$ |
| | Random | $2.042 \pm 0.048$ | $0.024 \pm 0.008$ | $0.050 \pm 0.006$ |
| | VGAE | $0.760 \pm 0.127$ | $0.592 \pm 0.052$ | $0.715 \pm 0.019$ |
| **Raw** | DeepWalk | $2.400 \pm 0.033$ | $0.597 \pm 0.003$ | $0.690 \pm 0.002$ |
| | DGI | $1.569 \pm 0.100$ | $0.077 \pm 0.006$ | $0.075 \pm 0.007$ |
| | FUSE | $5.087 \pm 0.844$ | $0.364 \pm 0.192$ | $0.448 \pm 0.155$ |
| | Given | $4.887 \pm 0.062$ | $0.058 \pm 0.007$ | $0.140 \pm 0.019$ |
| | Node2Vec | $2.510 \pm 0.067$ | $0.579 \pm 0.036$ | $0.673 \pm 0.028$ |
| | Random | $9.147 \pm 0.034$ | $-0.000 \pm 0.000$ | $0.001 \pm 0.000$ |
| | VGAE | $1.943 \pm 0.068$ | $0.199 \pm 0.069$ | $0.468 \pm 0.048$ |

Table 33: Clustering results for Photo (30-70)

| Classifier | Embedding | DB | ARI | V-Measure |
|---|---|---|---|---|
| **GAT** | DeepWalk | $1.075 \pm 0.026$ | $0.612 \pm 0.013$ | $0.729 \pm 0.004$ |
| | DGI | $1.628 \pm 0.050$ | $0.336 \pm 0.034$ | $0.426 \pm 0.028$ |
| | FUSE | $0.875 \pm 0.025$ | $0.814 \pm 0.007$ | $0.802 \pm 0.003$ |
| | Given | $1.034 \pm 0.039$ | $0.636 \pm 0.032$ | $0.702 \pm 0.018$ |
| | Node2Vec | $1.097 \pm 0.049$ | $0.607 \pm 0.015$ | $0.719 \pm 0.010$ |
| | Random | $3.877 \pm 0.297$ | $0.116 \pm 0.028$ | $0.216 \pm 0.031$ |
| | VGAE | $1.332 \pm 0.085$ | $0.528 \pm 0.027$ | $0.678 \pm 0.027$ |
| **GCN** | DeepWalk | $0.781 \pm 0.025$ | $0.040 \pm 0.019$ | $0.290 \pm 0.014$ |
| | DGI | $0.547 \pm 0.505$ | $0.024 \pm 0.032$ | $0.063 \pm 0.072$ |
| | FUSE | $1.004 \pm 0.025$ | $0.130 \pm 0.013$ | $0.401 \pm 0.038$ |
| | Given | $0.000 \pm 0.000$ | $0.000 \pm 0.000$ | $0.000 \pm 0.000$ |
| | Node2Vec | $0.767 \pm 0.072$ | $0.037 \pm 0.024$ | $0.287 \pm 0.047$ |
| | Random | $2.078 \pm 0.070$ | $0.017 \pm 0.004$ | $0.072 \pm 0.009$ |
| | VGAE | $1.046 \pm 0.163$ | $0.038 \pm 0.009$ | $0.261 \pm 0.030$ |
| **SAGE** | DeepWalk | $0.848 \pm 0.082$ | $0.667 \pm 0.064$ | $0.738 \pm 0.026$ |
| | DGI | $1.196 \pm 0.137$ | $0.523 \pm 0.023$ | $0.592 \pm 0.015$ |
| | FUSE | $0.576 \pm 0.096$ | $0.803 \pm 0.022$ | $0.792 \pm 0.015$ |
| | Given | $0.928 \pm 0.021$ | $0.768 \pm 0.019$ | $0.806 \pm 0.009$ |
| | Node2Vec | $0.904 \pm 0.029$ | $0.634 \pm 0.057$ | $0.713 \pm 0.019$ |
| | Random | $1.974 \pm 0.041$ | $0.005 \pm 0.001$ | $0.011 \pm 0.003$ |
| | VGAE | $0.844 \pm 0.058$ | $0.631 \pm 0.083$ | $0.721 \pm 0.022$ |
| **Raw** | DeepWalk | $2.400 \pm 0.033$ | $0.597 \pm 0.003$ | $0.690 \pm 0.002$ |
| | DGI | $1.657 \pm 0.169$ | $0.063 \pm 0.007$ | $0.065 \pm 0.008$ |
| | FUSE | $4.317 \pm 0.437$ | $0.510 \pm 0.109$ | $0.638 \pm 0.046$ |
| | Given | $4.887 \pm 0.062$ | $0.058 \pm 0.007$ | $0.140 \pm 0.019$ |
| | Node2Vec | $2.510 \pm 0.067$ | $0.579 \pm 0.036$ | $0.673 \pm 0.028$ |
| | Random | $9.147 \pm 0.034$ | $-0.000 \pm 0.000$ | $0.001 \pm 0.000$ |
| | VGAE | $1.973 \pm 0.067$ | $0.204 \pm 0.049$ | $0.486 \pm 0.031$ |

Table 34: Clustering results for WikiCS (70-30)

| Classifier | Embedding | DB | ARI | V-Measure |
|---|---|---|---|---|
| **GAT** | DeepWalk | $1.963 \pm 0.051$ | $0.458 \pm 0.015$ | $0.519 \pm 0.010$ |
| | DGI | $1.601 \pm 0.065$ | $0.098 \pm 0.012$ | $0.174 \pm 0.019$ |
| | FUSE | $1.342 \pm 0.110$ | $0.688 \pm 0.039$ | $0.674 \pm 0.020$ |
| | Given | $1.663 \pm 0.050$ | $0.510 \pm 0.036$ | $0.557 \pm 0.018$ |
| | Node2Vec | $1.926 \pm 0.091$ | $0.449 \pm 0.034$ | $0.521 \pm 0.007$ |
| | Random | $3.792 \pm 0.179$ | $0.120 \pm 0.015$ | $0.214 \pm 0.016$ |
| | VGAE | $1.778 \pm 0.052$ | $0.333 \pm 0.019$ | $0.443 \pm 0.018$ |
| **GCN** | DeepWalk | $0.477 \pm 0.065$ | $-0.003 \pm 0.006$ | $0.079 \pm 0.017$ |
| | DGI | $0.238 \pm 0.207$ | $0.001 \pm 0.002$ | $0.006 \pm 0.008$ |
| | FUSE | $1.140 \pm 0.107$ | $0.055 \pm 0.014$ | $0.222 \pm 0.015$ |
| | Given | $0.371 \pm 0.061$ | $-0.002 \pm 0.005$ | $0.046 \pm 0.015$ |
| | Node2Vec | $0.541 \pm 0.058$ | $0.003 \pm 0.005$ | $0.066 \pm 0.035$ |
| | Random | $1.927 \pm 0.080$ | $0.002 \pm 0.011$ | $0.041 \pm 0.005$ |
| | VGAE | $0.939 \pm 0.086$ | $0.007 \pm 0.002$ | $0.086 \pm 0.022$ |
| **SAGE** | DeepWalk | $1.013 \pm 0.070$ | $0.512 \pm 0.039$ | $0.563 \pm 0.008$ |
| | DGI | $1.348 \pm 0.192$ | $0.176 \pm 0.046$ | $0.279 \pm 0.048$ |
| | FUSE | $0.782 \pm 0.099$ | $0.476 \pm 0.046$ | $0.653 \pm 0.014$ |
| | Given | $1.002 \pm 0.048$ | $0.541 \pm 0.022$ | $0.631 \pm 0.005$ |
| | Node2Vec | $0.986 \pm 0.176$ | $0.511 \pm 0.096$ | $0.569 \pm 0.013$ |
| | Random | $1.872 \pm 0.064$ | $0.021 \pm 0.004$ | $0.049 \pm 0.004$ |
| | VGAE | $0.983 \pm 0.034$ | $0.464 \pm 0.051$ | $0.525 \pm 0.011$ |
| **Raw** | DeepWalk | $3.228 \pm 0.152$ | $0.359 \pm 0.038$ | $0.452 \pm 0.015$ |
| | DGI | $2.716 \pm 0.170$ | $0.030 \pm 0.003$ | $0.046 \pm 0.002$ |
| | FUSE | $5.185 \pm 0.306$ | $0.131 \pm 0.074$ | $0.164 \pm 0.088$ |
| | Given | $2.639 \pm 0.025$ | $0.145 \pm 0.003$ | $0.252 \pm 0.003$ |
| | Node2Vec | $3.240 \pm 0.113$ | $0.345 \pm 0.017$ | $0.448 \pm 0.008$ |
| | Random | $8.884 \pm 0.023$ | $-0.000 \pm 0.000$ | $0.002 \pm 0.001$ |
| | VGAE | $1.896 \pm 0.047$ | $0.144 \pm 0.011$ | $0.332 \pm 0.015$ |

Table 35: Clustering results for WikiCS (30-70)

| Classifier | Embedding | DB | ARI | V-Measure |
|---|---|---|---|---|
| **GAT** | DeepWalk | $1.988 \pm 0.076$ | $0.419 \pm 0.057$ | $0.492 \pm 0.025$ |
| | DGI | $1.725 \pm 0.190$ | $0.088 \pm 0.008$ | $0.164 \pm 0.018$ |
| | FUSE | $1.523 \pm 0.139$ | $0.581 \pm 0.015$ | $0.580 \pm 0.008$ |
| | Given | $1.705 \pm 0.125$ | $0.453 \pm 0.047$ | $0.518 \pm 0.017$ |
| | Node2Vec | $2.107 \pm 0.060$ | $0.409 \pm 0.040$ | $0.487 \pm 0.012$ |
| | Random | $4.328 \pm 0.132$ | $0.048 \pm 0.012$ | $0.112 \pm 0.016$ |
| | VGAE | $1.831 \pm 0.049$ | $0.295 \pm 0.022$ | $0.412 \pm 0.017$ |
| **GCN** | DeepWalk | $0.548 \pm 0.165$ | $-0.001 \pm 0.004$ | $0.132 \pm 0.021$ |
| | DGI | $0.000 \pm 0.000$ | $0.001 \pm 0.002$ | $0.003 \pm 0.004$ |
| | FUSE | $1.110 \pm 0.059$ | $0.021 \pm 0.022$ | $0.186 \pm 0.034$ |
| | Given | $0.419 \pm 0.035$ | $-0.003 \pm 0.017$ | $0.086 \pm 0.040$ |
| | Node2Vec | $0.602 \pm 0.116$ | $0.000 \pm 0.007$ | $0.127 \pm 0.012$ |
| | Random | $1.947 \pm 0.078$ | $0.006 \pm 0.007$ | $0.045 \pm 0.006$ |
| | VGAE | $0.869 \pm 0.048$ | $0.007 \pm 0.009$ | $0.110 \pm 0.019$ |
| **SAGE** | DeepWalk | $1.023 \pm 0.160$ | $0.348 \pm 0.051$ | $0.497 \pm 0.008$ |
| | DGI | $1.442 \pm 0.089$ | $0.193 \pm 0.051$ | $0.272 \pm 0.024$ |
| | FUSE | $0.824 \pm 0.139$ | $0.420 \pm 0.070$ | $0.537 \pm 0.026$ |
| | Given | $1.137 \pm 0.029$ | $0.520 \pm 0.047$ | $0.578 \pm 0.011$ |
| | Node2Vec | $1.012 \pm 0.213$ | $0.376 \pm 0.087$ | $0.511 \pm 0.023$ |
| | Random | $1.860 \pm 0.023$ | $0.002 \pm 0.003$ | $0.010 \pm 0.002$ |
| | VGAE | $1.116 \pm 0.107$ | $0.401 \pm 0.040$ | $0.494 \pm 0.016$ |
| **Raw** | DeepWalk | $3.228 \pm 0.152$ | $0.359 \pm 0.038$ | $0.452 \pm 0.015$ |
| | DGI | $2.608 \pm 0.126$ | $0.033 \pm 0.002$ | $0.047 \pm 0.002$ |
| | FUSE | $4.698 \pm 0.389$ | $0.336 \pm 0.067$ | $0.415 \pm 0.020$ |
| | Given | $2.639 \pm 0.025$ | $0.145 \pm 0.003$ | $0.252 \pm 0.003$ |
| | Node2Vec | $3.240 \pm 0.113$ | $0.345 \pm 0.017$ | $0.448 \pm 0.008$ |
| | Random | $8.884 \pm 0.023$ | $-0.000 \pm 0.000$ | $0.002 \pm 0.001$ |
| | VGAE | $1.937 \pm 0.064$ | $0.137 \pm 0.015$ | $0.323 \pm 0.016$ |

Table 36: Clustering results for ArXiV (70-30) (single seed)

| Classifier | Embedding | DB | ARI | V-Measure |
|---|---|---|---|---|
| **GAT** | DeepWalk | 2.429 | 0.228 | 0.448 |
| | DGI | 0.729 | -0.003 | 0.015 |
| | FUSE | 1.186 | 0.680 | 0.720 |
| | Given | 2.213 | 0.132 | 0.346 |
| | Node2Vec | 2.540 | 0.233 | 0.441 |
| | Random | 4.650 | 0.002 | 0.008 |
| | VGAE | 1.047 | -0.002 | 0.015 |
| **GCN** | DeepWalk | 1.251 | 0.058 | 0.280 |
| | DGI | 0.000 | 0.010 | 0.007 |
| | FUSE | 1.447 | 0.050 | 0.332 |
| | Given | 1.015 | 0.016 | 0.218 |
| | Node2Vec | 1.243 | 0.049 | 0.241 |
| | Random | 2.028 | -0.014 | 0.037 |
| | VGAE | 0.000 | 0.010 | 0.007 |
| **SAGE** | DeepWalk | 1.364 | 0.247 | 0.442 |
| | DGI | 0.900 | -0.003 | 0.006 |
| | FUSE | 0.861 | 0.791 | 0.793 |
| | Given | 1.571 | 0.189 | 0.358 |
| | Node2Vec | 1.404 | 0.280 | 0.434 |
| | Random | 1.877 | 0.000 | 0.003 |
| | VGAE | 1.346 | -0.000 | 0.006 |
| **Raw** | DeepWalk | 3.617 | 0.185 | 0.402 |
| | DGI | 1.107 | -0.003 | 0.008 |
| | FUSE | 2.432 | 0.732 | 0.776 |
| | Given | 3.495 | 0.070 | 0.221 |
| | Node2Vec | 3.751 | 0.179 | 0.385 |
| | Random | 7.524 | 0.000 | 0.001 |
| | VGAE | 2.219 | -0.002 | 0.007 |

Table 37: Clustering results for ArXiV (30-70) (single seed)

| Classifier | Embedding | DB | ARI | V-Measure |
|---|---|---|---|---|
| **GAT** | DeepWalk | 2.461 | 0.233 | 0.447 |
| | DGI | 0.924 | 0.002 | 0.010 |
| | FUSE | 1.692 | 0.515 | 0.576 |
| | Given | 2.199 | 0.132 | 0.344 |
| | Node2Vec | 2.528 | 0.234 | 0.441 |
| | Random | 4.860 | 0.002 | 0.007 |
| | VGAE | 1.093 | -0.003 | 0.013 |
| **GCN** | DeepWalk | 1.412 | 0.039 | 0.278 |
| | DGI | 0.000 | 0.000 | 0.000 |
| | FUSE | 0.948 | 0.037 | 0.320 |
| | Given | 1.270 | 0.006 | 0.245 |
| | Node2Vec | 1.342 | 0.015 | 0.274 |
| | Random | 2.035 | -0.019 | 0.033 |
| | VGAE | 0.000 | -0.000 | 0.000 |
| **SAGE** | DeepWalk | 1.305 | 0.213 | 0.435 |
| | DGI | 1.012 | -0.001 | 0.006 |
| | FUSE | 0.995 | 0.520 | 0.572 |
| | Given | 1.499 | 0.184 | 0.353 |
| | Node2Vec | 1.375 | 0.279 | 0.438 |
| | Random | 1.787 | 0.000 | 0.002 |
| | VGAE | 1.406 | 0.000 | 0.006 |
| **Raw** | DeepWalk | 3.617 | 0.185 | 0.402 |
| | DGI | 1.088 | -0.003 | 0.007 |
| | FUSE | 0.766 | 0.397 | 0.534 |
| | Given | 3.495 | 0.070 | 0.221 |
| | Node2Vec | 3.751 | 0.179 | 0.385 |
| | Random | 7.519 | -0.000 | 0.001 |
| | VGAE | 2.033 | -0.002 | 0.007 |

| Rates | Accuracy | | | F1 Score | | |
|---|---|---|---|---|---|---|
| | 0.2 | 0.5 | 0.8 | 0.2 | 0.5 | 0.8 |
| **GCN** | | | | | | |
| FUSE | **0.81** ±0.02 | **0.78** ±0.01 | **0.77** ±0.01 | **0.80** ±0.02 | **0.77** ±0.01 | **0.76** ±0.01 |
| Node2Vec | 0.78 ±0.01 | 0.76 ±0.01 | 0.68 ±0.04 | 0.76 ±0.02 | 0.75 ±0.01 | 0.67 ±0.04 |
| DeepWalk | 0.78 ±0.01 | 0.75 ±0.02 | 0.71 ±0.01 | 0.76 ±0.01 | 0.74 ±0.02 | 0.69 ±0.01 |
| VGAE | 0.78 ±0.01 | 0.74 ±0.02 | 0.66 ±0.02 | 0.76 ±0.01 | 0.72 ±0.02 | 0.65 ±0.02 |
| DGI | 0.32 ±0.05 | 0.36 ±0.07 | 0.32 ±0.05 | 0.10 ±0.08 | 0.16 ±0.12 | 0.15 ±0.10 |
| Random | 0.52 ±0.02 | 0.39 ±0.02 | 0.29 ±0.03 | 0.50 ±0.02 | 0.35 ±0.02 | 0.25 ±0.03 |
| **GAT** | | | | | | |
| FUSE | **0.84** ±0.02 | **0.82** ±0.01 | **0.77** ±0.02 | **0.83** ±0.02 | **0.81** ±0.01 | **0.75** ±0.02 |
| Node2Vec | 0.83 ±0.01 | 0.80 ±0.01 | 0.74 ±0.02 | 0.82 ±0.02 | 0.79 ±0.01 | 0.73 ±0.02 |
| DeepWalk | **0.84** ±0.02 | 0.80 ±0.02 | 0.74 ±0.02 | **0.83** ±0.02 | 0.78 ±0.02 | 0.73 ±0.02 |
| VGAE | 0.79 ±0.02 | 0.75 ±0.02 | 0.71 ±0.02 | 0.78 ±0.02 | 0.73 ±0.02 | 0.69 ±0.02 |
| DGI | 0.70 ±0.05 | 0.64 ±0.08 | 0.60 ±0.07 | 0.68 ±0.08 | 0.59 ±0.10 | 0.56 ±0.09 |
| Random | 0.66 ±0.02 | 0.50 ±0.03 | 0.33 ±0.04 | 0.63 ±0.03 | 0.47 ±0.03 | 0.27 ±0.04 |
| **SAGE** | | | | | | |
| FUSE | 0.85 ±0.02 | 0.82 ±0.01 | 0.76 ±0.01 | **0.84** ±0.02 | 0.81 ±0.01 | 0.74 ±0.01 |
| Node2Vec | 0.85 ±0.02 | **0.83** ±0.01 | 0.77 ±0.01 | **0.84** ±0.02 | 0.81 ±0.01 | **0.76** ±0.01 |
| DeepWalk | **0.86** ±0.01 | **0.83** ±0.01 | **0.78** ±0.01 | **0.84** ±0.01 | **0.82** ±0.01 | **0.76** ±0.01 |
| VGAE | 0.79 ±0.02 | 0.73 ±0.01 | 0.67 ±0.02 | 0.78 ±0.02 | 0.71 ±0.01 | 0.64 ±0.02 |
| DGI | 0.60 ±0.05 | 0.59 ±0.04 | 0.58 ±0.04 | 0.52 ±0.09 | 0.50 ±0.08 | 0.50 ±0.07 |
| Random | 0.51 ±0.02 | 0.35 ±0.02 | 0.26 ±0.02 | 0.46 ±0.03 | 0.26 ±0.02 | 0.17 ±0.02 |
| **MLP** | | | | | | |
| FUSE | 0.81 ±0.02 | 0.79 ±0.01 | 0.73 ±0.01 | 0.79 ±0.03 | 0.77 ±0.01 | 0.71 ±0.01 |
| Node2Vec | **0.84** ±0.01 | **0.82** ±0.01 | **0.76** ±0.01 | 0.83 ±0.01 | **0.81** ±0.01 | 0.74 ±0.02 |
| DeepWalk | 0.85 ±0.01 | 0.81 ±0.01 | 0.77 ±0.02 | **0.84** ±0.01 | 0.80 ±0.01 | **0.75** ±0.02 |
| VGAE | 0.65 ±0.02 | 0.63 ±0.02 | 0.62 ±0.01 | 0.63 ±0.02 | 0.61 ±0.01 | 0.60 ±0.02 |
| DGI | 0.53 ±0.05 | 0.49 ±0.07 | 0.48 ±0.06 | 0.44 ±0.09 | 0.35 ±0.13 | 0.36 ±0.11 |
| Random | 0.18 ±0.02 | 0.18 ±0.01 | 0.19 ±0.01 | 0.15 ±0.02 | 0.14 ±0.01 | 0.15 ±0.01 |

Table 38: Classification experiments on different masking rates for the MCAR scenario on the Cora dataset. The mean and standard deviation over 10 iterations are reported. The best and second-best in each metric, for each masking rate and each classifier, are highlighted in **bold** and underline respectively.

| | **Accuracy** | | | **F1 Score** | | |
|---|---|---|---|---|---|---|
| **Rates** | 0.2 | 0.5 | 0.8 | 0.2 | 0.5 | 0.8 |
| | | | **GCN** | | | |
| FUSE | **0.81** ± 0.01 | **0.78** ± 0.01 | **0.76** ± 0.02 | **0.80** ± 0.01 | **0.76** ± 0.01 | **0.75** ± 0.02 |
| Node2Vec | 0.79 ± 0.02 | 0.76 ± 0.01 | 0.68 ± 0.02 | 0.77 ± 0.02 | 0.75 ± 0.02 | 0.66 ± 0.02 |
| DeepWalk | 0.77 ± 0.02 | 0.77 ± 0.02 | 0.68 ± 0.02 | 0.76 ± 0.02 | **0.76** ± 0.02 | 0.66 ± 0.02 |
| VGAE | 0.77 ± 0.02 | 0.72 ± 0.02 | 0.66 ± 0.03 | 0.76 ± 0.02 | 0.72 ± 0.02 | 0.64 ± 0.03 |
| DGI | 0.28 ± 0.02 | 0.30 ± 0.03 | 0.36 ± 0.08 | 0.06 ± 0.00 | 0.08 ± 0.04 | 0.20 ± 0.14 |
| Random | 0.51 ± 0.02 | 0.40 ± 0.02 | 0.29 ± 0.03 | 0.48 ± 0.03 | 0.36 ± 0.02 | 0.24 ± 0.03 |
| | | | **GAT** | | | |
| FUSE | **0.85** ± 0.01 | **0.81** ± 0.01 | **0.77** ± 0.01 | **0.85** ± 0.01 | **0.80** ± 0.01 | **0.76** ± 0.02 |
| Node2Vec | 0.84 ± 0.01 | 0.80 ± 0.01 | 0.75 ± 0.02 | 0.83 ± 0.01 | 0.79 ± 0.01 | 0.73 ± 0.02 |
| DeepWalk | 0.83 ± 0.01 | 0.80 ± 0.01 | 0.75 ± 0.01 | 0.82 ± 0.01 | 0.79 ± 0.01 | 0.74 ± 0.01 |
| VGAE | 0.78 ± 0.01 | 0.75 ± 0.01 | 0.70 ± 0.01 | 0.77 ± 0.01 | 0.73 ± 0.01 | 0.68 ± 0.02 |
| DGI | 0.68 ± 0.05 | 0.68 ± 0.03 | 0.64 ± 0.03 | 0.64 ± 0.08 | 0.67 ± 0.03 | 0.62 ± 0.04 |
| Random | 0.65 ± 0.03 | 0.50 ± 0.03 | 0.34 ± 0.03 | 0.63 ± 0.03 | 0.46 ± 0.04 | 0.25 ± 0.04 |
| | | | **SAGE** | | | |
| FUSE | **0.85** ± 0.01 | 0.81 ± 0.01 | 0.76 ± 0.02 | **0.84** ± 0.01 | 0.80 ± 0.01 | 0.75 ± 0.02 |
| Node2Vec | **0.85** ± 0.01 | **0.83** ± 0.01 | **0.78** ± 0.01 | **0.84** ± 0.02 | **0.82** ± 0.01 | **0.77** ± 0.01 |
| DeepWalk | **0.85** ± 0.02 | **0.83** ± 0.01 | **0.78** ± 0.01 | 0.83 ± 0.02 | **0.82** ± 0.01 | **0.77** ± 0.02 |
| VGAE | 0.76 ± 0.02 | 0.72 ± 0.01 | 0.67 ± 0.01 | 0.74 ± 0.02 | 0.70 ± 0.01 | 0.63 ± 0.03 |
| DGI | 0.57 ± 0.07 | 0.59 ± 0.04 | 0.53 ± 0.06 | 0.47 ± 0.08 | 0.51 ± 0.06 | 0.42 ± 0.10 |
| Random | 0.49 ± 0.02 | 0.35 ± 0.02 | 0.27 ± 0.01 | 0.43 ± 0.03 | 0.27 ± 0.03 | 0.17 ± 0.01 |
| | | | **MLP** | | | |
| FUSE | 0.80 ± 0.02 | 0.77 ± 0.01 | 0.72 ± 0.02 | 0.79 ± 0.02 | 0.76 ± 0.01 | 0.70 ± 0.02 |
| Node2Vec | **0.84** ± 0.01 | 0.81 ± 0.01 | **0.76** ± 0.01 | **0.83** ± 0.01 | **0.81** ± 0.01 | **0.75** ± 0.01 |
| DeepWalk | **0.84** ± 0.02 | **0.82** ± 0.01 | **0.76** ± 0.01 | 0.82 ± 0.02 | **0.81** ± 0.01 | **0.75** ± 0.01 |
| VGAE | 0.65 ± 0.01 | 0.63 ± 0.01 | 0.61 ± 0.02 | 0.63 ± 0.02 | 0.61 ± 0.01 | 0.59 ± 0.02 |
| DGI | 0.54 ± 0.03 | 0.50 ± 0.07 | 0.50 ± 0.06 | 0.44 ± 0.07 | 0.40 ± 0.12 | 0.39 ± 0.11 |
| Random | 0.17 ± 0.01 | 0.18 ± 0.01 | 0.19 ± 0.01 | 0.14 ± 0.01 | 0.14 ± 0.01 | 0.14 ± 0.01 |

Table 39: Classification experiments on different masking rates for the MAR scenario on the Cora dataset. The mean and standard deviation over 10 iterations are reported. The best and second-best in each metric, for each masking rate and each classifier, are highlighted in **bold** and underlined, respectively.

| Rates | Accuracy | | | F1 Score | | |
|---|---|---|---|---|---|---|
| | 0.2 | 0.5 | 0.8 | 0.2 | 0.5 | 0.8 |
| **GCN** | | | | | | |
| FUSE | **0.80** ± 0.01 | **0.78** ± 0.02 | **0.76** ± 0.02 | **0.79** ± 0.01 | **0.76** ± 0.01 | **0.74** ± 0.02 |
| Node2Vec | 0.76 ± 0.05 | 0.75 ± 0.02 | 0.66 ± 0.02 | 0.74 ± 0.06 | 0.73 ± 0.02 | 0.63 ± 0.02 |
| DeepWalk | 0.78 ± 0.02 | 0.75 ± 0.03 | 0.68 ± 0.03 | 0.76 ± 0.03 | 0.74 ± 0.03 | 0.65 ± 0.04 |
| VGAE | 0.77 ± 0.02 | 0.73 ± 0.01 | 0.64 ± 0.03 | 0.75 ± 0.02 | 0.72 ± 0.01 | 0.61 ± 0.03 |
| DGI | 0.30 ± 0.03 | 0.32 ± 0.05 | 0.32 ± 0.09 | 0.08 ± 0.03 | 0.12 ± 0.08 | 0.16 ± 0.11 |
| Random | 0.48 ± 0.03 | 0.40 ± 0.03 | 0.29 ± 0.04 | 0.45 ± 0.03 | 0.36 ± 0.03 | 0.24 ± 0.03 |
| **GAT** | | | | | | |
| FUSE | 0.84 ± 0.02 | **0.80** ± 0.01 | **0.75** ± 0.02 | **0.83** ± 0.02 | 0.78 ± 0.01 | **0.74** ± 0.02 |
| Node2Vec | 0.84 ± 0.02 | **0.80** ± 0.02 | 0.73 ± 0.02 | **0.83** ± 0.02 | **0.79** ± 0.02 | 0.71 ± 0.02 |
| DeepWalk | **0.85** ± 0.01 | **0.80** ± 0.02 | 0.74 ± 0.02 | **0.83** ± 0.02 | **0.79** ± 0.02 | 0.72 ± 0.02 |
| VGAE | 0.77 ± 0.02 | 0.73 ± 0.01 | 0.69 ± 0.01 | 0.75 ± 0.02 | 0.72 ± 0.01 | 0.68 ± 0.02 |
| DGI | 0.61 ± 0.10 | 0.68 ± 0.04 | 0.59 ± 0.06 | 0.58 ± 0.13 | 0.65 ± 0.06 | 0.54 ± 0.07 |
| Random | 0.64 ± 0.02 | 0.49 ± 0.03 | 0.32 ± 0.04 | 0.62 ± 0.03 | 0.44 ± 0.04 | 0.24 ± 0.04 |
| **SAGE** | | | | | | |
| FUSE | 0.85 ± 0.01 | 0.80 ± 0.02 | 0.74 ± 0.02 | 0.83 ± 0.02 | 0.79 ± 0.02 | 0.72 ± 0.02 |
| Node2Vec | **0.86** ± 0.01 | **0.83** ± 0.01 | 0.76 ± 0.02 | **0.85** ± 0.01 | **0.82** ± 0.01 | 0.73 ± 0.03 |
| DeepWalk | 0.85 ± 0.01 | **0.83** ± 0.01 | **0.77** ± 0.01 | 0.83 ± 0.01 | 0.81 ± 0.01 | **0.75** ± 0.02 |
| VGAE | 0.75 ± 0.02 | 0.71 ± 0.02 | 0.65 ± 0.02 | 0.73 ± 0.02 | 0.69 ± 0.02 | 0.61 ± 0.03 |
| DGI | 0.55 ± 0.05 | 0.57 ± 0.05 | 0.53 ± 0.05 | 0.47 ± 0.08 | 0.48 ± 0.08 | 0.43 ± 0.06 |
| Random | 0.49 ± 0.03 | 0.35 ± 0.02 | 0.25 ± 0.02 | 0.43 ± 0.03 | 0.25 ± 0.03 | 0.17 ± 0.01 |
| **MLP** | | | | | | |
| FUSE | 0.81 ± 0.01 | 0.76 ± 0.01 | 0.71 ± 0.02 | 0.79 ± 0.01 | 0.74 ± 0.02 | 0.69 ± 0.02 |
| Node2Vec | **0.85** ± 0.01 | **0.82** ± 0.01 | 0.75 ± 0.01 | **0.84** ± 0.02 | **0.81** ± 0.01 | 0.73 ± 0.02 |
| DeepWalk | **0.85** ± 0.01 | 0.81 ± 0.01 | **0.76** ± 0.02 | 0.83 ± 0.01 | 0.80 ± 0.01 | **0.74** ± 0.03 |
| VGAE | 0.63 ± 0.02 | 0.63 ± 0.02 | 0.60 ± 0.01 | 0.61 ± 0.01 | 0.61 ± 0.03 | 0.58 ± 0.02 |
| DGI | 0.50 ± 0.06 | 0.48 ± 0.09 | 0.49 ± 0.04 | 0.41 ± 0.09 | 0.35 ± 0.14 | 0.39 ± 0.06 |
| Random | 0.18 ± 0.01 | 0.18 ± 0.01 | 0.18 ± 0.01 | 0.14 ± 0.02 | 0.14 ± 0.01 | 0.14 ± 0.01 |

Table 40: Classification experiments on different masking rates for the MNAR scenario on the Cora dataset. The mean and standard deviation over 10 iterations are reported. The best and second-best in each metric, for each masking rate and each classifier, are highlighted in **bold** and underlined, respectively.

| Rates | Accuracy | | | F1 Score | | |
|---|---|---|---|---|---|---|
| | 0.2 | 0.5 | 0.8 | 0.2 | 0.5 | 0.8 |
| **GCN** | | | | | | |
| FUSE | **0.66** ± 0.01 | **0.67** ± 0.01 | **0.59** ± 0.01 | **0.63** ± 0.01 | **0.64** ± 0.01 | **0.55** ± 0.01 |
| Node2Vec | 0.58 ± 0.02 | 0.54 ± 0.01 | 0.46 ± 0.02 | 0.52 ± 0.02 | 0.50 ± 0.01 | 0.42 ± 0.02 |
| DeepWalk | 0.57 ± 0.02 | 0.53 ± 0.01 | 0.44 ± 0.01 | 0.52 ± 0.02 | 0.50 ± 0.01 | 0.41 ± 0.01 |
| VGAE | 0.54 ± 0.02 | 0.50 ± 0.01 | 0.42 ± 0.02 | 0.50 ± 0.02 | 0.46 ± 0.01 | 0.38 ± 0.02 |
| DGI | 0.30 ± 0.07 | 0.32 ± 0.06 | 0.32 ± 0.03 | 0.19 ± 0.09 | 0.20 ± 0.09 | 0.25 ± 0.04 |
| Random | 0.34 ± 0.03 | 0.28 ± 0.02 | 0.24 ± 0.02 | 0.32 ± 0.03 | 0.26 ± 0.02 | 0.21 ± 0.02 |
| **GAT** | | | | | | |
| FUSE | **0.72** ± 0.01 | **0.68** ± 0.01 | **0.59** ± 0.01 | 0.68 ± 0.01 | **0.64** ± 0.01 | **0.55** ± 0.01 |
| Node2Vec | 0.71 ± 0.02 | 0.65 ± 0.01 | 0.56 ± 0.01 | **0.69** ± 0.02 | 0.62 ± 0.01 | 0.53 ± 0.01 |
| DeepWalk | 0.71 ± 0.01 | 0.64 ± 0.01 | 0.55 ± 0.02 | 0.67 ± 0.01 | 0.61 ± 0.01 | 0.52 ± 0.01 |
| VGAE | 0.61 ± 0.02 | 0.56 ± 0.02 | 0.47 ± 0.02 | 0.57 ± 0.02 | 0.52 ± 0.01 | 0.43 ± 0.02 |
| DGI | 0.49 ± 0.03 | 0.48 ± 0.02 | 0.45 ± 0.02 | 0.42 ± 0.05 | 0.43 ± 0.02 | 0.40 ± 0.02 |
| Random | 0.48 ± 0.02 | 0.40 ± 0.01 | 0.28 ± 0.02 | 0.45 ± 0.02 | 0.37 ± 0.01 | 0.25 ± 0.01 |
| **SAGE** | | | | | | |
| FUSE | **0.72** ± 0.01 | **0.67** ± 0.01 | **0.58** ± 0.01 | **0.69** ± 0.01 | **0.63** ± 0.01 | **0.54** ± 0.01 |
| Node2Vec | 0.70 ± 0.01 | 0.66 ± 0.01 | 0.57 ± 0.01 | 0.66 ± 0.01 | 0.62 ± 0.01 | **0.54** ± 0.02 |
| DeepWalk | 0.71 ± 0.01 | 0.66 ± 0.01 | 0.57 ± 0.01 | 0.67 ± 0.01 | 0.62 ± 0.01 | **0.54** ± 0.01 |
| VGAE | 0.57 ± 0.02 | 0.50 ± 0.01 | 0.44 ± 0.02 | 0.51 ± 0.02 | 0.46 ± 0.01 | 0.40 ± 0.01 |
| DGI | 0.45 ± 0.03 | 0.46 ± 0.02 | 0.42 ± 0.01 | 0.38 ± 0.03 | 0.40 ± 0.02 | 0.35 ± 0.02 |
| Random | 0.38 ± 0.03 | 0.29 ± 0.01 | 0.22 ± 0.01 | 0.33 ± 0.02 | 0.25 ± 0.01 | 0.19 ± 0.01 |
| **MLP** | | | | | | |
| FUSE | **0.72** ± 0.01 | **0.66** ± 0.01 | **0.57** ± 0.01 | 0.67 ± 0.01 | 0.62 ± 0.01 | **0.53** ± 0.01 |
| Node2Vec | **0.72** ± 0.02 | 0.65 ± 0.01 | 0.56 ± 0.01 | **0.69** ± 0.02 | 0.62 ± 0.01 | **0.53** ± 0.02 |
| DeepWalk | 0.71 ± 0.01 | **0.66** ± 0.01 | 0.55 ± 0.01 | 0.68 ± 0.02 | **0.63** ± 0.01 | 0.52 ± 0.01 |
| VGAE | 0.42 ± 0.02 | 0.40 ± 0.01 | 0.38 ± 0.01 | 0.38 ± 0.01 | 0.37 ± 0.01 | 0.36 ± 0.01 |
| DGI | 0.39 ± 0.07 | 0.41 ± 0.03 | 0.37 ± 0.04 | 0.31 ± 0.09 | 0.34 ± 0.05 | 0.30 ± 0.05 |
| Random | 0.18 ± 0.01 | 0.17 ± 0.01 | 0.17 ± 0.01 | 0.17 ± 0.01 | 0.16 ± 0.01 | 0.16 ± 0.01 |

Table 41: Classification experiments on different masking rates for the MCAR scenario on the CiteSeer dataset. The mean and standard deviation over 10 iterations are reported. The best and second-best in each metric, for each masking rate and each classifier, are highlighted in **bold** and underlined, respectively.

| Rates | Accuracy | | | F1 Score | | |
|---|---|---|---|---|---|---|
| | 0.2 | 0.5 | 0.8 | 0.2 | 0.5 | 0.8 |
| **GCN** | | | | | | |
| FUSE | **0.68** ± 0.01 | **0.68** ± 0.01 | **0.58** ± 0.01 | **0.64** ± 0.01 | **0.64** ± 0.01 | **0.55** ± 0.01 |
| Node2Vec | 0.57 ± 0.01 | 0.54 ± 0.01 | 0.42 ± 0.02 | 0.51 ± 0.02 | 0.51 ± 0.01 | 0.39 ± 0.02 |
| DeepWalk | 0.58 ± 0.02 | 0.54 ± 0.02 | 0.42 ± 0.03 | 0.52 ± 0.02 | 0.51 ± 0.02 | 0.40 ± 0.02 |
| VGAE | 0.54 ± 0.03 | 0.48 ± 0.02 | 0.41 ± 0.03 | 0.51 ± 0.03 | 0.45 ± 0.02 | 0.38 ± 0.03 |
| DGI | 0.29 ± 0.10 | 0.33 ± 0.07 | 0.32 ± 0.02 | 0.18 ± 0.13 | 0.21 ± 0.10 | 0.23 ± 0.05 |
| Random | 0.34 ± 0.02 | 0.26 ± 0.01 | 0.23 ± 0.02 | 0.32 ± 0.01 | 0.24 ± 0.01 | 0.21 ± 0.02 |
| **GAT** | | | | | | |
| FUSE | **0.72** ± 0.01 | **0.68** ± 0.01 | **0.59** ± 0.01 | **0.68** ± 0.01 | **0.64** ± 0.01 | **0.54** ± 0.01 |
| Node2Vec | 0.71 ± 0.02 | 0.65 ± 0.02 | 0.54 ± 0.03 | 0.67 ± 0.02 | 0.61 ± 0.01 | 0.51 ± 0.02 |
| DeepWalk | 0.71 ± 0.01 | 0.65 ± 0.02 | 0.54 ± 0.02 | 0.67 ± 0.01 | 0.61 ± 0.02 | 0.51 ± 0.02 |
| VGAE | 0.62 ± 0.01 | 0.57 ± 0.01 | 0.47 ± 0.01 | 0.58 ± 0.02 | 0.54 ± 0.01 | 0.43 ± 0.01 |
| DGI | 0.51 ± 0.02 | 0.48 ± 0.03 | 0.44 ± 0.03 | 0.45 ± 0.02 | 0.42 ± 0.04 | 0.39 ± 0.03 |
| Random | 0.48 ± 0.02 | 0.38 ± 0.02 | 0.27 ± 0.02 | 0.44 ± 0.02 | 0.35 ± 0.02 | 0.25 ± 0.02 |
| **SAGE** | | | | | | |
| FUSE | **0.72** ± 0.01 | **0.67** ± 0.01 | **0.58** ± 0.01 | **0.68** ± 0.01 | **0.63** ± 0.01 | **0.54** ± 0.01 |
| Node2Vec | 0.71 ± 0.01 | 0.66 ± 0.01 | 0.57 ± 0.02 | 0.66 ± 0.02 | 0.62 ± 0.01 | **0.54** ± 0.01 |
| DeepWalk | 0.71 ± 0.01 | 0.66 ± 0.01 | 0.57 ± 0.01 | 0.66 ± 0.01 | 0.62 ± 0.01 | 0.53 ± 0.01 |
| VGAE | 0.57 ± 0.01 | 0.51 ± 0.02 | 0.43 ± 0.01 | 0.52 ± 0.01 | 0.47 ± 0.02 | 0.39 ± 0.02 |
| DGI | 0.47 ± 0.03 | 0.45 ± 0.03 | 0.42 ± 0.03 | 0.40 ± 0.02 | 0.38 ± 0.03 | 0.35 ± 0.04 |
| Random | 0.36 ± 0.02 | 0.29 ± 0.02 | 0.21 ± 0.01 | 0.31 ± 0.02 | 0.25 ± 0.01 | 0.19 ± 0.01 |
| **MLP** | | | | | | |
| FUSE | 0.71 ± 0.01 | **0.66** ± 0.01 | **0.57** ± 0.01 | 0.66 ± 0.01 | 0.62 ± 0.01 | **0.53** ± 0.01 |
| Node2Vec | **0.72** ± 0.01 | **0.66** ± 0.01 | 0.55 ± 0.02 | **0.68** ± 0.01 | 0.62 ± 0.01 | 0.52 ± 0.01 |
| DeepWalk | **0.72** ± 0.01 | **0.66** ± 0.02 | 0.56 ± 0.01 | **0.68** ± 0.01 | **0.63** ± 0.02 | **0.53** ± 0.01 |
| VGAE | 0.42 ± 0.02 | 0.41 ± 0.01 | 0.39 ± 0.01 | 0.40 ± 0.02 | 0.38 ± 0.01 | 0.36 ± 0.01 |
| DGI | 0.42 ± 0.05 | 0.41 ± 0.02 | 0.37 ± 0.03 | 0.35 ± 0.05 | 0.34 ± 0.03 | 0.30 ± 0.05 |
| Random | 0.17 ± 0.01 | 0.18 ± 0.01 | 0.18 ± 0.01 | 0.16 ± 0.01 | 0.16 ± 0.01 | 0.17 ± 0.00 |

Table 42: Classification experiments on different masking rates for the MAR scenario on the Cite-Seer dataset. The mean and standard deviation over 10 iterations are reported. The best and second-best in each metric, for each masking rate and each classifier, are highlighted in **bold** and underlined, respectively.

| Rates | Accuracy | | | F1 Score | | |
|---|---|---|---|---|---|---|
| | 0.2 | 0.5 | 0.8 | 0.2 | 0.5 | 0.8 |
| **GCN** | | | | | | |
| FUSE | **0.68** ± 0.02 | **0.68** ± 0.01 | **0.58** ± 0.01 | **0.64** ± 0.02 | **0.64** ± 0.01 | **0.54** ± 0.01 |
| Node2Vec | 0.58 ± 0.02 | 0.53 ± 0.01 | 0.42 ± 0.02 | 0.52 ± 0.02 | 0.50 ± 0.01 | 0.39 ± 0.02 |
| DeepWalk | 0.59 ± 0.02 | 0.53 ± 0.01 | 0.43 ± 0.02 | 0.54 ± 0.02 | 0.49 ± 0.01 | 0.40 ± 0.02 |
| VGAE | 0.56 ± 0.01 | 0.49 ± 0.02 | 0.38 ± 0.03 | 0.51 ± 0.02 | 0.46 ± 0.01 | 0.35 ± 0.02 |
| DGI | 0.31 ± 0.10 | 0.32 ± 0.05 | 0.28 ± 0.05 | 0.20 ± 0.12 | 0.23 ± 0.07 | 0.18 ± 0.06 |
| SGCL | 0.36 ± 0.01 | 0.26 ± 0.02 | 0.21 ± 0.02 | 0.33 ± 0.01 | 0.24 ± 0.02 | 0.19 ± 0.02 |
| Random | 0.35 ± 0.01 | 0.27 ± 0.01 | 0.21 ± 0.03 | 0.32 ± 0.01 | 0.25 ± 0.01 | 0.19 ± 0.02 |
| **GAT** | | | | | | |
| FUSE | **0.73** ± 0.02 | **0.69** ± 0.01 | **0.58** ± 0.01 | **0.68** ± 0.02 | **0.64** ± 0.01 | **0.54** ± 0.01 |
| Node2Vec | 0.71 ± 0.01 | 0.65 ± 0.02 | 0.55 ± 0.02 | 0.67 ± 0.02 | 0.61 ± 0.01 | 0.52 ± 0.02 |
| DeepWalk | **0.73** ± 0.02 | 0.65 ± 0.02 | 0.55 ± 0.03 | 0.67 ± 0.02 | 0.61 ± 0.02 | 0.52 ± 0.02 |
| VGAE | 0.62 ± 0.01 | 0.56 ± 0.01 | 0.44 ± 0.02 | 0.58 ± 0.02 | 0.52 ± 0.01 | 0.42 ± 0.02 |
| DGI | 0.52 ± 0.03 | 0.48 ± 0.04 | 0.41 ± 0.03 | 0.44 ± 0.03 | 0.42 ± 0.05 | 0.36 ± 0.03 |
| Random | 0.47 ± 0.02 | 0.38 ± 0.01 | 0.25 ± 0.01 | 0.44 ± 0.02 | 0.35 ± 0.02 | 0.22 ± 0.01 |
| **SAGE** | | | | | | |
| FUSE | **0.73** ± 0.02 | **0.67** ± 0.01 | **0.57** ± 0.01 | **0.68** ± 0.02 | **0.63** ± 0.01 | 0.53 ± 0.01 |
| Node2Vec | 0.70 ± 0.01 | 0.66 ± 0.01 | **0.57** ± 0.01 | 0.64 ± 0.01 | 0.62 ± 0.01 | 0.53 ± 0.01 |
| DeepWalk | 0.72 ± 0.02 | **0.67** ± 0.01 | **0.57** ± 0.01 | 0.66 ± 0.03 | 0.62 ± 0.01 | **0.54** ± 0.01 |
| VGAE | 0.57 ± 0.02 | 0.52 ± 0.02 | 0.42 ± 0.02 | 0.51 ± 0.02 | 0.47 ± 0.02 | 0.39 ± 0.02 |
| DGI | 0.48 ± 0.04 | 0.47 ± 0.02 | 0.39 ± 0.02 | 0.40 ± 0.03 | 0.40 ± 0.03 | 0.33 ± 0.04 |
| Random | 0.37 ± 0.03 | 0.27 ± 0.02 | 0.21 ± 0.01 | 0.32 ± 0.03 | 0.24 ± 0.02 | 0.17 ± 0.01 |
| **MLP** | | | | | | |
| FUSE | 0.72 ± 0.02 | **0.66** ± 0.02 | 0.55 ± 0.01 | 0.67 ± 0.02 | 0.62 ± 0.02 | 0.52 ± 0.01 |
| Node2Vec | 0.71 ± 0.01 | **0.66** ± 0.01 | **0.56** ± 0.02 | 0.67 ± 0.01 | 0.62 ± 0.01 | **0.53** ± 0.02 |
| DeepWalk | **0.73** ± 0.01 | **0.66** ± 0.01 | 0.55 ± 0.02 | **0.68** ± 0.02 | **0.63** ± 0.02 | 0.52 ± 0.01 |
| VGAE | 0.42 ± 0.01 | 0.40 ± 0.01 | 0.37 ± 0.01 | 0.38 ± 0.01 | 0.38 ± 0.01 | 0.35 ± 0.01 |
| DGI | 0.41 ± 0.04 | 0.40 ± 0.03 | 0.37 ± 0.03 | 0.33 ± 0.04 | 0.34 ± 0.03 | 0.31 ± 0.03 |
| Random | 0.18 ± 0.02 | 0.18 ± 0.01 | 0.17 ± 0.01 | 0.16 ± 0.02 | 0.16 ± 0.01 | 0.16 ± 0.01 |

Table 43: Classification experiments on different masking rates for the MNAR scenario on the CiteSeer dataset. The mean and standard deviation over 10 iterations are reported. The best and second-best in each metric, for each masking rate and each classifier, are highlighted in **bold** and underlined, respectively.

