# OpenReview forum: "FUSE: Fast Semi-Supervised Node Embedding Learning via Structural and Label-Aware Optimization"
_ICLR.cc/2026/Conference — ICLR 2026 Conference Withdrawn Submission_

### Official Review · Reviewer_3uSk · 2025-11-01

**Soundness:** 3
**Presentation:** 2
**Contribution:** 3
**Rating:** 4
**Confidence:** 2

**Summary:**

This paper proposes FUSE, a fast semi-supervised node embedding framework that jointly optimizes three complementary objectives to learn informative node representations on graphs with missing node features. The learned embeddings can then be subsequently applied to downstream classification tasks.

**Strengths:**

1. The proposed linear-time approximation of modularity gradient optimization achieves comparable or slightly superior performance to baseline methods on most datasets while substantially reducing computational cost.

2. The paper presents extensive experiments, including ablation studies, hyperparameter sensitivity analysis, scalability evaluation, and label-missingness analysis, providing a well-rounded empirical assessment of the proposed framework.

3. The authors address the class imbalance among unlabeled nodes observed in real-world graphs, which enhances the adaptability of FUSE in practical deployment scenarios.

**Weaknesses:**

1. It remains unclear how embeddings generated by FUSE perform on other downstream graph tasks, such as link prediction (which requires multi-hop reasoning) or community detection. Is FUSE inherently tailored to node classification, or can it generalize effectively to other objectives?

2. Since annotation requires a certain cost, an important question is whether the proportion of known labels affects the quality of embeddings generated by FUSE. What is the relationship between the proportion of labeled data and the overall embedding quality?

3. How transferable are the node embeddings generated by FUSE?

**Questions:**

See the weakness section.

---

> ### Author Response · Authors · 2025-11-20
>
> We thank the reviewer for acknowledging the computational cost efficiency of our method while achieving near equivalent results to baseline methods, and for pointing out the method's applicability to graph datasets with extensive class imbalance among the unlabeled nodes. We address their comments below:
> 1. **Other downstream tasks** - We request the reviewer to refer to our response to reviewer 2cEk, specifically the last paragraph of the first point. In summary, Tables 26-37 and Figures 5 and 6 in the updated manuscript show that FUSE remains competitive in the node clustering task.
> 2. **Impact of label imbalance** - This is an important observation and we thank the reviewer to point at this. As shown in Figure 4, the accuracy varies marginally on average for different rates of label availability. It can thus be said that FUSE is slightly affected by the proportion of label availability which can be further supported by the clustering performance plots, Fig 5 and Fig 6. We argue that the embedding quality learnt initially is superior for FUSE compared to the other methods as the intrinsic metric takes lower values while the extrinsic metrics are higher after refinement through GAT. This indicates how FUSE learns from the local structure to the global graph structure gradually irrespective of the proportion of labels available.
> 3. **Transferability of the embeddings** - We are currently experimenting with an extension of FUSE where we use FUSE to generate embeddings in a dynamic graph scenario. In these experiments, we transfer the node embeddings of a graph already learnt at some time $t$ to a graph at time $t+1$, by initializing the known node embeddings at time $t+1$ with the ones already learnt at time $t$. The unknown or new node embeddings are then initialised randomly and learned with the same objective by keeping the known node embeddings fixed. Preliminary experiments, therefore, indicate that the embeddings are transferable.

---

> > ### Comment · Reviewer_3uSk · 2025-11-26
> >
> > Thanks for the rebuttal. After carefully reviewing your response and considering the comments from other reviewers, along with your feedback, I have decided to maintain my original score.

---

### Official Review · Reviewer_sNXf · 2025-11-01

**Soundness:** 2
**Presentation:** 1
**Contribution:** 2
**Rating:** 2
**Confidence:** 3

**Summary:**

The paper presents FUSE, a lightweight framework for fast semi-supervised node embedding on graphs. It aims to bridge the gap between classical shallow embedding techniques (like DeepWalk and Node2Vec) and graph neural networks (GNNs), which, while powerful, are computationally expensive. FUSE proposes a linear fusion approach that efficiently integrates unsupervised structural information with partial label information, generating high-quality embeddings in a fraction of the time required by GNNs. The key insight is that semi-supervised node representations can be derived directly through algebraic fusion of unsupervised embeddings and label propagation signals, without iterative message passing or gradient-based training. This yields an interpretable and efficient alternative to deep GNN models.

**Strengths:**

1. Unified, label-aware embedding objective with linear-time modularity:
The method jointly optimizes (i) an unsupervised modularity term, (ii) a supervised compactness term that pulls labeled nodes of the same class together, and (iii) a semi-supervised label-propagation term with attention—combined in one iterative scheme. This is a clean, well-motivated formulation that directly fuses structural and label signals. Value: principled objective that leverages labels without requiring node features or heavy GNN stacks.

2. Scalable gradient approximation with explicit complexity gains:
The paper proposes a linearized modularity gradient and shows the overall complexity is , explicitly contrasting it with spectral methods. Value: strong algorithmic contribution that explains why the method is fast on large graphs.

3. Clear and sizable runtime advantages, validated empirically:
The paper reports FUSE is ~5× faster on average than Node2Vec/DeepWalk and >7× faster on ArXiv-scale data, while maintaining comparable accuracy. Value: practical efficiency that matters for large or time-sensitive workloads.

**Weaknesses:**

1. Presentation and proofreading quality undermine the submission's professionalism:
The manuscript contains numerous typographical errors, inconsistent mathematical notation, and formatting issues that detract from its clarity and reproducibility—issues that should have been caught prior to submission to a top-tier venue like ICLR. Examples include:
- Spelling errors: "yelding" (p. 1), "formative" instead of "informative" (p. 1).
- Mathematical notation inconsistencies: transpose symbol alternates between ⊺, ^⊺, and ^T; vectors are inconsistently bolded (d, 1, S); subscripts use both Si,: and S_{i,:}; summation indices lack bounds (e.g., ∑_{i,j}).

2. Complete lack of visualizations in the main paper severely limits interpretability and empirical transparency:
Despite strong claims of superior classification accuracy, runtime efficiency, and scalability, the main body contains zero figures or plots — no accuracy vs. label rate curves, no runtime scaling plots, no t-SNE/UMAP embeddings, no ablation bar charts, and no comparison tables. All quantitative results are presumably deferred to appendices or not shown at all in the provided excerpt.

3. Overstated generality:
"the algorithm uses labels if available, but can be adapted to scenarios where labels are completely unavailable with some compromise in performance." overstated. Setting  =  = 0 reduces FUSE to modularity-only optimization, which is not evaluated in unsupervised mode. No ablation shows performance "with some compromise" vs. Node2Vec/DGI. Claiming adaptability without evidence is speculative.

4. No theoretical guarantees for the linearized modularity gradient approximation:
The method replaces the exact degree-weighted term with an unweighted global mean for “stability and efficiency,” but provides no error bounds, convergence analysis, or conditions under which the approximation preserves modularity-driven structure. This limits the conceptual strength of the contribution beyond empirical performance. A theorem or approximation-error analysis (and a small-graph comparison to the exact gradient) would materially strengthen the paper.

**Questions:**

Q1. Approximation guarantees:
The linearized modularity gradient replaces the degree-weighted term with an unweighted global mean “for stability and efficiency.” Can you provide any error bounds, convergence guarantees, or conditions under which this approximation preserves modularity structure?

Q2. Scope of “state-of-the-art” claim:
The abstract says FUSE is “at par with or superior to state-of-the-art,” yet the baseline set is mostly classic shallow/self-supervised methods plus GraFN/ReVAR. Do you consider recent strong semi-supervised GNNs/graph transformers out of scope, or can you expand comparisons to support the SOTA claim?

Q3. Biased vs. uniform walks:
You state that label-biased random walks “preferentially visit labeled nodes,” but Algorithm 2 samples the next node uniformly from neighbors. Is the walk actually biased, and if so, where is the bias implemented? Please reconcile the text and the pseudocode.

---

> ### Author Response · Authors · 2025-11-20
>
> We thank the reviewer for their valuable comments. We address their concerns below.
> # Weaknesses
> 1. **Proofreading quality** - We acknowledge that the initial submission contained several typographical errors, inconsistent notation choices, and formatting oversights. We have carefully proofread the entire manuscript and corrected all spelling errors, standardized the transpose operator, enforced consistent boldface conventions for vectors and matrices, and added missing index bounds in summations. We appreciate the reviewer bringing these matters to our attention and believe that the revised version is now significantly clearer, more consistent, and more polished.
> 2. **Lack of visualization** - We thank the reviewer for addressing this. Though we had the visualizations ready, we did not include them in the main manuscript due to space constraints. However, as per the recommendation, we have decided to add them to the Appendix of our paper. We are presenting the embedding visualizations (both raw and those learned downstream by GNNs during node classification) for both the splits (30-70 and 70-30) corresponding to the datasets Cora and Pubmed starting from Fig 7 to Fig 10, the runtime scaling plots in Fig 2 followed by the ablation bar charts along with the runtimes in Fig 3 and the accuracy vs label rate curves in Fig 4.
> 3. **Overstated generality** - We request the reviewer to refer to Appendix C.2 of our manuscript (Tables 6-9), where we have provided the results in the tables when FUSE runs with **"Only Unsupervised Component"** of the objective. As expected, the performance of FUSE in terms of accuracy and F1 Score does get compromised.
>
>     The best-case accuracy for full FUSE is ~82\% and ~79\% for the 30-70 and 70-30 splits, respectively (both with GAT). The average times taken by full FUSE to generate embeddings for all datasets are ~52s and ~66s, respectively. For the unsupervised-objective-restricted FUSE, the best-case performances are ~71\% and ~75\% for the 30-70 and 70-30 splits, respectively (both for GraphSAGE). The average time taken for the restricted FUSE algorithm is ~17s and ~15s, respectively. Thus, the statement *"the algorithm uses labels if available, but can be adapted to scenarios where labels are completely unavailable with some compromise in performance.''* has an objective basis.
> 4. **Theoretical guarantees** - We thank the reviewer for this suggestion. In response, we have added a theoretical analysis clarifying the behaviour of the linearised modularity gradient. Specifically, we show that the surrogate gradient has a bounded operator norm, ensuring numerical stability and ruling out unbounded updates. We could also show that if $\sum_{i}r_id_i<2m^2$, where $r_i=\sum_{j:v_j\in\mathcal{N}(v_i)}d_i$, then the Fröbenius inner product of the proposed and exact gradients is positive, which indicates that the gradient updates do not point in opposing directions. Stronger theoretical guarantees will require more investigation.
>
> # Questions
> 1. We request the reviewer to refer to our answer to weakness 4.
> 2. Our work is set in a fast, feature-agnostic, partially label-aware setting, where only the graph structure and a small number of labels are available. In this setting, recent semi-supervised GNNs and graph transformers are not directly comparable because they rely on informative node features; when forced to use random features, their performance reduces to the level of the simple GNN classifiers we already include. For fairness, we therefore compare our results against the strongest structure-driven and feature-free embedding methods that can operate under our assumptions (DeepWalk, Node2Vec, DGI). We will clarify that our 'state-of-the-art' claim specifically refers to this class of feature-agnostic, partially label-aware embedding methods.
>    Let's clarify once more that the focus here is on fast semi-supervised embedding generation. To our knowledge, FUSE is one of the first methods that explicitly targets the construction of fast embeddings from scratch in a feature-free regime. This is particularly important for real-world scenarios where embeddings must be recomputed or updated in real-time, such as dynamic recommendation systems, cybersecurity monitoring, or fraud detection pipelines. In such settings, the ability to generate high-quality embeddings significantly faster than existing structure-based methods is a practical advantage. We revised the manuscript to clarify the scope and motivation.
> 3. We request the reviewer to refer to point 2 in our answer to reviewer b6i3.
> Formally, the transition probability is given by:
> $$P(v_{t+1}=u \\mid v_t) =
> \\begin{cases}
> \\frac{1}{|\\mathcal{N}_L(v_t)|}, & \text{if } u \\in \mathcal{N}_L(v_t) \\text{ and labeled\\_count} < L', \\\\[6pt]
> \\frac{1}{|\\mathcal{N}(v_t)|}, & \text{otherwise.}
> \\end{cases}$$
> The walk is label-biased due to conditional neighbor selection, not due to non-uniform probabilities within a chosen set.

---

### Official Review · Reviewer_2cEk · 2025-11-03

**Soundness:** 2
**Presentation:** 2
**Contribution:** 2
**Rating:** 4
**Confidence:** 5

**Summary:**

This paper proposes a node embedding generation algorithm called FUSE (Fast Unified Semi-supervised Node Embedding Learning from Scratch), whose objective function integrates an unsupervised, a semi-supervised and a supervised component. The experimental results show the effectiveness of the proposed FUSE algorithm.

**Strengths:**

S1. FUSE has three components and does not require predefined features.

S2. The experimental results demonstrate the effectiveness of the FUSE algorithm.

**Weaknesses:**

W1. The applicability of the proposed FUSE seems limited. For example, as presented in the last section (DISCUSSION AND CONCLUSION) of the paper, the dynamic graphs are not applicable for the current FUSE algorithm. In addition, the size of the used ArXiV dataset is not large enough to show the scalability of the proposed methods. Also, how about the proposed FUSE algorithm for complex graph datasets, e.g., directed graphs and social networks? Last but not least, how about the proposed FUSE algorithm for other downstream tasks, e.g., link prediction and node clustering?

W2. It seems that the experimental results are not convincing. For instance, Table 1 shows the classification accuracy and F1-score across embedding methods and three classifiers for all the datasets (except ArXiV). Why not present the results for each dataset respectively? The authors should provide the reasons.

W3. The statements of the paper should be reasonable. For example, this paper claims that "FUSE achieves the best or second-best performance across classifiers." However, as shown in Figure 1, DeepWalk, Node2Vec, and VGAE clearly outperform FUSE for 30-70 train-test split with SAGE.

W4. The authors should carefully polish the submission to avoid typos and grammar mistakes, such as "Tables 1 summarize" in the first sentence of the first paragraph of Section 4.3.1.

**Questions:**

W1-W4

---

> ### Author Response · Authors · 2025-11-20
>
> We thank the reviewer for their acknowledgment of the novelty of FUSE's three-component structure without requiring predefined features and their appreciation of the demonstrated effectiveness in the experimental results. We address their comments below:
>
> 1. **Applicability to other tasks and types of graphs** - FUSE targets fast, feature-agnostic, partially label-aware embedding generation, a combination of constraints that can be practically feasible, yet not the primary focus of most existing methods, which are typically either fully unsupervised or rely on informative features. Our motivation stems from scenarios where embeddings must be computed repeatedly and updated quickly. Such situations are common in areas like computational biology, where protein-interaction networks differ slightly across tissues, cell lines or multiple conditions, and in cybersecurity, where communication patterns change continuously and embeddings need to be refreshed rapidly. In these cases, speed is a primary requirement, not a secondary concern.
>         For further scalability analysis, we request the reviewer to refer to our response for the first weakness as pointed out by reviewer b6i3.
> 	In this paper, we principally focus on the quality of node embeddings generated by the FUSE algorithm, which is why the primary task we focused on is downstream node classification. We consider the prospect of link prediction to be too broad for this work, as the current version of FUSE uses node classification-oriented supervision. The framework of the current version of FUSE is limited to static, undirected graphs, although some of the benchmarking datasets, such as ArXiV and MAG, are directed in nature. However, directed networks require their own forms of modularity, and adapting FUSE to those choices would lead to a different methodological direction.
>
>
> 	Upon the reviewer's recommendation, we conducted an experiment on node clustering to evaluate how FUSE performs compared to existing baselines. We gained valuable insights from these experiments.  We measured the DB index, an intrinsic metric, and the ARI and V-Measure scores, extrinsic metrics, for the clustering experiment. The dataset-wise results are presented in Tables 26-37 and Figures 5 and 6 in the updated manuscript. We observe that the embeddings for FUSE have the least DB index, indicating superior cluster separation in the learned embeddings for most datasets. Moreover, we observe that the embeddings for FUSE have the highest V-measure for all datasets. This indicates that FUSE-initialised classifiers can learn embeddings where clusters are consistent with known class labels.
>
> 2. **Low accuracy and dataset-wise results** - Primarily, we reported the averaged results across datasets, except for ArXiV, due to concerns about making the manuscript too cluttered. However, as recommended by the reviewer, we have added the dataset-wise results to the Appendix to maintain full transparency of the results. These results are presented in Tables 21 to 25 of the updated manuscript.
>
> 3. **Unreasonable statements** - We thank the reviewer for bringing this to our attention. The statement in the paper was too strong as written. Our intention was not to claim that FUSE is always the best or second-best in accuracy for every classifier and split.
> We will revise the statement to more accurately reflect our results. A precise formulation is: *“FUSE achieves competitive accuracy across datasets and classifiers while offering substantially lower embedding generation times, providing a favorable accuracy–runtime trade-off.”* This better captures the empirical behavior: although FUSE does not win every accuracy comparison, it consistently delivers comparable performance at two to five times lower runtime. Since the goal of FUSE is to generate fast, feature-agnostic, partially label-aware embeddings, we believe this trade-off is important and should be evaluated alongside accuracy.
>
> 4. **Typos and grammatical errors** - Thank you for highlighting the need for closer proofreading. We apologize for the grammatical errors and typos present in the initial submission. We have meticulously reviewed and corrected the manuscript and believe the revised version is now significantly clearer.

---

### Official Review · Reviewer_b6i3 · 2025-11-11

**Soundness:** 2
**Presentation:** 2
**Contribution:** 2
**Rating:** 4
**Confidence:** 4

**Summary:**

The main contribution of the paper is FUSE, a fast semi-supervised embedding framework for node classification when the node features are not available. It jointly optimizes for three objectives, an unsupervised objective based on a linearized modularity surrogate, a supervised compactness objective that clusters labeled nodes and a semi supervised propagation term using random walk label spreading with attention weighted similarity, with normalization of the emebddings. The authors show the gradient approximation and argue that it is numerically stable. Experimental results are provided on 6 graph datasets.

**Strengths:**

1. The setting that the authors look at, of no node features where the label information needs to be integrated is interesting and practically useful for the graph learning community.


2. The algorithm and update proposed by the authors is straightforward and should be easy to implement.


3. The authors provide runtime and robustness analysis in their results, which is appreciated.

**Weaknesses:**

1. The paper has a heavy focus on scalability as one of the advantages, however the graph datasets used in the paper are all small scale datasets, with the largest dataset Arxiv (from OGB) only being of the order of a hundred thousand nodes which is not enough to demonstrate real scalability in practical settings. There are multiple other large scale datasets from the OGB suite itself, of the order of a million nodes which can be used to demonstrate scalability


2. Some of the details around the method and experimentation are unclear.  The paper mentions that the random walk “preferentially visit labeled nodes” (L223), however the algorithm 2 uniformly samples the neighbors without biasing the probabilities, which is different from how the method is described.

3. The complexity details mentioned by the authors does not include the cost of orthonormalization, which is not cheap and the cost can be high when the graphs are sparse.


4. Several baselines used by the authors require features and the authors inject random features while keeping the architecture fixed, which is known to show much worse performance than, say using identity features (which is not included in the paper). This would not be a fair comparison and these baselines will show significantly lower results than what they actually are.


5. The empirical results of FUSE are close or within the error range of DeepWalk/Node2vec and hence the margin of improvement is small

**Questions:**

See weaknesses section above.

---

> ### Author Response · Authors · 2025-11-20
>
> We thank the reviewer for acknowledging that the setting is interesting and for their appreciation of the ease of implementation and robustness of the results. We address their comments below.
> 1. We performed additional scalability analyses on two large datasets: MAG (\~736K nodes, \~8M citation edges) and ogbn products (\~2.45M nodes, \~61.9M edges) using a 30-70 split (70\% label masking). Since DeepWalk and Node2Vec consistently achieve the strongest accuracy and F1 scores among the baseline embedding methods, and because their performance remains stable even with shorter walk lengths (5) and fewer walks (10), we report comparisons against DeepWalk using these reduced parameters. We also include the given embedding as a high-end benchmark. We exclude GAT from these comparisons due to its high computational overhead and instead evaluate against GCN and GraphSAGE.
>   Our observations (Tables 14 and 15) remain consistent with our earlier findings:
>     * FUSE remains faster on MAG compared to DeepWalk, with the unsupervised variant being at least three times faster. On the ogbn products dataset, the unsupervised version of FUSE completes in approximately 2.5 hours and the full version takes little more than 10 hours. In contrast, DeepWalk could not complete within 24 hours using the standard Python implementation with a single CPU worker and no GPU.
>     * While FUSE is fast, in few cases, it sacrifices accuracy and F1-score, and this performance gap becomes more pronounced on larger datasets. Therefore, the applicability of FUSE is most relevant in feature-agnostic settings where fast embedding generation is the primary requirement.
>     * FUSE is compatible with GCN but performs less effectively with GraphSAGE for larger datasets.
>
> 2. We thank the reviewer for pointing out the discrepancy. In the proposed FUSE algorithm, random walks are label-conditioned, in the sense that whenever a labeled neighbor is available and the number of labeled steps taken so far is below a limit $L'$, the walk preferentially selects from labeled neighbors. Otherwise, it samples uniformly from all neighbors. This results in a controlled bias toward labeled regions of the graph. We revised Algorithm 2 in the manuscript to clarify this point.
>
> 3. We conducted an experiment in which, instead of orthonormalizing the embedding matrix after every iteration, we orthonormalized it at the very end. From the results (See Tables 10 and 11 in the manuscript), we observe that orthonormalizing at the very end instead of every iteration indeed takes a slightly lesser time (the margin is more when performed on a stronger CPU) but degrades the performance in some cases like ArXiV. Hence we recommend using orthonormalization per iteration which incurs a cost of $O(nk^{2})$ which we have updated in the portion of the time complexity analysis in the actual manuscript. The total time taken by FUSE is therefore $O\left(|E|k + nk + n d_{\max} k + w\ell + nk^{2}\right)$ where the $O(nk^{2})$ term is typically dominated by the sparse matrix multiplication $O(|E|k)$ for moderate $k$.
>
> 4. We thank the reviewer for pointing this out. The design choice of using random feature initialization is intentional and stems from the motivation of addressing feature-scarce graph settings. Using an identity matrix as the feature matrix would make the input dimensionality equal to the number of nodes, thereby defeating the purpose of learning a compact, low-dimensional representation. More importantly, this would significantly increase both the memory footprint and the runtime, and make the embedding dimensionality inconsistent across algorithms. For example, in the case of VGAE, we found that the authors of that paper explicitly recommend identity initialisation under the feature-unavailability scenario, and we adhered to that. However, this also made the embeddings much slower to compute, leading to memory failure for VGAE in some cases, which negatively impacts the scalability of these algorithms.
>
> 5. We would like to emphasise that our performance discussion always considers both accuracy and runtime. The focus of this study is to build a fast, feature-agnostic, label-aware embedding generation algorithm.
>     * Under default parameter settings of all models, FUSE, DeepWalk, and Node2Vec produce around **82\% accuracy** under the 70-30 train-test split (Please refer to Table 1).
>     * Under the 30-70 split, Node2Vec and DeepWalk produce around **79\%** and FUSE produces around **78\% accuracy** (Table 1).
> In terms of average runtime, FUSE is five to six times faster under the default parameter setting. Even for the limited walk length of Node2Vec and DeepWalk, we observe that FUSE is at least around 2-3 times faster on the denser and larger datasets. For accuracy and F1-score, the margin is small; however, we argue that FUSE achieves equivalent performance with a much lower runtime and memory footprint.

---

### Note · Authors · 2026-01-13

**Comment:**

We would like to withdraw our manuscript to strengthen our theoretical analysis and perform additional experiments on other downstream tasks before submission at a different venue.

**Withdrawal Confirmation:**

I have read and agree with the venue's withdrawal policy on behalf of myself and my co-authors.